# Generalizing from SIMPLE to HARD Visual Reasoning: Can We Mitigate Modality Imbalance in VLMs?

Simon Park [* 1]   Abhishek Panigrahi [* 1]   Yun Cheng [* 1]   Dingli Yu [1]   Anirudh Goyal [2]   Sanjeev Arora [1]

## Abstract

Vision Language Models (VLMs) are impressive at visual question answering and image captioning. But they underperform on multi-step visual reasoning—even compared to LLMs on the same tasks presented in text form—giving rise to perceptions of *modality imbalance* or *brittleness*. Towards a systematic study of such issues, we introduce a synthetic framework for assessing the ability of VLMs to perform algorithmic visual reasoning, comprising three tasks: *Table Readout*, *Grid Navigation*, and *Visual Analogy*. Each has two levels of difficulty, SIMPLE and HARD, and even the SIMPLE versions are difficult for frontier VLMs. We propose strategies for training on the SIMPLE version of tasks that improve performance on the corresponding HARD task, i.e., simple-to-hard (S2H) generalization. This controlled setup, where each task also has an equivalent text-only version, allows a quantification of the modality imbalance and how it is impacted by training strategy. We show that 1) explicit image-to-text conversion is important in promoting S2H generalization on images, by transferring reasoning from text; 2) conversion can be internalized at test time. We also report results of mechanistic study of this phenomenon. We identify measures of gradient alignment that can identify training strategies that promote better S2H generalization. Ablations highlight the importance of chain-of-thought [1].

## 1. Introduction

Many Vision Language Models (VLMs) (e.g., LLaVA-series (Liu et al., 2023c;b; 2024a)) fuse an LLM with visual encoders which allows them to harness the impressive reasoning abilities of pre-trained LLMs towards solving visual reasoning tasks (Monajatipoor et al., 2023; Carbune et al., 2024; Zhang et al., 2024a). However, VLMs are usually felt to exhibit more *brittle* reasoning than the underlying LLM, and recent works have tried to understand this as a **modality imbalance** problem (Peng et al., 2022; Huang et al., 2022; Fan et al., 2023; Wei et al., 2024). For example, presenting the task in an image form can lead to a lower performance than when the same task is presented in a text form (Zhang et al., 2023; 2024c; Wang et al., 2024b; Zhang et al., 2024d; Fu et al., 2024). Mitigating this modality imbalance is still an open problem.

Here, we introduce a concrete methodology to precisely study such issues. First, we design visual tasks where the image information relevant to the task can also be represented as text (e.g., LaTeX code). This allows a direct comparison of the effect of training strategies in individual modalities and combinations. Second, to allow a clear comparison of different training strategies, we measure the *brittleness* of learning with **simple-to-hard (S2H) generalization**, where models are trained on SIMPLE examples of a task and evaluated on HARD examples.

We create a set of synthetic tasks[2] that involve algorithmic visual reasoning (Ghosal et al., 2024; Cherian et al., 2023; Zhang et al., 2024b): *Table Readout* (reading out table entries in an order specified visually), *Grid Navigation* (finding valid paths through grid-like structures while avoiding obstacles), and *Visual Analogy* (identifying logical patterns across sets of abstract visual examples and applying analogical reasoning). Each task requires many reasoning steps while dynamically shifting attention over a sequence of small regions in the image. SIMPLE and HARD examples differ in the length and complexity of the necessary reasoning steps.

The SIMPLE tasks are difficult for current frontier VLMs such as GPT-4o and Claude-3.5 Sonnet (Achiam et al., 2023; Anthropic, 2024) (Appendix I.1). Since we work with smaller open-parameter models, our methodology consists of using supervised training to precisely inject capability at a task in one modality and then study how variations in training affect the gap in S2H generalization between modalities.

---

[*]Equal contribution   [1]Princeton Language and Intelligence, Princeton University [2]Meta AI. Correspondence to: <{juhyunp, ap34, yc6206}@princeton.edu>.

*Proceedings of the 42nd International Conference on Machine Learning*, Vancouver, Canada. PMLR 267, 2025. Copyright 2025 by the author(s).

[1]Code is available at VLM-S2H PLI codebase

[2]Creating such tasks was more nontrivial than expected, for reasons described in Appendix C.6.

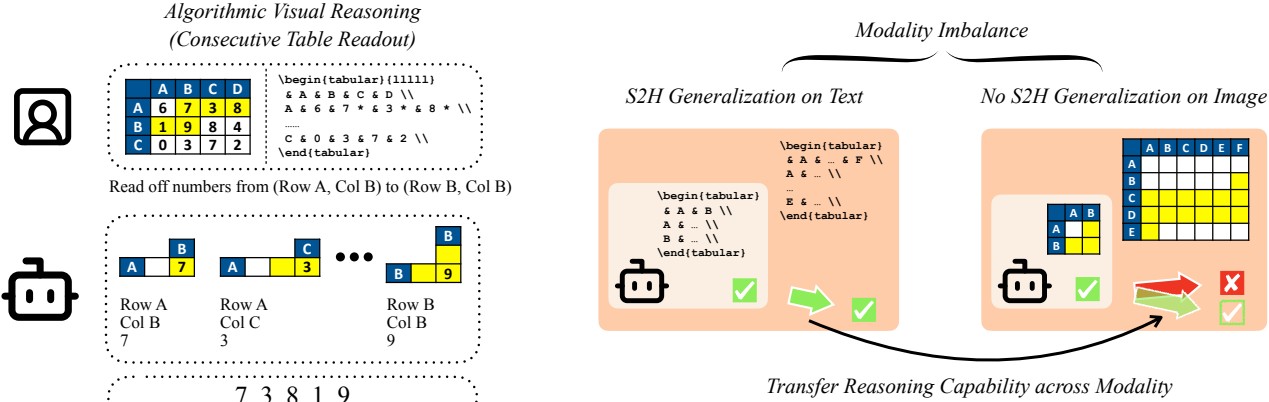

*Figure 1.* **(Left) Example Data Point for *Consecutive Table Readout*.** Input table can be provided as an image or LaTeX code. The task is to sequentially read numbers from a start cell to an end cell in row major order. **(Right) Illustration of Key Concepts** using examples from *Consecutive Table Readout*. We observe that current models can S2H generalize on text – when trained to read short sequences from small LaTeX-formatted tables, the models can read longer paths from larger tables, also provided in LaTeX code. However, they fail to length-generalize on images. To address the generalization gap and imbalanced learning of different modalities, our goal is to transfer the generalization behavior from text to image modality.

Since the tasks are difficult for frontier VLMs, we expect the takeaways from our study to be of broader interest.

**Illustrative example of *Consecutive Table Readout*:** Given a table of numbers and indices of two table cells $(i, j)$ and $(k, l)$, the model needs to output every table entry between these two cells in a row-major order. The input table can be provided as an image or as text (i.e., LaTeX code), allowing the kind of study sketched in Figure 1. In the SIMPLE task, the length of the output sequence is 5 to 10, whereas in the HARD task, it can be as long as 30. Therefore, S2H generalization here is a type of *length generalization*, a well-studied concept in LLMs (Zhou et al., 2024a). SFT on $8 \times 10^4$ SIMPLE-text examples yields 80% accuracy on HARD-text examples. However, training on the SIMPLE-image examples results in only 20% accuracy on the HARD-image examples. The 60%p difference is a measure of the *modality gap* or *modality imbalance*.

### 1.1. Paper Overview

We study training strategies that incorporate various types of supervision: text-based, image-based, and combinations of the two (Section 2.4). We find that the most reliable way to alleviate the gap is to teach the model **image reasoning via text conversion** — explicitly extracting information from the image in text form before generating the solution using CoT. Specifically, we find: (i) for tasks where the model exhibits S2H generalization in the text modality, training on **image reasoning via text conversion** greatly helps to mitigate the gap (Section 3); (ii) for tasks where the S2H generalization failed in both modalities, applying the idea from (i) while also injecting reasoning capability on the HARD task in the text modality leads to S2H gen-

eralization in the image modality (Section 4). The findings in (ii) should be interpreted as suggesting that simple image-to-text conversion could be a promising intervention to reduce modality imbalance in future VLMs whose base LLM does exhibit S2H generalization in the text modality.

A surprising finding is that even though explicitly training on image-to-text conversion seems necessary for S2H generalization, the final trained model can generate the correct solution without explicitly extracting the image content as text: the image-to-text conversion skill gets internalized! (This also greatly reduces the inference-time cost.) Therefore, we try to understand the effectiveness of this key intervention at the level of training gradients. We find that gradients from SIMPLE-image reasoning examples can help reduce loss on HARD-image inputs with the above intervention (Section 5); this gradient alignment merits further study.

On tasks where we need to inject reasoning capability on the HARD task, our findings about gradients inspired a more effective two-phase training (Section 4.3). The first phase teaches the model to do image reasoning via text conversion on a few SIMPLE examples. We find that inclusion of this phase substantially improves gradient alignment in the earlier phases of training, when gradients have larger norms, which allows for more effective S2H generalization on the image modality. This finding is in accord with previous empirical evidence that highlights the importance of visual-language alignment in VLM training (Fan et al., 2024).

(a) *Table Readout*        (b) *Grid Navigation*        (c) *Visual Analogy*

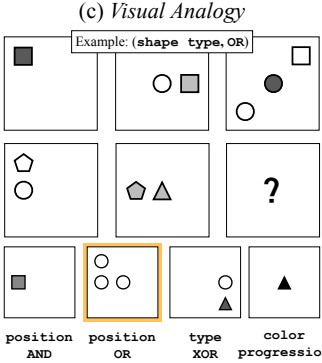

*Figure 2.* **Illustration of our synthetic tasks:** *Table Readout* involves reading numbers along a specified path in a table. *Grid Navigation* involves navigating a grid to collect objects while avoiding obstacles. *Visual Analogy* involves solving analogical reasoning queries using two in-context examples. **More details on *Visual Analogy*:** For the example of *Visual Analogy* above, we only include one in-context example for simplicity and provide annotations for clarity. In the first row, the first cell contains a rectangle, whereas the second and third cells contain a circle and a rectangle. Therefore, the in-context example is consistent with applying the OR relation along the shape type domain. The model then needs to identify the correct option that corresponds to applying the OR relation to the first two cells of the query along some (potentially different) domain. See Appendix C for non-annotated example images that are provided to the model.

## 2. General Setup

### 2.1. Model

In line with Shi et al. (2025) that show the benefit of combining multiple image encoders in VLMs, we trained *Eagle-X2-Llama3-8B*, a variation of *Eagle-X5* that uses Llama3-8B-Instruct (Dubey et al., 2024) as the LLM backbone and CLIP-448 (Radford et al., 2021) and ConvNeXt (Liu et al., 2022) as visual encoders. Since the original paper found only minor benefit beyond the two encoders, we do not use all five visual encoders. See Appendix D for more details on the training. In Appendix F, we replicate some of the experiments on *Qwen2.5-VL-3B-Instruct* and *7B-Instruct* (Bai et al., 2025) and observe consistent results.

### 2.2. Tasks

We briefly describe the tasks that we consider and the SIMPLE and HARD setup of each task below (summarized in Table 1; fully detailed in Appendix C).

- *Table Readout*: The model sequentially reads numbers along a highlighted path in a table (given in either image or its LaTeX code). SIMPLE examples consist of 1–4 linear segments in spiral or sinusoidal path patterns with an average length of 12 (Figure 32). HARD examples consist of > 4 linear segments, featuring longer and arbitrary compositions of spiral or sinusoidal path patterns with an average length of 35 (Figure 33).

- *Consecutive Table Readout*: This is a variant of *Table Readout*, modified to make the reasoning simpler[3]. The model sequentially reads numbers in a *row major*

*order*. The number of cells to read in SIMPLE and HARD examples is respectively 5-10 and 25-30. Only for this task, we additionally prepare a set of MEDIUM difficulty level, where the number of cells to read is 15-20. Training on SIMPLE examples and evaluating on MEDIUM examples can also measure S2H generalization.

- *Grid Navigation*: The model navigates in a 2D grid (given in either image or its LaTeX code) from a designated start cell to an end cell while collecting all specified objects and avoiding obstacles. SIMPLE examples contain 1–2 objects and 1 type of obstacle (Figure 34). HARD examples involve ≥ 2 distinct objects and ≥ 3 types of obstacles (Figure 35). The task can be solved by depth-first search (DFS). Recent works (Kim et al., 2024; Wu et al., 2024a; Wang et al., 2024b) explored similar synthetic tasks in LLM and VLM evaluation.

- *Visual Analogy*: The model reasons about attributes and relations between geometric figures in a puzzle (given in the image or text description). It analyzes two in-context examples and applies an analogous reasoning to choose 1 from 4 options to complete the query. SIMPLE puzzles have examples and query vary along the *same* attribute following a common relation (Figure 36). HARD puzzles have examples and query vary along *different* attributes following a relation, and the combinations of attribute and relation *held-out* from training (Figure 37). This task is adapted from Barrett et al. (2018) and Hill et al. (2019).

- *Pattern-Heldout Visual Analogy*: This is a variant of *Visual Analogy*, modified to make the reasoning simpler. See Appendix G.2 for more details.

---

[3]There is a fixed underlying rule for where the next cell should be. The model doesn't need to make a decision at each step.

## 2.3. Training Data

Formally, we let $f : \mathcal{X} \to \mathcal{Z}$ denote a reasoning task, where $\mathcal{X}$ refers to a set of input data, further split into $\mathcal{X}_{\text{SIMPLE}}$ and $\mathcal{X}_{\text{HARD}}$, and $\mathcal{Z}$ refers to a set of answers. Each input $\mathbf{x} \in \mathcal{X}$ can be presented in text format $\mathbf{x}^{(t)}$ or in image format $\mathbf{x}^{(i)}$.

For each pair of data $\mathbf{x}$ and solution $f(\mathbf{x})$, we also create a chain-of-thought reasoning trace, which we denote by $CoT(\mathbf{x})$. We also define a prompt $P_{convert}$ that we optionally prepend at the start of chain-of-thought to signal explicit image-to-text conversion on image input[4]. Hence, our training dataset is defined by input $\mathbf{x}$ (which can be given either as $\mathbf{x}^{(t)}$ or $\mathbf{x}^{(i)}$), chain-of-thought $CoT(\mathbf{x})$, and the final answer $f(\mathbf{x})$.

For each task $f$, we use the same Python script and a fixed template to generate all tuples $(\mathbf{x}^{(t)}, \mathbf{x}^{(i)}, CoT(\mathbf{x}), f(\mathbf{x}))$.

## 2.4. Types of Supervision

Our controlled experiments study the effect of the following types of supervision on SIMPLE examples during training:

(a) **Text** supervision: given a text input $\mathbf{x}^{(t)} \in \mathcal{X}_{\text{SIMPLE}}$, we train on the gold output containing a chain-of-thought trace $CoT(\mathbf{x})$ and the final answer $f(\mathbf{x})$.

(b) **Image** supervision: given an image input $\mathbf{x}^{(i)} \in \mathcal{X}_{\text{SIMPLE}}$, we train on the gold output containing a chain-of-thought trace $CoT(\mathbf{x})$ and the final answer $f(\mathbf{x})$.

(c) **Image-via-Text** supervision: given image input $\mathbf{x}^{(i)} \in \mathcal{X}_{\text{SIMPLE}}$, we train on the gold output containing the conversion prompt $P_{convert}$, converted text $\mathbf{x}^{(t)}$, a chain-of-thought trace $CoT(\mathbf{x})$, and the final answer $f(\mathbf{x})$.

(d) **Text+Image** supervision: we train on an equal mix of **Text** and **Image** supervisions.

(e) **Mix** supervision: we train on an equal mix of **Text**, **Image**, and **Image-via-Text** supervisions.

We train the model on one of the above supervision types with auto-regressive loss ($l$) that takes in the model's logits on an input example and returns the average loss on a selected set of tokens. For example, for **Image** supervision, we will represent the input example as $\{\mathbf{x}^{(i)}, CoT(\mathbf{x}), f(\mathbf{x})\}$, and compute the loss on $\{CoT(\mathbf{x}), f(\mathbf{x})\}$. During the evaluation, we test whether the model predicts $f(\mathbf{x})$ correctly for a given input.

In Section 4, we will adapt some of the above supervision strategies to also include HARD **Text** supervision[5]. The

---

[4]e.g., "Convert the provided image to text"

[5]Identical as **Text** except we use a HARD-text example, i.e., $\mathbf{x}^{(t)} \in \mathcal{X}_{\text{HARD}}$. We note the subtle difference between a "HARD-text example," which refers to the data $\mathbf{x}^{(t)}$, and "HARD **Text**," which is a type of supervision with a prescribed (input, output) structure:

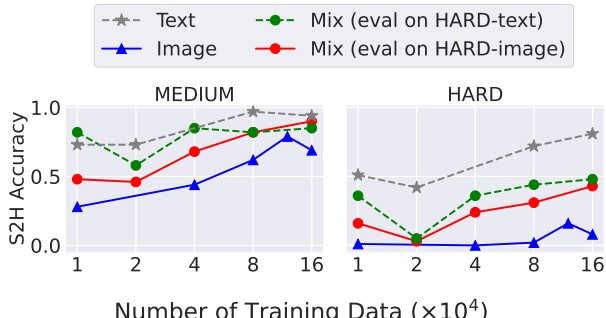

*Figure 3.* **S2H Generalization** of different supervisions for *Consecutive Table Readout* to MEDIUM (left) and HARD (right) examples. S2H generalization on text of **Text** ($\star$) outperforms S2H generalization on image of **Image** ($\blacktriangle$), highlighting modality imbalance. **Mix** ($\bullet$) mitigates this imbalance.

adapted supervision strategies will have a **+** sign appended to represent this additional component (e.g., **Mix+** adapted from **Mix** supervision).

## 3. Modality Imbalance in *Consecutive Table Readout*

We use *Consecutive Table Readout* introduced in Section 1 and Section 2.2 to illustrate the S2H generalization gap between different modalities and propose training strategies needed to address it. We compare different types of supervision by training on the prescribed SIMPLE examples and measuring the improvements on the exact match accuracy[6] on two different difficulty levels: (a) MEDIUM: reading 15–20 consecutive numbers and (b) HARD: 25–30 numbers (more challenging).

To demonstrate the modality imbalance, we compare **Text** and **Image** supervision. Figure 3 shows that the S2H generalization gap between the two is substantial. For HARD, while **Text** supervision achieves $80\%$ accuracy on HARD-text examples, **Image** supervision achieves only $20\%$ on HARD-image examples.

In order to reduce the gap, we consider training strategies that can leverage strong S2H generalization of **Text** supervision to help S2H generalization of **Image** supervision. Two candidates are **Text+Image** supervision, which simply mixes in **Text** and **Image** supervision, and **Image-via-Text** supervision, which trains the model to first convert the image input to its text format and then output the solution. **Text+Image** supervision induces the model to implicitly make the connection that the image and text formats are equivalent, while **Image-via-Text** supervision makes this

---

$\{\mathbf{x}^{(t)}, CoT(\mathbf{x}), f(\mathbf{x})\}$. Similarly, a "HARD-image example" is distinct from HARD **Image**.

[6]Correctness requires all generated numbers to be in the correct order; see Appendix E.

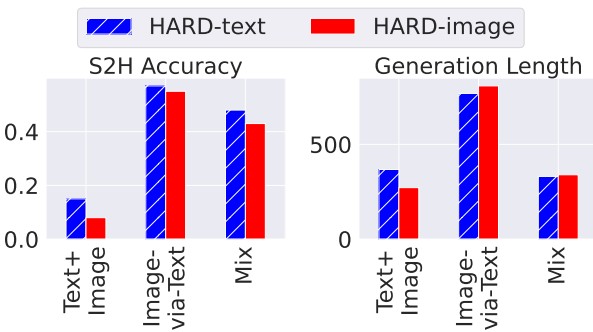

*Figure 4.* **Effect of Image-via-Text on *Consecutive Table Readout*:** S2H Generalization (left) and Generation Length (right) for HARD task. Number of training data is $16 \times 10^4$. **Text+Image** underperforms **Mix** and **Image-via-Text** supervision. **Image-via-Text** supervision improves performance slightly but at the cost of longer generation due to explicit image-to-text conversion at inference.

connection explicit. We compare the two training strategies in Figure 4 and show that **Image-via-Text** supervision shows much better performance on HARD-images.

However, **Image-via-Text** supervision has a key drawback: trained models have significantly higher inference costs, since the conversion of image to text before generating the solution leads to $3\times$ longer outputs, which limits the real-world practicality. To address this, we propose **Mix** supervision, which combines **Image-via-Text** and **Text+Image** supervision. This teaches the model to align the modalities, while also teaching it to not always rely on the image-to-text conversion.

> **Mix** can mitigate the modality imbalance by improving S2H generalization on images, while maintaining inference cost.

**Mix** supervision retains most of the S2H generalization performance of **Image-via-Text** supervision while reducing generation length by directly solving reasoning tasks from images (Figure 4). In Figure 3 (left), we show that it can almost completely match the S2H generalization performance of **Text** supervision for MEDIUM. On HARD level, even though it does not fully close the gap between text and image input (Figure 3, right), the gap can be further reduced with a short text-only warm-up training. We discuss this further in Section 6.

**Consistent results across tasks:** In Appendix G.2, we show similar results on *Pattern-Heldout Visual Analogy*.

## 4. Full Study: *Table Readout, Grid Navigation, Visual Analogy*

We now consider our three main tasks: *Table Readout, Grid Navigation*, and *Visual Analogy* (Figure 2). These tasks require the model to generalize to HARD examples by composing reasoning patterns learned from SIMPLE training examples, which has been known to be difficult for LLMs (Yu et al., 2024; Zhao et al., 2024; Wu et al., 2024b; Huang et al., 2023; Dziri et al., 2024).

These tasks are considered *non S2H-generalizing* because the model struggles to generalize to HARD instances after being trained on SIMPLE examples. In any of the three settings, training with **Text**, **Image**, and **Mix** supervision (which only include SIMPLE examples) cannot achieve more than 25% S2H generalization on either text or image.

The failure to S2H-generalize in either input modality highlights the insufficient general reasoning capacity of existing models on these tasks. We then adapt **Mix** supervision to include HARD **Text** in training and measure whether the improved performance on HARD-text can result in better S2H generalization in the image modality.

### 4.1. Improved performance on HARD-text can transfer to S2H generalization on image

**Mix+** supervision, adapted from **Mix** from Section 3, trains the model with an equal mix of HARD **Text** supervision and SIMPLE **Mix** supervision.

> **Mix+** supervision shows significantly better image S2H generalization, demonstrating an effective transfer of reasoning capability from text to image.

With only $3 \times 10^4$ data, **Mix+** quickly improves the model's accuracy on HARD-text examples to $\geq 95\%$. At the same time, **Mix+** supervision leads to a significant improvement on image S2H generalization — the model can achieve 64%, 92% and 35% S2H accuracy on HARD-images (respectively, *Table Readout, Grid Navigation*, and *Visual Analogy*) after being trained on $12 \times 10^4$ data (Figure 5). We conclude that **Mix+** supervision can effectively transfer the injected reasoning on HARD-text to S2H generalization on images.

### 4.2. Dual capability of Mix+

Motivated by the observed benefit of **Image-via-Text** supervision from Section 3, we also measure the image S2H generalization of **Image-via-Text+** supervision (an equal mix of HARD **Text** supervision and SIMPLE **Image-via-Text** supervision). On *Table Readout* and *Visual Analogy*, we observe that **Image-via-Text+** supervision outperforms **Mix+** supervision in S2H performance on HARD-image by a substantial (20-30%p) gap (Figure 6).

To close this gap, we prompt **Mix+** models with an addi-

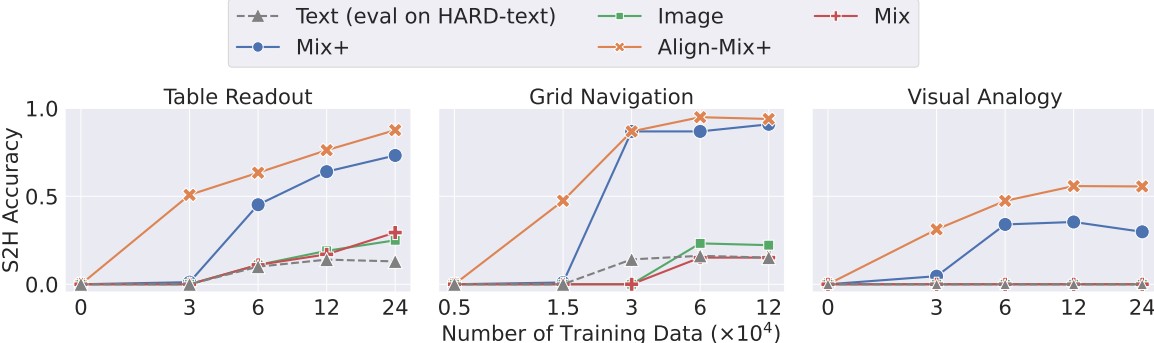

*Figure 5.* **Results on non S2H-generalizing tasks:** We report the S2H generalization on image on *Table Readout* (left), *Grid Navigation* (middle), and *Visual Analogy* (right). S2H generalization on text from **Text** supervision serves as a reference (in gray dashed line). **Text**, **Image**, and **Mix** supervisions fail to generalize, highlighting the gap between SIMPLE and HARD examples. **Mix+** improves performance, while **Align-Mix+** further enhances generalization with an initial alignment phase.

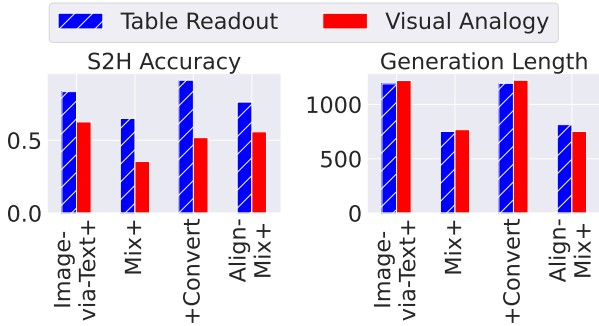

*Figure 6.* **Image-via-Text+** on *Table Readout* and *Visual Analogy*: S2H Generalization on image (left) and Generation Length (right) with $12 \times 10^4$ training examples. **Image-via-Text+** achieves good performance but with higher inference cost. **Mix+** matches the performance of **Image-via-Text+** by appending "`Convert`" to the prompt (**+Convert**) or by adding an alignment phase (**Align-Mix+**).

tional inference time token, "`Convert`", which appears at the start of the **Image-via-Text** responses (Section 2.4). We observe that the models respond with an accurate text conversion before generating the reasoning tokens.

> **Mix+** models exhibit a **dual capability** in reasoning with or without image-to-text conversion.

This is in line with the findings in Su et al. (2025) of the dual learning capability of LLMs in short and long reasoning. Note that when explicitly prompting **Mix+** models to perform image reasoning via text conversion, this still incurs a similar cost in generation length as **Image-via-Text+** (Figure 6). We discuss more in Appendix I.6.

### 4.3. Benefits of two-phase training

Given we previously observe that **Image-via-Text** supervision helps with S2H generalization, we add an initial phase that trains the model with **Text** and **Image-via-Text** supervision on SIMPLE examples. The goal is to precondition the model (via SIMPLE **Image-via-Text** supervision) to align text and image reasoning on SIMPLE examples. Intuitively, the preconditioning must be useful to generalize this knowledge on HARD examples later when trained with **Mix+** supervision. We call this two-phase approach **Align-Mix+**[7].

**Align - Mix+** significantly boosts S2H generalization on image to an accuracy of 76%, 96%, and 56% on HARD-images (respectively *Table Readout*, *Grid Navigation*, and *Visual Analogy*) after training on $12 \times 10^4$ data (Figure 5). **Align-Mix+** also maintains inference cost (Figure 6).

> **Align-Mix+** further improves image S2H generalization, while maintaining inference cost.

## 5. A Study on Loss Dynamics and Gradient Alignment for S2H generalization

Our findings show that S2H generalization can be transferred across modalities by simply mixing different types of supervision. This happens without any explicit matching of representations, which motivates us to explore training gradients to obtain insights into how each strategy contributes to S2H generalization. Here, we analyze the evaluation loss behavior on HARD **Image** and HARD **Image-via-Text**[8] examples during training. Similar gradient studies have

---

[7]For the main experiments, we use $1 \times 10^4$ training examples in the first phase. See Appendix I.3 for ablations on the number and composition of data used in the alignment phase.

[8]Identical as **Image** and **Image-via-Text** except image input $\mathbf{x}^{(i)} \in \mathcal{X}_{\text{HARD}}$.

been proposed for measuring influence (Koh & Liang, 2017) of training data points on evaluation tasks (Park et al., 2023; Xia et al., 2024; Engstrom et al., 2024).

## 5.1. A study on *Consecutive Table Readout*

In Section 3, we showed that **Mix** outperforms **Text+Image** supervision in S2H generalization on images. The key factor driving this improvement was the inclusion of **Image-via-Text** supervision. Here, we show that **Mix** supervision reduces evaluation loss on HARD **Image** examples (therefore improving evaluation accuracy) through a better gradient signal. To do so, we measure the alignment between gradients on SIMPLE and HARD **Image** examples.

Let $l_{(I;S)}(\mathbf{x})$ denote the *loss on solution given image*, i.e.,

$$l_{(I;S)}(\mathbf{x}) := l(f_\theta(\{\mathbf{x}^{(i)}, \mathbf{y}\}), \mathbf{y})) \tag{1}$$

where $\mathbf{y}$ contains both $CoT(\mathbf{x})$ and the answer $f(\mathbf{x})$. We also denote the *loss on solution given* HARD *image* as

$$l_{(I;S)}^{(H)} := \mathbb{E}_{\mathbf{x} \in \mathcal{X}_{\text{HARD}}} l_{(I;S)}(\mathbf{x}) \tag{2}$$

If $\mathbf{g}_{\text{SIMPLE}}$ and $\mathbf{g}_{\text{HARD}}$ denote average gradients on $\mathcal{X}_{\text{SIMPLE}}$ and $\mathcal{X}_{\text{HARD}}$ (i.e. $\mathbb{E}_{\mathbf{x} \in \mathcal{X}_{\text{SIMPLE}}} \nabla l_{(I;S)}(\mathbf{x})$ and $\mathbb{E}_{\mathbf{x} \in \mathcal{X}_{\text{HARD}}} \nabla l_{(I;S)}(\mathbf{x})$ respectively)[9], then we define the **gradient alignment score** as [10]:

$$\langle \mathbf{g}_{\text{SIMPLE}}, \mathbf{g}_{\text{HARD}} \rangle / \langle \mathbf{g}_{\text{HARD}}, \mathbf{g}_{\text{HARD}} \rangle \tag{3}$$

Intuitively, the gradient alignment score measures how much the evaluation loss (on HARD **Image**) can be reduced by taking gradient updates from the training data (SIMPLE **Image**), relative to training on evaluation data directly (see Theorem H.1 for a formal statement). In Figure 7, we plot this score against the gradient norms on the training data. A stronger gradient alignment at larger values of gradient norm is preferred because the evaluation loss can be reduced more when the training gradients are larger.

> **Mix** achieves a high gradient alignment score, especially when gradient norms are large. This improved alignment leads to a significant initial drop in the evaluation loss (*loss on solution given* HARD-*image*), which then continues to improve throughout training.

## 5.2. A study on *Table Readout*

In Section 4, we showed that **Mix+** improves S2H generalization over **Mix**, while **Align-Mix+** can further improve over **Mix+** with an additional alignment training. Here, we study how each included component helps S2H generalization across the training strategies.

---

[9]Following Park et al. (2023), we apply a random projection on gradients to 4096 dimension for an efficient storage.

[10]In Appendix H.3, we give two alternative measures of gradient alignment. Our takeaways remain the same.

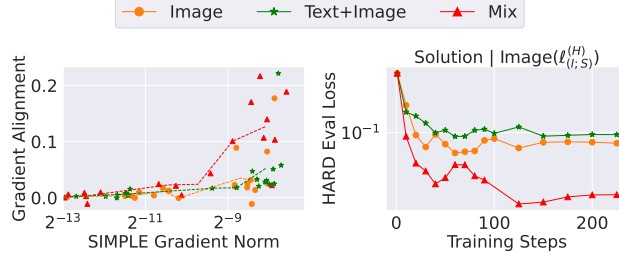

*Figure 7.* **Analysis of gradients on *Consecutive Table Readout*:** (Left) Average Gradient Norm on SIMPLE **Image** examples ($\mathbb{E}_{\mathbf{x} \in \mathcal{X}_{\text{SIMPLE}}} \|\nabla l_{(I;S)}(\mathbf{x})\|_2$) vs. Gradient Alignment Score (Equation (3)) for different training checkpoints; (Right) Average *Loss on solution given* HARD *image* ($l_{(I;S)}^{(H)}$) during training. Larger gradients for **Mix** have higher alignment score compared to **Text+Image** and **Image**, showing the importance of **Image-via-Text** supervision for generalization.

**Insights from the evaluation loss dynamics:** We use the following additional notations to report the average loss on specific tokens on a HARD **Image-via-Text** example and understand which components help the model learn to reason on HARD-images via text conversion, and how it translates to a direct solution on HARD-image examples.

- HARD *image-to-text conversion*: Average loss on converted text tokens given the image and the conversion prompt [11]):

$$l_{(I\#;T)}^{(H)} := \mathbb{E}_{\mathbf{x} \in \mathcal{X}_{\text{HARD}}} l(f_\theta(\{\mathbf{x}^{(i)}, P_{convert}, \mathbf{x}^{(t)}\}), \mathbf{x}^{(t)}). \tag{4}$$

- *Solution given* HARD *image and text*: Average loss on solution tokens given the image, the conversion prompt, and the converted text:

$$l_{(I,\#T;S)}^{(H)} := \mathbb{E}_{\mathbf{x} \in \mathcal{X}_{\text{HARD}}} l(f_\theta(\{\mathbf{x}^{(i)}, P_{convert}, \mathbf{x}^{(t)}, \mathbf{y}\}), \mathbf{y}), \tag{5}$$

where $\mathbf{y}$ contains both $CoT(\mathbf{x})$ and the answer $f(\mathbf{x})$.

In Figure 8, we report the above losses for **Mix**, **Mix+**, and **Align-Mix+**. Since the model does not see HARD-image examples during training, these losses (along with $l_{(I;S)}^{(H)}$) evaluate the S2H generalization on image. We observe:

1. HARD *image-to-text conversion loss* (Equation (4)) of **Mix** matches **Mix+**, showing that training on SIMPLE **Image-via-Text** examples suffices to generalize the conversion subtask to HARD-images.

2. There is a significant gap in the *loss on solution given* HARD *image and text* (Equation (5)) between **Mix** and **Mix+**. This implies that including HARD **Text** is necessary to fully generalize reasoning to HARD-images.

---

[11]# indicates the additional conversion prompt $P_{convert}$

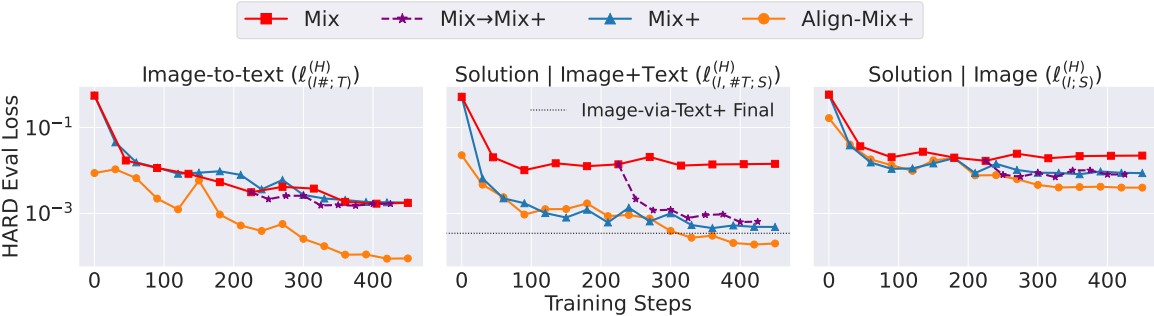

*Figure 8.* **Analysis of evaluation losses on HARD examples on *Table Readout*:** (Left) HARD *image-to-text conversion loss* ($l_{(I\#;T)}^{(H)}$ (Eq.4)); (Middle) *loss on solution given* HARD *image and text* ($l_{(I,\#T;S)}^{(H)}$ (Eq.5)); (Right) *loss on solution given* HARD *image* ($l_{(I;S)}^{(H)}$ (Eq.2)). **Mix** matches **Mix+** in $l_{(I\#;T)}^{(H)}$, showing that training on SIMPLE **Image-via-Text** examples is sufficient for HARD image-to-text conversion. **Mix** performs worse in $l_{(I,\#T;S)}^{(H)}$, showing the need for HARD **Text** examples for generalization. Taking an intermediate checkpoint of **Mix** and completing the training with **Mix+** (**Mix → Mix+**) leads to evaluation loss values comparable to **Mix+**, suggesting that HARD **Text** examples can be introduced later. **Align-Mix+** starts with smaller $l_{(I\#;T)}^{(H)}$ and $l_{(I,\#T;S)}^{(H)}$ losses, which helps the model achieve lower $l_{(I,\#T;S)}^{(H)}$ loss than even **Image-via-Text+**, that reflects in lower $l_{(I;S)}^{(H)}$ loss.

As an ablation, we took an intermediate **Mix** checkpoint and completed the training with **Mix+** supervision[12]. This transition resulted in negligible changes to HARD *image-to-text conversion loss* (Equation (4)), while *loss on solution given* HARD *image and text* (Equation (5)) and *loss on solution given* HARD-*image* (Equation (2)) decreased significantly, approaching the values for **Mix+**.

3. Losses on HARD **Image-via-Text** examples start significantly lower for **Align-Mix+** after the alignment phase. This shows that training on SIMPLE **Image-via-Text** examples can return a favorable starting point, even if they aren't sufficient for generalization. It then achieves a better *loss on solution given* HARD *image and text* (Equation (5)) in the end, which also translates to an improved *loss on solution given* HARD *image* (Equation (2)).

**Insights from the gradient alignment score:** We can further quantify the differences in training strategies with the gradient alignment score (Equation (3)) between SIMPLE and HARD **Image** examples (Figure 9). Intuitively, a higher gradient alignment at each step should accumulate to a better generalization on HARD-images. We observe:

1. **Mix** exhibits lower gradient alignment compared to **Mix+**. Training solely on SIMPLE examples fails to provide gradients aligned to HARD **Image**. Including HARD **Text** examples significantly improves gradient alignment.

2. On the other hand, **Align-Mix+** has higher gradient alignment than **Mix+** in earlier training steps, when training gradient norms are large. We give a detailed analysis in Figure 17 in Appendix H.

---
[12]We preserved the optimizer states and learning rate schedule.

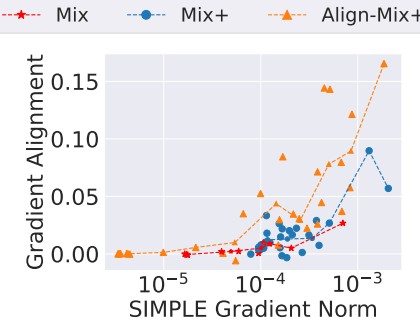

*Figure 9.* **Analysis of gradients on *Table Readout*:** Average Gradient Norm on SIMPLE **Image** examples ($\mathbb{E}_{\mathbf{x} \in \mathcal{X}_{\text{SIMPLE}}} \|\nabla l_{(I;S)}(\mathbf{x})\|_2$) vs. Gradient Alignment Score for different training checkpoints. Larger gradients for **Align-Mix+** have higher gradient alignment scores. **Mix+** has better gradient alignment scores than **Mix**.

# 6. Further Ablations

We perform several ablation studies to identify critical training components that underlie our findings. We push all the details and discussions to the appendix.

**Task interactions in multi-task training:** We compare **Mix**, **Mix+** and **Align-Mix+** with an equal mix of 3 tasks in Section 4. We observe that multi-task training significantly boosts performance on *Table Readout* and *Grid Navigation* but hurts on *Visual Analogy*, which shows the effect of task interactions in our strategies. See Appendix I.8.

**Transferring reasoning from image to text:** We also experiment with including HARD **Image** supervision in training and evaluating on HARD-text input, which gives much stronger results (Table 8 in Appendix G.4),

**Text warm-up pretraining:** We add a text warm-up pre-training (TW) phase before the training of VLM to simulate the effect of a stronger LLM backbone. This pretraining phase completely solves the modality imbalance or further boosts performance of Align-Mix+. See Appendix G.5.

**Importance of chain-of-thought:** Completely removing or progressively internalizing CoT (Deng et al., 2024) fails to achieve image S2H generalization, suggesting that CoT is crucial in our strategies. See Appendix I.7.

## 7. Discussion, Limitations and Future work

We explore the modality imbalance in VLMs by measuring S2H generalization. We show that on tasks where VLMs can reliably show generalization on text input after fine-tuning, Mix supervision can induce a similar level of generalization on image input. We then propose 3 algorithmic tasks, where models trained on SIMPLE examples fail to generalize to HARD examples in either modality. Mixing HARD Text examples in training can help the model generalize on HARD-image input, revealing S2H generalization transfer capabilities of these models.

**Related Works:** Current VLM benchmarks are often solvable without the visual input. To remove such bias, we designed controllable tasks and provided a framework (S2H generalization) to quantify and mitigate modality imbalance. While S2H generalization has been extensively studied for LLMs, similar investigations remain scarce for VLMs.

Prior strategies to address modality imbalance and cross-modal transfer often rely on matching representations or optimization techniques. However, through gradient alignment studies, we demonstrate that auto-regressive training effectively aligns reasoning across modalities.

For a more detailed discussion, see Appendix B.

**Utility to real-world benchmarks:** Extending our findings to real-world scenarios is also left for future work. It will require real-world scenarios with precise gradation of SIMPLE and HARD examples with respect to underlying abstract concepts. Our work suggests that the brittleness of VLMs could be mitigated by training them to create very detailed descriptions of the scene (and this capability could be internalized for faster inference).

We note that training even on our synthetically created datasets seems useful for improving the performance of VLMs in real-world settings. Specifically, including our synthetic datasets during pretraining of VLMs yielded significant improvements across different benchmarks (Table 9 in Appendix G.6). For example, including SIMPLE and HARD Image supervision examples from all synthetic datasets can improve performance on MMMU (Yue et al., 2024) by at least 3%p. Similarly, on a chart dataset (Wang et al., 2024c), including our synthetic datasets can improve

performance by 5.1%p on descriptive questions. Therefore, our synthetic datasets involve useful skills that can also help improve VLMs on real-world benchmarks.

**Limitations and possible future directions:** We believe Mix or Mix+ may not be the optimal approach to improve image generalization on tasks where the model exhibits S2H generalization in the text modality. Curriculum-based strategies (Xie et al., 2024; Mindermann et al., 2022) that dynamically adjust the data mixture could yield better results. However, our goal is to emphasize the HARD generalization gap between text and image inputs, which can be bridged by transferring learning from the dominant modality (text) to the weaker one (image). Therefore, we focus on the effectiveness of our training strategies in transferring knowledge learned on text input to image input.

In the interest of crispness, we restricted the scope of our study with a small set of prompts and a limited (and synthetic) image distribution. But doing so allowed a clearer and quantitative look at modality imbalance and how it can be bridged.

Our results highlight that chain-of-thought (CoT) reasoning can play an important role. However, even minor modifications to CoT significantly affect the transferred S2H generalization results on image inputs, and mitigating this brittleness through robust training strategies beyond Mix+ is crucial. Future work could focus on mechanistic insights into our trained models to design more generalizable strategies targeting specific model components.

## Acknowledgements

SP, AP, CY, and SA acknowledge fundings from NSF, PLI, DARPA, ONR, and OpenAI. CY is additionally supported by the Francis Robbins Upton Fellowship in engineering. We thank Xingyu Zhu, Bingbin Liu, Nikunj Saunshi, Sadhika Malladi, Samy Jelassi, Misha Khodak, Zirui Wang, Mengzhou Xia, Yihe Dong, Haoyu Zhao, Danqi Chen, Tri Dao, and Benjamin Eysenbach for discussions, suggestions, and proof-reading at various stages of the paper.

## Impact Statement

This paper presents work whose goal is to advance the field of Machine Learning. It primarily is a basic scientific exploration of the capabilities of Vision Language Models. It may lead to the development of better VLMs, but we do not anticipate any negative societal impact.

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

# Appendix

## Table of Contents

## A. Appendix Structure

The appendix provides omitted experimental details, additional empirical explorations, and theoretical statements, which we outline below.

**Related works:**    In Appendix B, we provide an overview of relevant lines of research in VLM benchmarks and evaluations, modality imbalance, cross-modal transfer of generalization, and simple-to-hard generalization. We highlight the contributions that differentiate our work from the similar ones.

**Experimental details:**    We provide all details of our synthetic data generation in Appendix C. We present our data generation algorithm for creating training data in Appendix C.1, details on *Consecutive Table Readout*, *Table Readout*, *Visual Analogy*, and *Grid Navigation* in Appendices C.2 to C.5 respectively. We show examples from our training data for each synthetic setting in Figures 32 to 37. We present details on training and evaluation in Appendices D and E respectively.

**Consistent results on another model family and size:**    We replicate some experiments from the main paper with *Qwen2.5-VL-3B-Instruct* and *Qwen2.5-VL-7B-Instruct*. We report the results in Appendix F.

**Continued discussion from main paper:**    We continue the discussion in the main paper in Appendix G. We present results on *Consecutive Table Readout* after normalizing the number of unique samples used across training strategies (Appendix G.1), present results on *Pattern-Heldout Visual Analogy* — a S2H-generalizing version of *Visual Analogy* (Appendix G.2), compare training strategies on non S2H-generalizing tasks by normalizing the total number of training data used (Appendix G.3), discuss further on transferring reasoning from image to text modality (Appendix G.4), discuss further on text warm-up pretraining (Appendix G.5), and report the utility of our created synthetic datasets for real-world benchmarks (Appendix G.6).

**Continued discussion on gradients:**    We continue our discussion on gradient alignment in Appendix H. We first show that the gradient alignment score connects to the expected drop in evaluation loss with SGD on training gradients (Theorem H.1). We then propose results on additional measures — gradient cosine similarity and Adam update alignment score (Appendix H.3) — that better capture the Adam gradient updates used for optimization.

**Ablation studies:**    We conduct extensive ablation studies to measure the effect of each experimental design decision in our training strategies on non S2H-generalizing tasks and report the results in Appendix I. We report the performance on other multimodal models on our synthetic data (Appendix I.1). We study design choices in **Mix+** (Appendix I.2), design choices in **Align-Mix+** (Appendix I.3), design choices in text warm-up pretraining (Appendix I.4), the effect of the choice of a text representation (Appendix I.5), the effect of text conversion (Appendix I.6), the role of chain-of-thought (Appendix I.7), the effect of multi-task training (Appendix I.8), and the effect of repeated training examples (Appendix I.9).

**Interpretability experiments:**    We further conduct interpretability experiments on our trained models. We use gradient attribution to track the focus of the model on different image pixels during chain-of-thought generation (Appendix J). We also report failure modes of models trained on our synthetic data when evaluated on HARD examples (Appendix K).

# B. Related Works

**Benchmarks and evaluations for VLMs** VLMs are evaluated on benchmarks such as visual question answering (VQA) (Antol et al., 2015), image captioning (Chen et al., 2015), zero-shot image classification (Deng et al., 2009), and compositional reasoning (Thrush et al., 2022; Yuksekgonul et al., 2023; Hsieh et al., 2024). However, these benchmarks often suffer from language bias, allowing solutions to use shortcuts with minimal visual information (Agrawal et al., 2016; Goyal et al., 2017; Zhang et al., 2024c). Although recent work (Rahmanzadehgervi et al., 2024; Wang et al., 2024b; Kil et al., 2024) proposed new benchmarks that aim to evaluate the spatial understanding and reasoning of VLM, most evaluation tasks are in the form of VQA questions that only require "single-hop" reasoning or relatively fewer reasoning steps. To create a controlled setting with well-defined SIMPLE and HARD tasks, we focus on algorithmic visual reasoning tasks. These tasks allow us to precisely control the number of steps in the step-by-step reasoning process and the level of dynamic interaction between textual and visual inputs. Closely related works have explored graph-based algorithmic reasoning in LLMs (Taylor et al., 2024; McLeish et al., 2024; Zhang et al., 2024f; Wang et al., 2024a; Sanford et al., 2024) but such studies remain limited for VLMs.

**Modality imbalance** Studies have shown that models exhibit different learning capabilities and learning speed on multimodal inputs (Wang et al., 2020; Nguyen et al., 2024). The imbalanced contribution of individual modality to the final prediction can result in overreliance on a few dominant, optimized modalities, while underutilizing signals of the weak ones. Peng et al. (2022) and Lin et al. (2024) attempt to rebalance the convergence speed of all modalities by modulating the learning rate or gradients. Fan et al. (2023) propose a representative embedding to guide the slow-learning modality and regularize the fast-learning one. Zhang et al. (2024e) propose an alternating unimodal training to minimize interference between modalities. Despite their success in traditional multimodal joint training, it remains challenging to repeat the same for adapter-based VLMs due to significant differences in architecture and training pipeline. Our work aims to address this issue specifically for VLMs from the perspective of transferring the strong learning behaviors from the dominant modality (text) to the weak one (image).

**Generalization transfer between input modes** Given the high cost of training VLMs from scratch, recent research on adapter-based VLMs has been driven primarily by the idea of leveraging pretrained LLM backbones. The success of this approach is built on the idea of cross-modal generalization, which enables the model to harness information from the auxiliary modality (e.g. text) to improve unimodal task on the primary modality (e.g. image classification). This knowledge transference has been exploited for both small-scale multimodal models (Socher et al., 2013; Liang et al., 2021; Tan & Bansal, 2020) and more recent VLMs (Monajatipoor et al., 2023; Carbune et al., 2024; Zhang et al., 2024a). However, existing works often require explicit alignment of the modality, such as learning unified representation using contrastive learning (Xia et al., 2023), for models to transfer knowledge across modalities. The cost of curating a large, perfectly aligned multimodal dataset to learn the modality alignment becomes expensive as the model size increases. In our work, we find that transfer of generalization across input modes naturally emerges from auto-regressive training.

**S2H generalization** Recent studies have explored simple-to-hard generalization in LLMs, with a focus on length generalization in transformers. These works evaluate models on tasks requiring longer computations than those seen during training, using synthetic datasets like parity, Dyck-1 languages, decimal addition, structural recursion, and finite state automata (Anil et al., 2022; Lee et al., 2024; Jelassi et al., 2023; Li & McClelland, 2023; Kazemnejad et al., 2024; Liu et al., 2023a; Abbe et al., 2024; Bhattamishra et al., 2020; Zhou et al., 2024b; Fan et al., 2025). Zhou et al. (2024a) connect length generalization to the RASP programming language (Weiss et al., 2021), offering a unified perspective. Sun et al. (2024) recently propose easy-to-hard generalization to measure generalizable verification for math and code datasets. OOD generalization beyond human supervision remains an important open question for the advancement of current AI models (Burns et al., 2023).

*Table 1.* **Summary of the SIMPLE and HARD task setup** for *Table Readout*, *Grid Navigation*, and *Visual Analogy*

| Setting | Attribute | SIMPLE | HARD |
|---|---|---|---|
| *Table Readout* | Mean Length | 12 | 35 |
| | # Turns | $1-4$ | $>4$ |
| | Pattern | Spiral / Sinusoidal | Composition of Spiral / Sinusoidal |
| *Grid Navigation* | # DFS steps | $[10, 25]$ | $[26, 60]$ |
| | # Objects | $\{1, 2\}$ | $\{2, 3, 4, 5\}$ |
| | # Obstacle type | $\{1\}$ | $\{3, 4, 5\}$ |
| *Visual Analogy* | Example Patterns | Same | Different |
| | Query Pattern | Seen | Held-out |

# C. Details on Synthetic Tasks

## C.1. Formal description of data generation

In Algorithm 1, we provide the pseudo-code for generating the training data mixture for the main experiments. Below we provide more details in the setup.

### C.1.1. WHEN TRAINING ONLY ON SIMPLE EXAMPLES

For *Consecutive Table Readout* and for any type of supervision among **Text**, **Image**, **Text+Image**, **Image-via-Text**, and **Mix**:

- For each unique data $\mathbf{x} \in \mathcal{X}$ and for each type of supervision — **Text**, **Image**, and **Image-via-Text**, we choose whether to include it in the training data, depending on whether these types of supervision are used for training (Section 2.4). We denote the number of unique data $\mathbf{x}$ used from $\mathcal{X}_{\text{SIMPLE}}$ as $N^u_{\text{SIMPLE}}$.

- We compare all training strategies with the total number of training data used, given by:

$$N_{\text{SIMPLE}} = \text{Number of epochs} \times N^u_{\text{SIMPLE}} \times \text{Number of types of supervision per input}$$

For a fair comparison, we keep the number of unique data $N^u_{\text{SIMPLE}}$ fixed across **Text+Image**, **Image-via-Text**, and **Mix**. Then to match $N_{\text{SIMPLE}}$, we set the number of epochs to $1.5$ for **Text+Image** ($50\%$ samples are repeated $2\times$), $3$ for **Image-via-Text**, and $1$ for **Mix**.

**Note on Text and Image for *Consecutive Table Readout*:** Since our result depends heavily on the success of **Text** and the failure of **Image** in *Consecutive Table Readout*, we carefully tune the number of training epochs to achieve optimal performance. We conduct ablations where instead of setting $N^u_{\text{SIMPLE}} = N_{\text{SIMPLE}}$, we also try setting $N^u_{\text{SIMPLE}}$ equal to $\frac{N_{\text{SIMPLE}}}{2}$ or $\frac{N_{\text{SIMPLE}}}{3}$ (respectively, the number of epochs is set at $2, 3$). The results presented in Figure 3 corresponds to $N^u_{\text{SIMPLE}} = \frac{N_{\text{SIMPLE}}}{2}$ for **Text** and $N^u_{\text{SIMPLE}} = \frac{N_{\text{SIMPLE}}}{3}$ for **Image**. We discuss further in Appendix G.1.

### C.1.2. WHEN ALSO TRAINING ON HARD EXAMPLES

For **Image-via-Text+** or **Mix+** on non S2H-generalizing tasks:

- We set $N_{\text{HARD}}$, the number of data from the HARD task, equal to $N_{\text{SIMPLE}}$, the number of data from the SIMPLE task.

- We generate a mixture of $N_{\text{SIMPLE}}$ examples under **Image-via-Text** or **Mix**. We include $N_{\text{HARD}}$ instances of HARD **Text**.

### C.1.3. REASONING ALIGNMENT (**Align**-) OR TEXT WARM-UP PRETRAINING ((**TW**))

When generating data for the reasoning alignment phase (**Align**-):

- We set $N = 10^4$ and include an equal number of SIMPLE **Text** and SIMPLE **Image-via-Text** examples.

When generating data for the text warm-up pretraining phase ((**TW**)):

- We set $N = 10^4$ and include an equal number of SIMPLE **Text** and HARD **Text** examples.

After training on (**TW**) and/or **Align**-, we continue with the main phase of supervision (e.g., **Mix+** for **Align-Mix+**).

---

**Algorithm 1** Data generation pipeline for main experiments

---

**Require:** Task $f : \mathcal{X} \to \mathcal{Z}$, Dataset $\mathcal{X} = \mathcal{X}_{\text{SIMPLE}} \cup \mathcal{X}_{\text{HARD}}$, Number of data to generate $N$, Type of supervision $s$.

  **if** $s \in \{$ Text, Image $\}$ **then**

    Initialize the number of data per difficulty $N_{\text{SIMPLE}} = N$, $N_{\text{HARD}} = 0$ and the number of unique examples $N^u_{\text{SIMPLE}} = N_{\text{SIMPLE}}$

  **else if** $s \in \{$ Text+Image, Image-via-Text, Mix $\}$ **then**

    Initialize the number of data per difficulty $N_{\text{SIMPLE}} = N$, $N_{\text{HARD}} = 0$ and the number of unique examples $N^u_{\text{SIMPLE}} = \frac{N_{\text{SIMPLE}}}{3}$

  **else if** $s \in \{$ Image-via-Text+, Mix+ $\}$ **then**

    Initialize the number of data per difficulty $N_{\text{SIMPLE}} = \frac{N}{2}$, $N_{\text{HARD}} = \frac{N}{2}$ and the number of unique examples $N^u_{\text{SIMPLE}} = \frac{N_{\text{SIMPLE}}}{3}$

  **else if** $s \in \{$ Align- $\}$ **then**

    Initialize the number of data per difficulty $N_{\text{SIMPLE}} = N$, $N_{\text{HARD}} = 0$ and the number of unique examples $N^u_{\text{SIMPLE}} = \frac{N_{\text{SIMPLE}}}{2}$

  **else if** $s \in \{$ (TW) $\}$ **then**

    Initialize the number of data per difficulty $N_{\text{SIMPLE}} = \frac{N}{2}$, $N_{\text{HARD}} = \frac{N}{2}$ and the number of unique examples $N^u_{\text{SIMPLE}} = N_{\text{SIMPLE}}$

  **end if**

  Initialize $\mathcal{S} = \Phi$.

  **for** $t = 1 \to N^u_{\text{SIMPLE}}$ **do**

    Sample $\mathbf{x} \sim \mathcal{X}_{\text{SIMPLE}}$.

    If $s \in \{$ Text, Text+Image, Image-via-Text, Mix, Image-via-Text+, Mix+, Align-, (TW) $\}$, then $\mathcal{S} \leftarrow \mathcal{S} \cup (\{\mathbf{x}^{(t)}, CoT(\mathbf{x}), f(\mathbf{x})\})$.

    If $s \in \{$ Image, Text+Image, Image-via-Text, Mix, Image-via-Text+, Mix+ $\}$, then $\mathcal{S} \leftarrow \mathcal{S} \cup (\{\mathbf{x}^{(i)}, CoT(\mathbf{x}), f(\mathbf{x})\})$.

    If $s \in \{$ Image-via-Text, Mix, Image-via-Text+, Mix+, Align- $\}$, then $\mathcal{S} \leftarrow \mathcal{S} \cup (\{\mathbf{x}^{(i)}, P_{convert}, \mathbf{x}^{(t)}, CoT(\mathbf{x}), f(\mathbf{x})\})$.

  **end for**

  Determine number of epochs to repeat $e = \frac{N_{\text{SIMPLE}}}{|\mathcal{S}|}$

  Randomly shuffle $\mathcal{S}$ and repeat it $e$ times (i.e., take the first $e \cdot |\mathcal{S}|$ elements from repeated copies of $\mathcal{S}$)

  **for** $t = 1 \to N_{\text{HARD}}$ **do**

    Sample $\mathbf{x} \sim \mathcal{X}_{\text{HARD}}$.

    $\mathcal{S} \leftarrow \mathcal{S} \cup (\{\mathbf{x}^{(t)}, CoT(\mathbf{x}), f(\mathbf{x})\})$.

  **end for**

  Randomly shuffle $\mathcal{S}$ and return $\mathcal{S}$.

---

*Figure 10.* **Pseudo-code for generating data mixture:** For ablation studies, the algorithm might be slightly modified.

### C.2. *Consecutive Table Readout*

Given a table with $n_r$ rows and $n_c$ columns, a start cell $(r_s, c_s)$ and an end cell $(r_e, c_e)$, the model is tasked to read all numbers between the start cell and end cell following the given rules.

- If $r_s < r_e$, move left-to-right within each row:

$$(r_s, c_s), (r_s, c_s + 1), \cdots, (r_s, n_c), (r_s + 1, 1), (r_s + 1, 2), \cdots, (r_e, 1), (r_e, 2), \cdots, (r_e, c_e)$$

- If $r_s > r_e$, move right-to-left within each row:

$$(r_s, c_s), (r_s, c_s - 1), \cdots, (r_s, 1), (r_s - 1, n_c), (r_s - 1, n_c - 1), \cdots, (r_e, n_c), (r_e, n_c - 1), \cdots, (r_e, c_e)$$

- If $r_s = r_e$, move from $(r_s, c_s)$ to $(r_e, c_e)$.

See example images in Figure 1.

### C.3. *Table Readout*

Given a table with $n_r$ rows and $n_c$ columns (where $n_r, n_c \in [8, 12]$), a start cell $(r_s, c_s)$, an end cell $(r_e, c_e)$, and a path of cells $P$ connecting the two cells (without any loops), the task is to read the numbers on the path starting from the start cell and ending at the end cell. Each path is continuous and is a concatenation of linear segments, where consecutive segments are separated by 90 degree turns. On the SIMPLE task, each path contains $1 - 4$ linear segments, following a spiral or sinusoidal pattern, and has an average length of 12. On the HARD task, each path contains $> 4$ linear segments, following a compositional spiral or sinusoidal pattern, and has an average length of 35. See example images in Figures 32 and 33 and an example pseudo-code to create the spiral or sinusoidal patterns in Algorithms 2 and 3.

**Algorithm 2** Spiral Path Generation that changes directions as right→down→left→up→right→ $\cdots$

**Require:** Table with $n_r$ rows and $n_c$ columns, start cell, $k$ linear segments
  • Initial $n_{seg} = 0$
  • Initialize current-cell coordinates as start cell coordinates
  • Initialize current-direction to "right"
  • Initialize Path= $\Phi$.
  • Direction-Change = {"right" : "down", "down" : "left", "left" : "up", "up" : "right"}
  • Coordinate-Update = {"right" : $(0, 1)$, "down" : $(1, 0)$, "left" : $(0, -1)$, "up" : $(-1, 0)$}
  **while** $n_{seg} \neq k$ **do**
    • Add current cell to Path.
    • Compute temporary-cell by adding coordinate update vector for current-direction from Coordinate-Update to current-cell.
    • If temporary-cell is out of bounds, update current-direction using Direction-Change and increment $n_{seg}$.
    • Update current-cell by adding coordinate update vector for current-direction from Coordinate-Update to current-cell.
  **end while**
  Return Path

**Algorithm 3** Sinusoidal Path Generation that changes directions as right→down→left→up→right→ $\cdots$, where down and up movements contain only 2 cells

**Require:** Table with $n_r$ rows and $n_c$ columns, start cell, $k$ linear segments
  • Initial $n_{seg} = 0$
  • Initialize current-cell coordinates as start cell coordinates
  • Initialize current-direction to "right"
  • Initialize Path= $\Phi$.
  • Direction-Change = {"right" : "left", "left" : "right"}
  • Coordinate-Update = {"right" : $(0, 1)$, "left" : $(0, -1)$}
  **while** $n_{seg} \neq k$ **do**
    • Add current cell to Path.
    • Compute temporary-cell by adding coordinate update vector for current-direction from Coordinate-Update to current-cell.
    **if** temporary-cell is out of bounds **then**
      • If $n_{seg} = k - 1$, break
      Loop twice
        • Increment column coordinate by 1 in current-cell
        • Add current-cell to Path.
      • Update current-direction using Direction-Change
      • Increment $n_{seg}$ by 2.
    **end if**
    • Update current-cell by adding coordinate update vector for current-direction from Coordinate-Update to current-cell.
  **end while**
  • Return Path

*Figure 11.* **Pseudo-code for generating spiral and sinusoidal paths on** *Table Readout***:** For simplicity, we present a single variant of each pattern. By permuting the Direction-Change map, the presented variants can be modified to include other direction patterns.

## C.4. *Grid Navigation*

Given a grid with $n_r$ rows and $n_c$ columns (where $n_r, n_c \in [8, 12]$), a start cell $(r_s, c_s)$, an end cell $(r_e, c_e)$, and a set of objects and obstacles placed at various positions within the grid, the task is to find a path from the start cell to the end cell that collects all specified objects while avoiding all obstacles.

For each generated grid, we randomly select several objects from a set of 30 possibilities: heart, crown, flag, star, flower, umbrella, plane, phone, spark, diamond, queen, hammer, club, gear, arrow, sun, bishop, note, coffee, anchor, cloud, pawn, castle, horse, infinity, moon, null, approx, integral, product, and sum. Each chosen object is represented as an Unicode character, as shown in Figure 12. Obstacles are chosen from the following five symbols: dot, cross, square, triangle, and plus. The names and representations of all these symbols—both objects and obstacles—have been verified using GPT-4o.

The SIMPLE task requires the model to collect $k \in [1, 2]$ objects spread across the grid, while avoiding a single kind of obstacle. The HARD task requires the model to collect $k \in [2, 5]$ objects spread across the grid, while avoiding a composition of $o \in [3, 5]$ obstacles. The SIMPLE task requires $t \in [10, 25]$ DFS steps, while the HARD task requires $t \in [25, 60]$ DFS steps.

See example images in Figures 34 and 35.

## C.5. *Visual Analogy*

We create a multimodal visual analogy dataset based on the Procedurally Generated Matrices (PGM) data proposed in Barrett et al. (2018) and Hill et al. (2019). Each instance consists of 2 examples of three images, a query of two images, and four answer options. Each instance has a latent logical relation $r \in \{\texttt{XOR}, \texttt{OR}, \texttt{AND}, \texttt{Progression}\}$ that will be applied to both the examples and the query. There are also three latent domains $d_1, d_2, d_{\text{query}}$ (for each example and the query, respectively), chosen from {line_type, line_color, shape type, shape_color, shape_size, shape_quantity, shape_position}. For each example $i$, the value of the domain $d_i$ in the third image follows from applying the relation $r$ to the values in the first two images. The task is to choose one of the four options so that there exists a domain $d_{\text{query}}$ where applying the relation $r$ along $d_{\text{query}}$ in the first two images of the query leads to the chosen option.

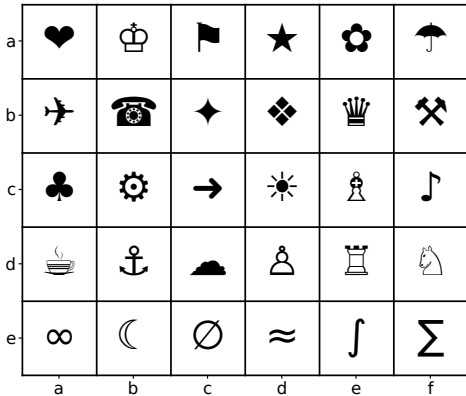

*Figure 12.* **Details on *Grid Navigation*:** Unicode characters used for specifying each object.

*Table 2.* **List of all possible attribute values for each domain in *Visual Analogy*:**, We reproduce Hill et al. (2019) with slight modifications. The diverse combination of the attribute values results in high complexity of this task, testing various both OOD and compositional generalizability of the model to a great extent.

| | |
|---|---|
| line type | {falling diagonal line, rising diagonal line, horizontal line, vertical line, diamond lines, circular line, V-shape facing up, V-shape facing left V-shape facing down, V-shape facing right} |
| line color | {0 (black), 90 (dark grey), 135 (grey), 189 (light grey)} |
| shape type | {circle, rectangle, triangle, pentagon, hexagon} |
| shape color | {0 (black), 90 (dark grey), 135 (grey), 189 (light grey), 255 (white)} |
| shape size | {20, 27, 34, 41} |
| shape quantity | {0, 1, 2, 3, 4, 5, 6, 7, 8, 9} |
| shape position | {(0, 0), (0, 1), (0, 2), (1, 0), (1, 1), (1, 2), (2, 0), (2, 1), (2, 2)} |

Note that following Hill et al. (2019), we exclude all spurious correlations of the examples and query such that they follow exactly one pattern $(d, r)$. Furthermore, we create three nontrivial confounding options such that each of them, when combined with the query images, is consistent with exactly one pattern $(d_{\text{option}_i}, r_{\text{option}_i})$ where $r_{\text{option}_i} \neq r_{\text{query}}$.

We also reserve a held-out set of combinations $\mathcal{S} = \{(d, r)\}$ that does not appear in the training images. On the SIMPLE task, $d_1 = d_2 = d_{\text{query}}$ and the query pattern $(d_{\text{query}}, r_{\text{query}})$ is never chosen from the held-out set. On the HARD task, $d_1, d_2, d_{\text{query}}$ are distinct and both $(d_i, r_i)$ and $(d_{\text{query}}, r_{\text{query}})$ are always chosen from the held-out set $\mathcal{S}$.

See example images in Figures 36 and 37 and the complete list of all possible attribute values in Table 2.

### C.6. Issues during synthetic data creation

Here, we outline the primary issues that we faced while creating the synthetic datasets, which might be of value to the general community.

#### C.6.1. *Consecutive Table Readout*, *Table Readout*

The primary issue that we faced during creation of these datasets were as follows:

- **Resolution issues:** For images, we found that representing numbers as their English names (e.g. 9 represented as NINE) improved the OCR performance substantially. When represented as numerics, the model often confused between pairs (7, 9), and (0, 8). These issues were largely mitigated by replacing numerics with English names.

- **Color:** The model's S2H generalization can vary drastically depending on the color used to highlight the cells. On HARD (15-20) images, the performance of the model trained with **Image** supervision can vary from 30% to 70% depending on which color (e.g. *purple* or *yellow*) was used.

- **CoT Trace:** Our original CoT Trace simply outlined the numbers on the path, without any mention of the row number, column number, row name, and column name of the cells in the highlighted path. This resulted in poor performance of

the model when trained with images. We then switched to a more verbose CoT, where the model was provided with the above details at each step of traversing the highlighted path, and the model's performance substantially improved.

**For *Consecutive Table Readout*, we find that the verbose CoT trace shows S2H generalization, and not the final solution that the model reports.** Hence, we report our evaluation performance for *Consecutive Table Readout* on the CoT trace.

### C.6.2. *Grid Navigation*

The major issue that we faced in *Grid Navigation* was the design of chain-of-thought reasoning steps to represent the Depth First Search trace. At multiple points, we found that current VLMs are fragile to read image inputs, and our CoT trace needed to be very explicit to train the model effectively on SIMPLE examples.

**An initial version of *Grid Navigation*:** In our first version, we designed an extremely simple dataset, where the grids only had a source cell, a destination cell, and a few cells marked by *red* color that represented *obstacles*.

- **Models failed to train on image-input without verbose details in CoT:** Our initial CoT would only provide the following at each DFS step: "[current cell]: [proposed next action]" without iterating through all invalid actions considered before proposing this action. e.g., a 3-step DFS step would look as follows:
  - (1, 1): right
  - (1, 2): down
  - (2, 2): backtrack

  where we don't explain why we need to "backtrack" at (2, 2). This made the model learn the following:

  1. answer formatting
  2. knowing how to retrieve the current location (row, col index) and the destination location
  3. knowing which action is preferred (the one that minimizes the distance towards destination)

  but the model never picked up on why we sometimes backtrack or sometimes take an action that is not the most preferred. At generation, it would ignore all obstacles and try to take the most preferred action.

  **On the other hand, we observed that the model could still recognize the reasoning for "backtracking" on text input and could get** $100\%$ **accuracy on SIMPLE text for Text supervision, and also** $100\%$ **accuracy on SIMPLE images for Mix supervision.** Thus, for cases where the model couldn't train with image-input but could train with text input, Mix was useful to train the model even for improving accuracy on in-domain examples. However, this setting was slightly different from our S2H generalization view, and so we decided to make the CoT more verbose.

- **In later attempts, we switched to a more verbose CoT:** We iterate through all possible actions at each state, giving reasons why that action is valid / invalid. e.g. a 3-step DFS trace that starts from cell (1, 1) will look as follows
  - Current cell: (1, 1): right would lead to (1, 2) which is available and not visited yet, so we can move right.
  - Current cell: (1, 2): down would lead to (2, 2) which is available and not visited yet, so we can move down
  - Current cell: (2, 2): right would lead to (2, 3) but it has an obstacle; down would lead to (3, 2) but it has an obstacle; left would lead to (2, 1) but it has an obstacle; we have no more action left, so backtrack

  The model now gets almost perfect S2H generalization on both text / image no matter which supervision we give. So we couldn't really compare the performance of different types of supervision. This was because once the model learns how to iterate through different actions and determine its validity, length generalization was trivial.

Thus, we switched to our current version of *Grid Navigation*, where the task additionally involved spatial reasoning of different combinations of objects and obstacles spread across the grid.

### C.6.3. *Visual Analogy*

Here, the main challenge is to recreate the Procedurally Generated Matrices (PGM) dataset first introduced in Hill et al. (2019) and Barrett et al. (2018), as the data generation code is not publicly available. Therefore, we try our best to recreate the data set with slight adaptations. Specifically, we have 10 variations in the attribute values for line_type, shape_quantity, and shape_position as in the original paper. For the rest of attributes line_color, shape_type, and shape_size,

we only include $\leq 5$ variations of attribute values. Meanwhile, as the original papers do not list all the attribute values used in the original data generation, nor was the source code publicly available, we decide upon the list of possible attribute values based on the consideration that they are clearly differentiable from a human perspective.

The original papers claim that solving PGM puzzles is a challenging vision task. While we acknowledge that our recreated version of the data reduces the complexity compared to the original version, we note that our adaptations do not qualitatively change the challenging nature of this task. As mentioned in Barrett et al. (2018), the challenge of effective knowledge composition comes mainly from the necessity to represent abstract logical rules in discrete symbolic explanations. They show that training with auxiliary information of *meta-targets* vectors that encode the relation, object, and attribute types as a binary string significantly helps abstract reasoning performance, and in particular, in terms of compositional generalization. Our text representations are inspired by the construction of the *meta-targets* vectors with many tweaks to fit into the context length of the model. We observe that by including the discrete representation of knowledge in the form of **Image-via-Text** supervision, **Mix** and **Mix+** show a much better S2H generalization on image input, which aligns with previous observations in Barrett et al. (2018).

# D. Training Details

We first prepare *Eagle-X2-Llama3-8B*, a variation of *Eagle-X5-8B* (Shi et al., 2025). We choose Llama3-8B-Instruct (Dubey et al., 2024) as the LLM backbone for its good reasoning capability. We choose CLIP-448 (Radford et al., 2021) and ConvNeXt (Liu et al., 2022) as the visual encoders because previous works show that combining the two leads to a significant improvement, whereas any additional visual encoder leads to marginal improvement (Shi et al., 2025).

At the beginning of the project, the codebase released by Shi et al. (2025) was incomplete. To incorporate the Llama3-8B model architecture and the tokenizer, we adapt the codebase from Tong et al. (2024).

We use the same 595k pretraining data from Liu et al. (2023b) and 1.8M finetuning (visual instruction tuning) data from Shi et al. (2025). We use Deepspeed ZeRO Stage 2 (Rasley et al., 2020) for a Distributed Data Parallel (DDP) training on 8 GPUs on a HPC Cluster. We use the AdamW optimizer with no weight decay (i.e., equivalent to Adam), a learning rate schedule with a linear warmup of 0.03 and cosine decay to zero. We truncate the trail of any text that exceeds the maximum number of text tokens (2048). During pretraining, only the adapter is trained, whereas in all other stages of training, all weights in the model are unfrozen.

With this *Eagle-X2-Llama3-8B* as the base model, we then continuously finetune it on different data mixtures across our synthetic tasks. In Table 3, we report some key hyperparameters.

*Table 3.* **Hyperparameter settings:** For all values not reported here, we use the same values as in Shi et al. (2025).

|                     | Batch Size | LR   | Epochs | Total # Data        | Max # Text Tokens |
|---------------------|------------|------|--------|---------------------|-------------------|
| Pretraining         | 256        | 1e-3 | 1      | 595k                | 2048              |
| Finetuning          | 128        | 2e-5 | 1      | 1,809k              | 2048              |
| Finetuning on Task  | 128        | 2e-5 | experiment-specific |          | 2048              |

# E. Evaluation Details

We extend the VLMEvalKit (Duan et al., 2024) to evaluate the finetuned *Eagle-X2-Llama3-8B* on held-out data. For generation, we apply greedy decoding and generate up to 2048 tokens.

## E.1. Evaluation on *Consecutive Table Readout* and *Table Readout*

$CoT(\mathbf{x})$ visits each cell sequentially on the path, by giving the row and column index, row and column names, and the value in the cell (see Figures 32 and 33). The final answer $f(\mathbf{x})$ gives the list of numbers again, and also sum of the numbers. We evaluate by simply checking whether the list of numbers are correct. Furthermore, because this list of numbers can be extracted from both the final answer and also the CoT, we report the best performance out of the two. On *Consecutive Table Readout*, we find that we get the best performance on HARD examples by extracting the numbers from CoT. On the other hand, for *Table Readout*, there isn't much difference between extracting numbers from CoT and extracting them from the final answer.

## E.2. Evaluation on *Grid Navigation*

$CoT(\mathbf{x})$ records the sequence of visited cells during a depth-first search (DFS) from the start to the end cell. At each visited cell, the trace includes a full description of neighboring cells and whether they are available for the next step. The DFS algorithm always prefers directions that minimize the distance towards the nearest uncollected object, or the destination (if all objects are collected). If no directions are possible, we backtrack to the most previously visited cell. The final answer $f(\mathbf{x})$ is a simplified sequence of directions (left, right, up, down) that connect the start and destination cells, where all backtrack movements are removed from the stack (see Figures 34 and 35). We evaluate by simulating the movements in the sequence returned by the model and checking if we arrive at the destination after collecting all objects and avoiding obstacles.

## E.3. Evaluation on *Visual Analogy*

$CoT(\mathbf{x})$ enumerates all the values of the tasks-relevant attributes for each panel with the conclusion of whether there exists a logical pattern among those values for each attribute domain in the examples. The trace includes a summary sentence of what (domain, relation) pattern the two examples demonstrate. After that, the trace performs the same enumeration process for the query panels. It then looks at the options and checks whether it is consistent with the desired relation given the attribute values in the query panels. The final answer $f(\mathbf{x})$ identifies the pattern in the form of (domain, relation) (e.g. `(line type, XOR)`) for all examples and the query combined with each option, as well as the final answer of the correct option. The evaluation checks whether the identified patterns and the final answer are correct.

# F. Consistent Results on Another Model Family and Size

## F.1. Training Details

We take *Qwen2.5-VL-3B-Instruct* and *7B-Instruct* (Bai et al., 2025) as the base model and finetune it with on different data mixtures across our synthetic tasks.

We use the `SFTTrainer` class in the `trl` package. We employ FSDP (Zhao et al., 2023) for training on 8 GPUs on a HPC Cluster. We use the AdamW optimizer with no weight decay (i.e., equivalent to Adam), a learning rate schedule with a linear warmup of 0.03 and cosine decay to zero.

Due to a deficiency in the `trl` package, we slightly modify Algorithm 1 to ensure that each gradient computation (before gradient accumulation) includes a training example with $\mathbf{x}^{(i)}$ unless we train exclusively on $\mathbf{x}^{(t)}$ (while also maintaining randomness in the data). Instead of concatenating and randomly shuffling the entire dataset, we first shuffle the examples within each supervision, then interleave the individually shuffled data. For example, to construct **Mix+**, we first shuffle **Text**, **Image**, **Image-via-Text**, and HARD-**Text** individually, then construct the final dataset by repeatedly taking the next examples from (**Text**, **Image**, **Image-via-Text**, HARD-**Text**, HARD-**Text**, HARD-**Text**) respectively.

In Table 4, we report some key hyperparameters. Note that Bai et al. (2025) do not report the hyperparameters for their internal training, so we used the hyperparameters for *Eagle-X2-Llama-8B* as closely as possible. However, we noticed that for *Qwen2.5-VL-7B*, training on *Grid Navigation* or *Visual Analogy* with a learning rate of 2e-5 often broke the model (e.g., model starts outputting Chinese tokens), so we had to adjust the learning rate to 5e-6 or 2e-6.

*Table 4.* **Hyperparameter settings for Qwen2.5-VL.**

| Model Size | Task | Batch Size | LR | Epochs | Total # Data |
|:---:|:---:|:---:|:---:|:---:|:---:|
| 3B | All | 128 | 2e-5 | | experiment-specific |
| 7B | *Consecutive Table Readout* | 128 | 2e-5 | | experiment-specific |
| 7B | *Table Readout* | 128 | 2e-5 | | experiment-specific |
| 7B | *Grid Navigation* | 128 | 2e-6, 5e-6 | | experiment-specific |
| 7B | *Visual Analogy* | 128 | 2e-6, 5e-6 | | experiment-specific |

## F.2. Evaluation Details

We extend the VLMEvalKit (Duan et al., 2024) to evaluate the finetuned *Qwen2.5-VL-3B-Instruct* and *7B-Instruct* on the same held-out data as used for the main experiments. For generation, we apply the default setting for the *Qwen2.5-VL* family (top $p$=0.001 and temperature=0.01) and generate up to 2048 tokens. We set the maximum number of pixels to be $1280 \times 28 \times 28$.

## F.3. Results

In Tables 5 and 6, we report the S2H-generalization on image for most supervision types we consider.

For *Consecutive Table Readout*, we find that *Qwen2.5-VL* (both 3B and 7B models) completely fail to solve HARD-text examples even when HARD **Text** is a part of the training data (e.g., **Mix+**). For this reason, we relax the definition of hardness and instead train with MEDIUM-text (if applicable) and evaluate on MEDIUM-text and MEDIUM-image (see Section 3 for the definitions of MEDIUM and HARD). Even then, *Qwen2.5-VL-3B-Instruct* fail to solve MEDIUM-text examples even when it is explicitly trained with MEDIUM **Text**. Therefore, none of our proposed methods can improve S2H-generalization on image. However, we find that even though *Qwen2.5-VL-7B-Instruct* does not S2H-generalize on text (which is understandable since different models can S2H-generalize on different tasks), our proposed supervision types for non-S2H generalizing tasks (**Mix+** and **Align-Mix+**) successfully improve the S2H-generalization on image.

For the other 3 tasks (*Table Readout*, *Grid Navigation*, and *Visual Analogy*), we generally observe a consistent result from the main text: 1) the models do not S2H-generalize on either text or image; 2) **Mix+** improves S2H-generalization on image by transferring the injected reasoning on HARD-text; 3) **Align-Mix+** further improves this generalization. Note that for *Grid Navigation*, and *Visual Analogy* on *Qwen2.5-VL-7B-Instruct*, we report the best result between the two learning rates (2e-6, 5e-6).

*Table 5.* **Results for Qwen2.5-VL-3B-Instruct:** For Text supervision, we evaluate on HARD-text, but for all other supervision types, we evaluate on HARD-image. For *Consecutive Table Readout*, we train with MEDIUM Text (if applicable) and evaluate on MEDIUM-image.

| Supervision | Consecutive Table Readout 30k | Table Readout 30k | 60k | Grid Navigation 30k | 60k | Visual Analogy 30k | 60k |
|---|---|---|---|---|---|---|---|
| Text (eval on HARD-text) | 3 | 8 | 11 | 0 | 15 | 0 | 0 |
| Image | 0 | 11 | 10 | 22 | 22 | 0 | 0 |
| Text+Image | 1 | 7 | 6 | 0 | 14 | 0 | 0 |
| Image-via-Text | 1 | 12 | 8 | 13 | 14 | 0 | 0 |
| Mix | 1 | 11 | 10 | 14 | 16 | 0 | 1 |
| Image-via-Text+ | 0 | 81 | 90 | 67 | 58 | 48 | 48 |
| Mix+ | 4 | 78 | 86 | 77 | 91 | 20 | 27 |
| Align-Mix+ | - | 66 | 91 | 80 | 91 | 38 | 42 |

*Table 6.* **Results for Qwen2.5-VL-7B-Instruct:** For Text supervision, we evaluate on HARD-text, but for all other supervision types, we evaluate on HARD-image. For *Consecutive Table Readout*, we train with MEDIUM Text (if applicable) and evaluate on MEDIUM-image.

| Supervision | Consecutive Table Readout 30k | Table Readout 30k | 60k | Grid Navigation 30k | 60k | Visual Analogy 30k | 60k |
|---|---|---|---|---|---|---|---|
| Text (eval on HARD-text) | 1 | 22 | 2 | 15 | 18 | 1 | 0 |
| Image | 0 | 18 | 17 | 14 | 29 | 0 | 0 |
| Text+Image | 4 | 8 | 5 | 6 | 11 | 0 | 0 |
| Image-via-Text | 36 | 9 | 13 | 13 | 18 | 0 | 0 |
| Mix | 52 | 8 | 17 | 15 | 12 | 0 | 0 |
| Image-via-Text+ | 73 | 82 | 88 | 75 | 67 | 41 | 44 |
| Mix+ | 72 | 13 | 66 | 69 | 85 | 12 | 17 |
| Align-Mix+ | - | 93 | 92 | 36 | 58 | 25 | 34 |

# G. Continued Discussion From Main Paper

## G.1. Comparisons at equal unique samples for *Consecutive Table Readout*

In Figure 3, we compare Text, Image, and Mix under the same $N_{\text{SIMPLE}}$, the total number of training data. Note that Mix is trained for only a single epoch, while the reported results for Text and Image are based on 2 and 3 epochs of training, respectively. We make these choices because, for Text, the S2H generalization performance peaks at 2 epochs and then declines sharply, whereas for Image, the S2H generalization performance sees a slight improvement between 2 and 3 epochs. As an illustrative example, Figure 13 shows the performance of Text and Image when $N_{\text{SIMPLE}}^u$, the number of unique samples, is fixed at $4 \times 10^4$. Consequently, in Figure 14, we revisit the results of Figure 3, this time explicitly indicating the number of unique samples $N_{\text{SIMPLE}}^u$ used.

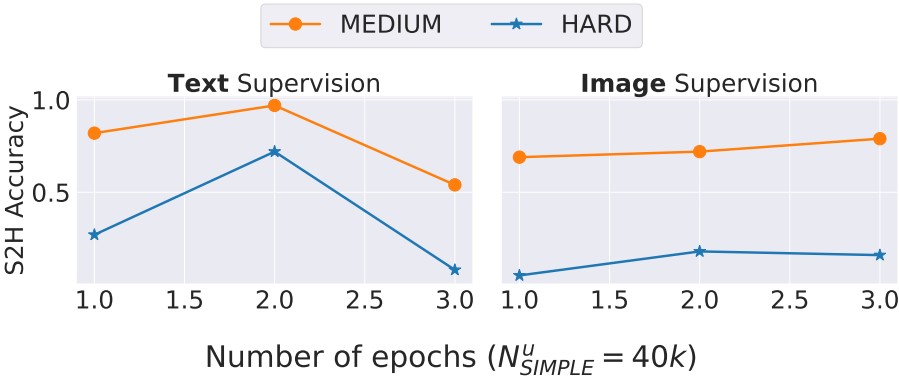

*Figure 13.* **Ablation on the number of epochs on *Consecutive Table Readout*:** We measure the S2H generalization performance of Text on HARD-text and Image on HARD-image with multi-epoch training, when $N_{\text{SIMPLE}}^u$ is fixed as $4 \times 10^4$. We observe that the generalization performance of Text supervision peaks at 2 epoch training, after which it drastically drops, while the generalization performance of Image supervision increases slightly between 2 and 3 epochs of training.

## G.2. Additional setting for an S2H-generalizing task

Here, we consider *Pattern-Heldout Visual Analogy* — a S2H-generalizing version of *Visual Analogy* — by defining an alternative version of HARD examples. We keep the definition of SIMPLE examples from Section 4 and Appendix C.5, but modify HARD instances to only measure analogical reasoning on held-out reasoning patterns, without requiring the domain to be different across the in-context examples.

That is, let $d_1, d_2, d_{\text{query}}$ denote the latent domains of the examples and the query, $r$ denote the latent logical relation to be applied on the latent domains, and $\mathcal{S}$ denote a held-out set of combinations $(d, r)$. The SIMPLE task contains puzzles where $d_1 = d_2 = d_{\text{query}}$ and $(d_{\text{query}}, r_{\text{query}}) \notin \mathcal{S}$, whereas the HARD task contains puzzles where $d_1 = d_2 = d_{\text{query}}$ and $(d_{\text{query}}, r_{\text{query}}) \in \mathcal{S}$. Note that in *Visual Analogy*, we had additionally required $d_1, d_2, d_{\text{query}}$ to be distinct for HARD puzzles.

In Table 7, we compare the S2H generalization performance of Text, Image, and Mix supervision on *Pattern-Heldout Visual Analogy*. The model learns the task more easily on text than on image: while the image S2H generalization for Image supervision is bounded by 32%, the text S2H generalization for Text supervision can reach 49% when trained on $24 \times 10^4$ data.

On the other hand, Mix supervision can transfer the S2H generalization from text to image and improve the performance on HARD-image (41% with $12 \times 10^4$ training data).

## G.3. Comparison at equal FLOPs for non S2H-generalizing tasks

Align-Mix+ uses an additional phase over Mix+, where training sequences from SIMPLE split are utilized. In Figure 5, however, we compare Mix+ and Align-Mix+ only in terms of the amount of training data used in the final phase. This raises a potential concern that Align-Mix+ might only appear stronger because it involves more total training FLOPs. To address this, Figure 15 presents a revised comparison, plotting Align-Mix+ against Mix+ in terms of the total training data employed across all stages. Under these conditions, Align-Mix+ still consistently outperforms Mix+.

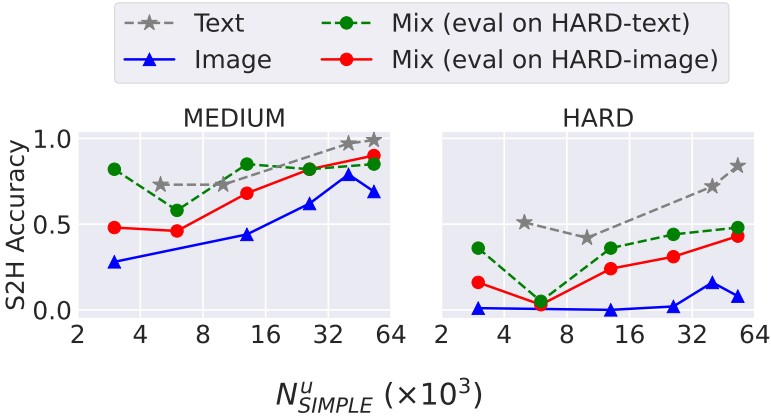

*Figure 14.* **Results on *Consecutive Table Readout* based on the number of unique samples** $N^u_{\text{SIMPLE}}$**:** Our observations from Section 3 hold true even when different types of supervision are compared at the same value $N^u_{\text{SIMPLE}}$, instead of $N_{\text{SIMPLE}}$.

*Table 7.* **Results on *Pattern-Heldout Visual Analogy*:** S2H generalization for **Text**, **Image**, and **Mix** supervision are reported on HARD-text and HARD-image examples after varying the number of training data in each strategy. S2H generalization on HARD-images under **Image** supervision peaks at 36%, while for HARD-text examples under **Text** supervision, it reaches 45.6% after $24 \times 10^4$ training examples. Leveraging the better performance on HARD-text, **Mix** supervision improves S2H generalization on HARD-images to 41% with $12 \times 10^4$ examples.

| | S2H accuracy on HARD-text | | | | S2H accuracy on HARD-image | | | |
| | Number of training data | | | | Number of training data | | | |
| **Supervision** | 30k | 60k | 120k | 240k | 30k | 60k | 120k | 240k |
|---|---|---|---|---|---|---|---|---|
| **Text** | - | 37.2 | 32.6 | 45.6 | 0.0 | 0.0 | 0.0 | 0.0 |
| **Image** | 0.0 | 0.0 | 0.0 | 0.0 | - | 27.0 | 35.6 | 34.0 |
| **Mix** | 31.2 | 42.4 | 49.0 | 39.8 | 24.6 | 35.0 | 41.0 | 39.6 |

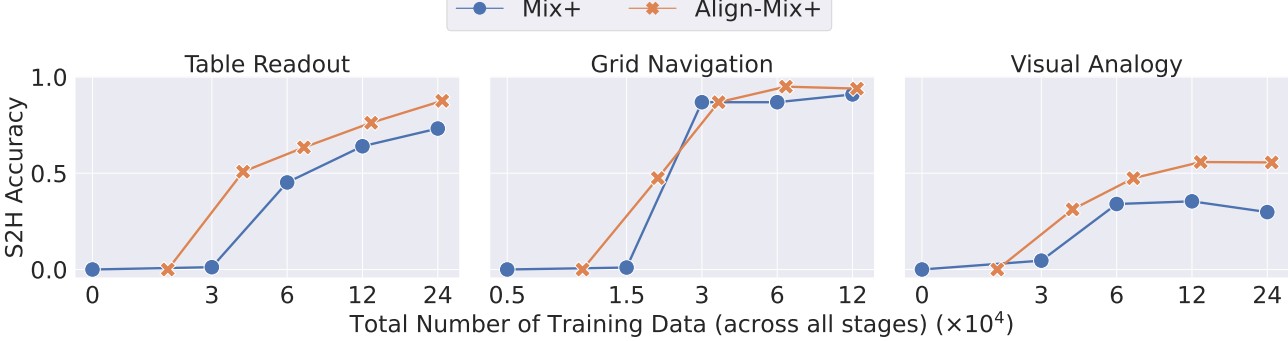

*Figure 15.* **Results on non S2H-generalizing tasks based on the total number of training data (Figure 6 but including the amount of data in the alignment stage):** **Align-Mix+** still outperforms **Mix+** when compared at the same amount of total training data.

### G.4. Transferring reasoning from image to text

In the main experiments, we tested whether S2H generalization can transfer from text inputs to image inputs. In Table 8, we observe that the transfer can happen in the opposite direction as well. After $24 \times 10^4$ training samples that now includes training data from HARD **Image** instead of HARD **Text**, a modified version of **Mix+** achieves S2H generalization accuracy of 86.0% on HARD-text input on *Table Readout* and 85.6% on HARD-text input on *Visual Analogy*. As a comparison, when trained with the same number of data, **Mix+** shows S2H generalization accuracy of 73.2% on HARD-image input on *Table Readout* and 35.4% on HARD-image input on *Visual Analogy*.

*Table 8.* **Ablation on transferring reasoning from image to text:** We modify **Mix+** to include HARD **Image** examples in training, instead of HARD **Text** examples, while keeping the same SIMPLE **Mix** supervision. Evaluation is now performed on HARD-text input. We observe that improving generalization performance on HARD-image input strongly transfers to HARD-text input.

| Number of training examples | Table Readout | | Visual Analogy | |
|---|---|---|---|---|
| | HARD-image (Included in training) | HARD-text (Excluded in training) | HARD-image (Included in training) | HARD-text (Excluded in training) |
| $3 \times 10^4$ | 72.0 | 34.0 | 86.8 | 81.4 |
| $6 \times 10^4$ | 98.2 | 70.4 | 97.8 | 80.0 |
| $12 \times 10^4$ | 99.4 | 76.4 | 94.6 | 86.4 |
| $24 \times 10^4$ | 99.8 | 86.0 | 99.6 | 85.6 |

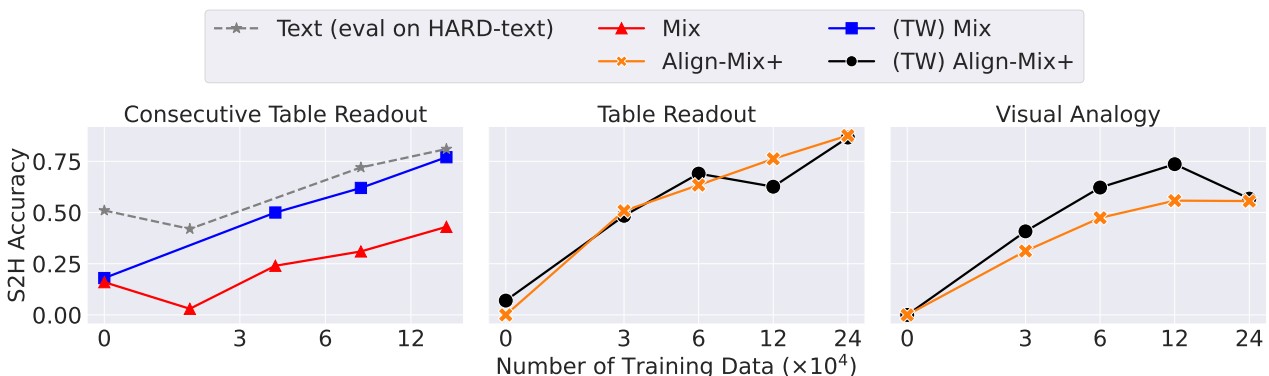

*Figure 16.* **Effect of text warm-up pretraining:** We report the S2H generalization on image with/without text warm-up before **Mix+** (*Consecutive Table Readout*) or **Align-Mix+** (*Table Readout* and *Visual Analogy*). S2H generalization on text from **Text** supervision serves as a reference (in gray dashed line). Text warm-up enhances image S2H generalization across tasks. (**TW**) **Mix** closes the text-image generalization gap on the HARD task for *Consecutive Table Readout*, while (**TW**) **Align-Mix+** outperforms **Align-Mix+** for *Visual Analogy*.

### G.5. Text warm-up pretraining

In Section 4, we observe that **Mix** fails to improve image generalization when the LLM backbone does not show strong generalization on text modality. Furthermore, models trained with **Mix+** show significantly better image generalization when the text reasoning capability of the LLM backbone is strengthened by fine-tuning on HARD **Text** examples. Hence, one may expect that the reasoning capability of the LLM backbone on HARD examples is a crucial factor for the reasoning capability to transfer to the image inputs. In this section, we investigate the effect of text generalization of the LLM backbone. Specifically, we simulate different levels of text reasoning ability by including a pretraining stage of the model on SIMPLE and HARD **Text** examples. We call this text-only training (during which only the LLM backbone is updated) before full finetuning of the VLM model *text warm-up pretraining* (**TW**). This stage of training only uses a small set of $10^4$ text examples (equal mix of SIMPLE and HARD for non S2H-generalizable tasks, just SIMPLE examples for *Consecutive Table Readout*). Our results are shown in Figure 16.

We observe that the additional TW training further boosts the image generalization. In particular, (**TW**) **Mix** closes the modality imbalance, reflected by the image-text generalization gap on the HARD (25-30) task for *Consecutive Table Readout*. On *Visual Analogy*, (**TW**) **Align-Mix+** outperforms **Align-Mix+** by $15\%$p with $12 \times 10^4$ training data. These results suggest that future stronger LLM backbone can further close the generalization gap between text and image modalities using our proposed strategy.

### G.6. Utility of our synthetic datasets for existing evaluation benchmarks

Existing evaluation benchmarks test the ability of VLMs to perform OCR, chart interpretation, image reasoning, and caption generation. However, they primarily test the ability to generate short answers on a given question. On the other hand, our created tasks evaluate long form reasoning generation from models. As such, our fine-tuned models quickly forget to return short responses during training and struggle on existing benchmarks.

Nonetheless, we assess the utility of our synthetic datasets to existing benchmarks by including them during visual instruction tuning. That is, we prepare two different versions of an alternate base model *Eagle+Synthetic-X2-Llama-8B*: 1) where 30 / 60 / 120k of our **Image** training mixture (equal mix of SIMPLE and HARD) has been mixed in with the 1.8M finetuning data; or where 240k of our **Mix+** training mixture (80k for each synthetic task) has been mixed in with the 1.8M finetuning data.

Results in Table 9 demonstrate consistent improvements across tasks such as OCR, chart interpretation, and multimodal understanding. However, a decline in performance is also observed on binary classification (yes/no response) benchmarks, such as MME. These findings indicate that the proposed synthetic datasets can be valuable for future research. Further investigation is necessary to determine how long reasoning datasets like our proposed tasks can be best leveraged to enhance general reasoning capabilities (e.g. Gao et al. (2024)).

*Table 9.* **Utility of our synthetic data:** We compare the benchmark results of *Eagle-X2-Llama-8B*, solely instruction tuned on Eagle-1.8M dataset, and *Eagle+Synthetic-X2-Llama-8B*, instruction tuned on a mixture of Eagle-1.8M and our synthetic data mixture. Including our data can improve model's performance on OCR and chart reasoning benchmarks, but may hurt performance on benchmarks where models need to output a yes/no answer (marked with *) or a short phrase (marked with **). †: performance reported on validation set.

| Evaluation Benchmark | Visual Instruction Tuning Dataset | | | | |
|---|---|---|---|---|---|
| | Eagle-1.8M | Eagle-1.8M + SIMPLE **Image** and HARD **Image** | | | Eagle-1.8M + **Mix+** mixture |
| | | (30k) | (60k) | (120k) | (240k) |
| MMMU† | 35.4 | 38.2 | **38.8** | 38.7 | 36.3 |
| MME* | **1529** | 1242 | 1377 | 1376 | 1364 |
| MMBench | 67.6 | **69.2** | 68.4 | 67.5 | **69.2** |
| POPE* | 86.6 | 88.7 | **88.9** | 87.6 | 87.5 |
| TextVQA** | 66.8 | 66.5 | **66.9** | 66.8 | 65.8 |
| OCR(Bench) | 47.3 | **50.9** | 50.4 | 47.0 | 48.2 |
| ChartQA | 69.6 | **71.6** | 69.8 | 70.5 | 69.4 |
| CharXiv-Reasoning† | 16.8 | 16.4 | 16.5 | 17.0 | **17.2** |
| CharXiv-Descriptive† | 30.7 | 28.3 | **35.8** | 31.1 | 34.4 |

**Reported benchmarks:** Here is a summary of the reported evaluation benchmarks.

- MMMU (Yue et al., 2024): Evaluates on multi-discipline tasks measuring college-level subject knowledge and reasoning.

- MME (Fu et al., 2023): Evaluates both perception and cognition abilities across 14 subtasks with yes/no answers.

- MMBench (Liu et al., 2025): Evaluates on VQA, which includes both multiple-choice and free-form answers.

- POPE (Li et al., 2023): Evaluates object hallucination with yes/no answers.

- TextVQA (Singh et al., 2019): Evaluates understanding and reading text within images with short-phrase answers.

- OCR(Bench) (Liu et al., 2024b): Evaluates on Character Recognition (OCR) capabilities across 29 datasets covering text / handwritten mathematical expression recognition, key information extraction, and scene text / document VQA.

- CharXiv (Wang et al., 2024c): Evaluates on chart understanding based on 2323 charts from arXiv papers, paired with descriptive and reasoning questions, covering 8 major academic subjects.

# H. Continued Discussion on Gradient Alignment

**Augmentation in notation:** Suppose the current model parameters are given by $\theta$. We will slightly augment our notation to include the model's current parameters. At model parameter $\theta$, we will use $l_{(I;S)}(\mathbf{x}; \theta)$ to denote the loss on **Image** example for data $\mathbf{x}$ and loss $l^{(H)}_{(I;S)}(\theta) = \mathbb{E}_{\mathbf{x} \in \mathcal{X}_{\text{HARD}}} l_{(I;S)}(\mathbf{x}; \theta)$. Then, $\mathbf{g}_{\text{SIMPLE}}(\theta)$ and $\mathbf{g}_{\text{HARD}}(\theta)$ denote average gradients on $\mathcal{X}_{\text{SIMPLE}}$ and $\mathcal{X}_{\text{HARD}}$, i.e.

$$\mathbf{g}_{\text{SIMPLE}}(\theta) = \mathbb{E}_{\mathbf{x} \in \mathcal{X}_{\text{SIMPLE}}} \nabla l_{(I;S)}(\mathbf{x}; \theta), \quad \mathbf{g}_{\text{HARD}}(\theta) = \mathbb{E}_{\mathbf{x} \in \mathcal{X}_{\text{HARD}}} \nabla l_{(I;S)}(\mathbf{x}; \theta)$$

Recall that the gradient alignment score from Equation (3) is given by

$$\langle \mathbf{g}_{\text{SIMPLE}}(\theta), \mathbf{g}_{\text{HARD}}(\theta) \rangle / \langle \mathbf{g}_{\text{HARD}}(\theta), \mathbf{g}_{\text{HARD}}(\theta) \rangle. \tag{6}$$

In the following theorem, we show that the gradient alignment score quantifies the amount of loss that we can decrease in expectation on HARD **Image** examples by taking gradients on SIMPLE **Image** examples, relative to taking gradients on HARD **Image** examples.

**Theorem H.1.** *Suppose for a model $f_\theta$ with parameter $\theta$, loss $l_{(I;S)}$ is Lipschitz and has bounded gradient norm on $\mathcal{X}$ around parameters $\theta$, with $\|\mathbf{g}_{\text{HARD}}(\theta)\|_2 \neq 0$. The following holds true for expected drop in $l^{(H)}_{(I;S)}$ with SGD when using a random training sample from the SIMPLE task, compared to using a random training sample from the HARD task:*

$$\lim_{\eta \to 0} \frac{\mathbb{E}_{\substack{\mathbf{x} \in \mathcal{X}_{\text{SIMPLE}} \\ \mathbf{g} := \nabla l_{(I;S)}(\mathbf{x};\theta)}} \left[ l^{(H)}_{(I;S)}(\theta - \eta \mathbf{g}) - l^{(H)}_{(I;S)}(\theta) \right]}{\mathbb{E}_{\substack{\mathbf{x} \in \mathcal{X}_{\text{HARD}} \\ \tilde{\mathbf{g}} := \nabla l_{(I;S)}(\mathbf{x};\theta)}} \left[ l^{(H)}_{(I;S)}(\theta - \eta \tilde{\mathbf{g}}) - l^{(H)}_{(I;S)}(\theta) \right]} = \langle \mathbf{g}_{\text{HARD}}(\theta), \mathbf{g}_{\text{SIMPLE}}(\theta) \rangle / \langle \mathbf{g}_{\text{HARD}}(\theta), \mathbf{g}_{\text{HARD}}(\theta) \rangle.$$

The proof follows from standard convergence analysis of gradient descent algorithm (Nesterov, 2018).

## H.1. Proof of Theorem H.1

*Proof.* Say $\mathbf{g} = \nabla l_{(I;S)}(\mathbf{x}; \theta)$. By Taylor's theorem, we have the following for a small enough learning rate $\eta$,

$$l^{(H)}_{(I;S)}(\theta - \eta \mathbf{g}) - l^{(H)}_{(I;S)}(\theta) = -\eta \langle \nabla l^{(H)}_{(I;S)}(\theta), \mathbf{g} \rangle + \eta^2 \mathbf{g}^{\mathsf{T}} \left( \nabla^2 l^{(H)}_{(I;S)}(\theta - \eta_0 \mathbf{g}) \right) \mathbf{g}$$

for some $\eta_0 \in [0, \eta]$. We first note that $\nabla l^{(H)}_{(I;S)}(\theta) = \mathbf{g}_{\text{HARD}}(\theta)$. Next, since the loss is assumed to be Lipschitz,

$$\left| \mathbf{g}^{\mathsf{T}} \left( \nabla^2 l^{(H)}_{(I;S)}(\theta - \eta_0 \mathbf{g}) \right) \mathbf{g} \right| \leq L \|\mathbf{g}\|_2^2$$

where $L$ is the Lipschitz constant for the loss. Since the gradient norms are also assumed to be bounded, we have

$$l^{(H)}_{(I;S)}(\theta - \eta \mathbf{g}) - l^{(H)}_{(I;S)}(\theta) = -\eta \langle \mathbf{g}_{\text{HARD}}(\theta), \mathbf{g} \rangle + \mathcal{O}(\eta^2),$$

First assume $\mathbf{x} \in \mathcal{X}_{\text{SIMPLE}}$. By taking expectation over $\mathbf{x}$,

$$\mathbb{E}_{\substack{\mathbf{x} \in \mathcal{X}_{\text{SIMPLE}} \\ \mathbf{g} := \nabla l_{(I;S)}(\mathbf{x};\theta)}} \left[ l^{(H)}_{(I;S)}(\theta - \eta \mathbf{g}) - l^{(H)}_{(I;S)}(\theta) \right] = -\eta \left\langle \mathbf{g}_{\text{HARD}}(\theta), \mathbb{E}_{\mathbf{x} \in \mathcal{X}_{\text{SIMPLE}}} \nabla l_{(I;S)}(\mathbf{x}; \theta) \right\rangle + \mathcal{O}(\eta^2)$$

$$= -\eta \left\langle \mathbf{g}_{\text{HARD}}(\theta), \mathbf{g}_{\text{SIMPLE}}(\theta) \right\rangle + \mathcal{O}(\eta^2)$$

Similarly, assume $\tilde{\mathbf{g}} = \nabla l_{(I;S)}(\mathbf{x}; \theta)$ where $\mathbf{x} \in \mathcal{X}_{\text{HARD}}$. By taking expectation over $\mathbf{x}$,

$$\mathbb{E}_{\substack{\mathbf{x} \in \mathcal{X}_{\text{HARD}} \\ \tilde{\mathbf{g}} := \nabla l_{(I;S)}(\mathbf{x};\theta)}} \left[ l^{(H)}_{(I;S)}(\theta - \eta \tilde{\mathbf{g}}) - l^{(H)}_{(I;S)}(\theta) \right] = -\eta \left\langle \mathbf{g}_{\text{HARD}}(\theta), \mathbb{E}_{\mathbf{x} \in \mathcal{X}_{\text{HARD}}} \nabla l_{(I;S)}(\mathbf{x}; \theta) \right\rangle + \mathcal{O}(\eta^2)$$

$$= -\eta \left\langle \mathbf{g}_{\text{HARD}}(\theta), \mathbf{g}_{\text{HARD}}(\theta) \right\rangle + \mathcal{O}(\eta^2)$$

Therefore, we have

$$\frac{\mathbb{E}_{\substack{\mathbf{x} \in \mathcal{X}_{\text{SIMPLE}} \\ \mathbf{g} := \nabla l_{(I;S)}(\mathbf{x};\theta)}} \left[ l^{(H)}_{(I;S)}(\theta - \eta \mathbf{g}) - l^{(H)}_{(I;S)}(\theta) \right]}{\mathbb{E}_{\substack{\mathbf{x} \in \mathcal{X}_{\text{HARD}} \\ \tilde{\mathbf{g}} := \nabla l_{(I;S)}(\mathbf{x};\theta)}} \left[ l^{(H)}_{(I;S)}(\theta - \eta \tilde{\mathbf{g}}) - l^{(H)}_{(I;S)}(\theta) \right]} = \frac{-\eta \left\langle \mathbf{g}_{\text{HARD}}(\theta), \mathbf{g}_{\text{SIMPLE}}(\theta) \right\rangle + \mathcal{O}(\eta^2)}{-\eta \left\langle \mathbf{g}_{\text{HARD}}(\theta), \mathbf{g}_{\text{HARD}}(\theta) \right\rangle + \mathcal{O}(\eta^2)} = \frac{\left\langle \mathbf{g}_{\text{HARD}}(\theta), \mathbf{g}_{\text{SIMPLE}}(\theta) \right\rangle + \mathcal{O}(\eta)}{\left\langle \mathbf{g}_{\text{HARD}}(\theta), \mathbf{g}_{\text{HARD}}(\theta) \right\rangle + \mathcal{O}(\eta)}$$

Note that $\mathbf{g}_{\text{HARD}}(\theta)$ and $\mathbf{g}_{\text{SIMPLE}}(\theta)$ do not depend on the value of $\eta$. Furthermore, by assumption, $\|\mathbf{g}_{\text{HARD}}(\theta)\|_2 \neq 0$. We conclude by taking $\eta \to 0$ on both sides of the equation above. $\square$

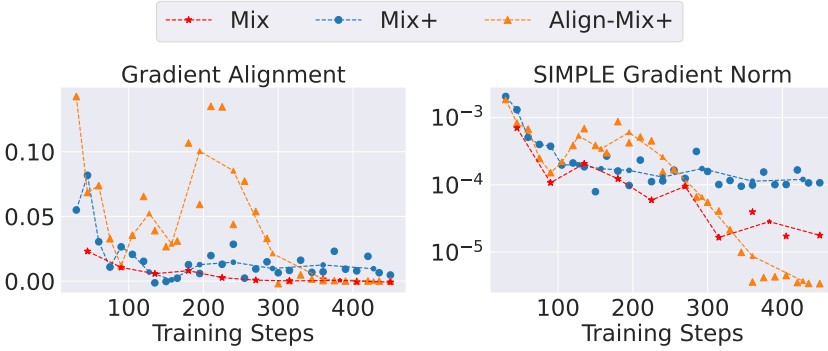

*Figure 17.* **Analysis of gradients on *Table Readout* (additional plots for Figure 9):** (Left) Gradient Alignment Score (Equation (3)); (Right) Average Gradient Norm on SIMPLE **Image** examples $\mathbb{E}_{\mathbf{x} \in \mathcal{X}_{\text{SIMPLE}}} \|\nabla l_{(I;S)}(\mathbf{x})\|_2$. **Align-Mix+** has higher gradient alignment score in the initial phases of training, where it also has higher gradient norm. **Mix+** shows higher gradient alignment score than **Mix** during the course of training.

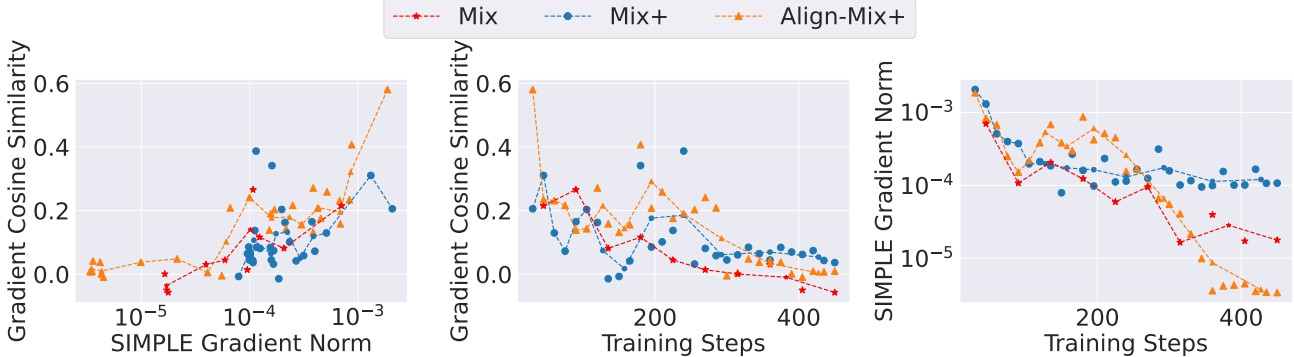

*Figure 18.* **Analysis of gradients on *Table Readout* (replacing gradient alignment score from Figures 9 and 17 with gradient cosine similarity):** (Left) Average Gradient Norm on SIMPLE **Image** examples ($\mathbb{E}_{\mathbf{x} \in \mathcal{X}_{\text{SIMPLE}}} \|\nabla l_{(I;S)}(\mathbf{x})\|_2$) vs. Gradient Cosine Similarity (Equation (7)) for different training checkpoints; (Middle) Gradient Cosine Similarity; (Right) Average Gradient Norm. Similar results hold.

## H.2. Additional measure 1: gradient cosine similarity

We additionally define the **gradient cosine similarity score** as the cosine similarity of gradients from $\mathcal{X}_{\text{SIMPLE}}$ and $\mathcal{X}_{\text{HARD}}$:

$$\text{Gradient Cosine Similarity:} \qquad \frac{\langle \mathbf{g}_{\text{SIMPLE}}(\theta), \mathbf{g}_{\text{HARD}}(\theta) \rangle}{\sqrt{\langle \mathbf{g}_{\text{HARD}}(\theta), \mathbf{g}_{\text{HARD}}(\theta) \rangle \cdot \langle \mathbf{g}_{\text{SIMPLE}}(\theta), \mathbf{g}_{\text{SIMPLE}}(\theta) \rangle}} \qquad (7)$$

Note that this measure ignores the norm of the gradients on $\mathcal{X}_{\text{SIMPLE}}$ that the model uses during training. Hence, this measure is not an entirely faithful measure on the alignment of the training updates to the loss on HARD **Image** examples. Figure 18 shows the gradient cosine similarity score across training strategies for *Table Readout*, which follows a similar pattern as the gradient alignment score in Figure 17.

## H.3. Additional measure 2: Adam update alignment

The gradient alignment score we defined earlier does not account for the fact that we use Adam optimizer (Kingma & Ba, 2015) during our experiments.

**Brief definition of Adam:** The Adam optimizer maintains two additional states, each representing the running average of the gradients and their squares during training. If $\mathbf{m}_t$ and $\mathbf{v}_t$ denote the two states, then the update rule at training step $t$

with a gradient $\mathbf{g}_t$ and learning rate $\eta$ is given by

$$\theta \leftarrow \theta - \eta h(\mathbf{g}_t), \qquad \text{where } h(\mathbf{g}_t) = \frac{(1-\beta_1)\mathbf{g}_t + \beta_1 \mathbf{m}_{t-1}}{\sqrt{(1-\beta_2)\mathbf{g}_t \odot \mathbf{g}_t + \beta_2 \mathbf{v}_{t-1}} + \epsilon}$$

$$\mathbf{m}_t \leftarrow (1-\beta_1)\mathbf{g}_t + \beta_1 \mathbf{m}_{t-1}, \qquad \mathbf{v}_t \leftarrow (1-\beta_2)\mathbf{g}_t \odot \mathbf{g}_t + \beta_2 \mathbf{v}_{t-1}$$

Here, $(\beta_1, \beta_2, \epsilon)$ are hyperparameters for the Adam optimizer and are set at $(0.9, 0.999, 10^{-8})$.

**Adam Update Alignment:** A true measure of alignment between SIMPLE and HARD training would be to compare $h(\cdot)$, the update vector under the Adam optimizer. However, that requires saving the Adam optimizer states throughout training. For storage efficiency purposes[13], we propose an alternate approximate measure called the **Adam update alignment score**. We compute the following two quantities for model $f_\theta$ with parameters $\theta$:

$$\mathbf{m}(\theta) := \mathbb{E}_{\mathbf{x} \in \mathcal{X}_{\text{SIMPLE}}} \left[ \nabla l_{(I;S)}(\mathbf{x}; \theta) \right]$$

$$\mathbf{v}(\theta) := \mathbb{E}_{\mathbf{x} \in \mathcal{X}_{\text{SIMPLE}}} \left[ \nabla l_{(I;S)}(\mathbf{x}; \theta) \odot \nabla l_{(I;S)}(\mathbf{x}; \theta) \right]$$

$\mathbf{m}$ and $\mathbf{v}$ are proxy measures for the Adam optimizer states. Then, we measure the alignment between gradients for $\mathcal{X}_{\text{HARD}}$ and $\mathcal{X}_{\text{SIMPLE}}$ as

$$\text{Adam Update Alignment Score:} \qquad \frac{\displaystyle \mathbb{E}_{\substack{\mathbf{x} \in \mathcal{X}_{\text{SIMPLE}} \\ \mathbf{g} := \nabla l_{(I;S)}(\mathbf{x}; \theta)}} \langle h(\mathbf{g}), \mathbf{g}_{\text{HARD}}(\theta) \rangle}{\displaystyle \mathbb{E}_{\substack{\mathbf{x} \in \mathcal{X}_{\text{HARD}} \\ \tilde{\mathbf{g}} := \nabla l_{(I;S)}(\mathbf{x}; \theta)}} \langle h(\tilde{\mathbf{g}}), \mathbf{g}_{\text{HARD}}(\theta) \rangle} \tag{8}$$

$$\text{where } h(\mathbf{g}) = \frac{(1-\beta_1)\mathbf{g} + \beta_1 \mathbf{m}(\theta)}{\sqrt{(1-\beta_2)\mathbf{g} \odot \mathbf{g} + \beta_2 \mathbf{v}(\theta)} + \epsilon} \text{ for any vector } \mathbf{g}$$

Intuitively, this measures how much the loss $l_{(I;S)}^{(H)}$ can be reduced in expectation by taking a gradient update step with Adam using SIMPLE **Image** examples, compared to taking a gradient update step with HARD **Image** examples, while maintaining the current Adam optimizer states. This can be formalized in the following theorem.

**Theorem H.2.** *Suppose for a model $f_\theta$ with parameter $\theta$, loss $l_{(I;S)}$ is Lipschitz and has bounded gradient norm on $\mathcal{X}$ around parameters $\theta$, with $\|\mathbf{g}_{\text{HARD}}(\theta)\|_2 \neq 0$. Consider a modified Adam update with learning rate $\eta$ with an arbitrary gradient $\mathbf{g}$, as follows:*

$$\theta \leftarrow \theta - \eta h(\mathbf{g})$$

$$\text{where } h(\mathbf{g}) = \frac{(1-\beta_1)\mathbf{g} + \beta_1 \mathbf{m}(\theta)}{\sqrt{(1-\beta_2)\mathbf{g} \odot \mathbf{g} + \beta_2 \mathbf{v}(\theta)} + \epsilon}.$$

*The following holds true for expected drop in $l_{(I;S)}^{(H)}$ with modified Adam update when using a random training sample from the SIMPLE task, compared to using a random training sample from the HARD task:*

$$\lim_{\eta \to 0} \frac{\displaystyle \mathbb{E}_{\substack{\mathbf{x} \in \mathcal{X}_{\text{SIMPLE}} \\ \mathbf{g} := \nabla l_{(I;S)}(\mathbf{x}; \theta)}} \left[ l_{(I;S)}^{(H)}(\theta - \eta h(\mathbf{g})) - l_{(I;S)}^{(H)}(\theta) \right]}{\displaystyle \mathbb{E}_{\substack{\mathbf{x} \in \mathcal{X}_{\text{HARD}} \\ \tilde{\mathbf{g}} := \nabla l_{(I;S)}(\mathbf{x}; \theta)}} \left[ l_{(I;S)}^{(H)}(\theta - \eta h(\tilde{\mathbf{g}})) - l_{(I;S)}^{(H)}(\theta) \right]} = \frac{\displaystyle \mathbb{E}_{\substack{\mathbf{x} \in \mathcal{X}_{\text{SIMPLE}} \\ \mathbf{g} := \nabla l_{(I;S)}(\mathbf{x}; \theta)}} \langle h(\mathbf{g}), \mathbf{g}_{\text{HARD}}(\theta) \rangle}{\displaystyle \mathbb{E}_{\substack{\mathbf{x} \in \mathcal{X}_{\text{HARD}} \\ \tilde{\mathbf{g}} := \nabla l_{(I;S)}(\mathbf{x}; \theta)}} \langle h(\tilde{\mathbf{g}}), \mathbf{g}_{\text{HARD}}(\theta) \rangle}$$

The proof is similar to that of Theorem H.1.

### H.4. Experimental results

In Figure 19, we present the analysis of gradients for different types of supervision on *Consecutive Table Readout*. Similar to the behavior of gradient alignment score in Figure 7, we observe that when measured against norm of gradients on SIMPLE **Image** examples, **Mix** achieves a higher Adam update alignment score than both **Text+Image** and **Image**. This shows that **Image-via-Text** supervision improves the alignment between SIMPLE and HARD **Image** gradients, when taking Adam gradient updates into account.

Similarly, for *Table Readout* in Figure 20, **Mix+** has a larger Adam update alignment during training. **Align-Mix+** further improves the Adam update alignment score when gradient norms are large during training.

---

[13]Remark: retrieving the actual Adam optimizer states requires an additional storage of 138GB per checkpoint.

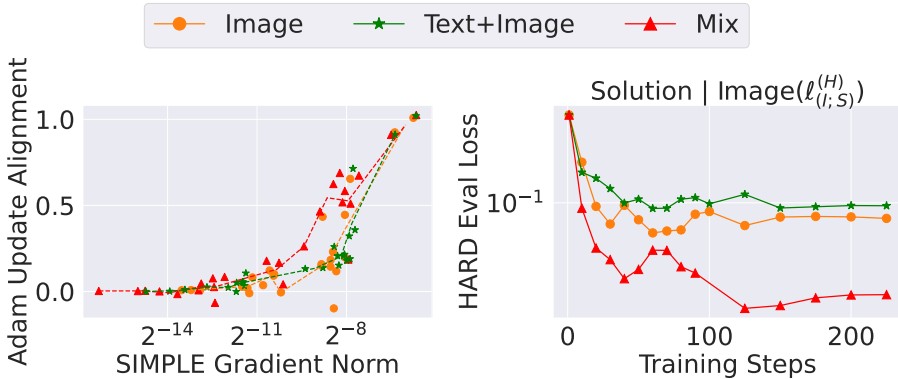

*Figure 19.* **Analysis of gradients on *Consecutive Table Readout* (replacing gradient alignment score from Figure 7 with Adam update alignment score):** (Left) Average Gradient Norm on SIMPLE Image examples ($\mathbb{E}_{\mathbf{x} \in \mathcal{X}_{\text{SIMPLE}}} \|\nabla l_{(I;S)}(\mathbf{x})\|_2$) vs. Adam Update Alignment Score (Equation (8)) for different training checkpoints; (Right) Average *Loss on solution given* HARD *image* ($l_{(I;S)}^{(H)}$) during training. Similar results hold.

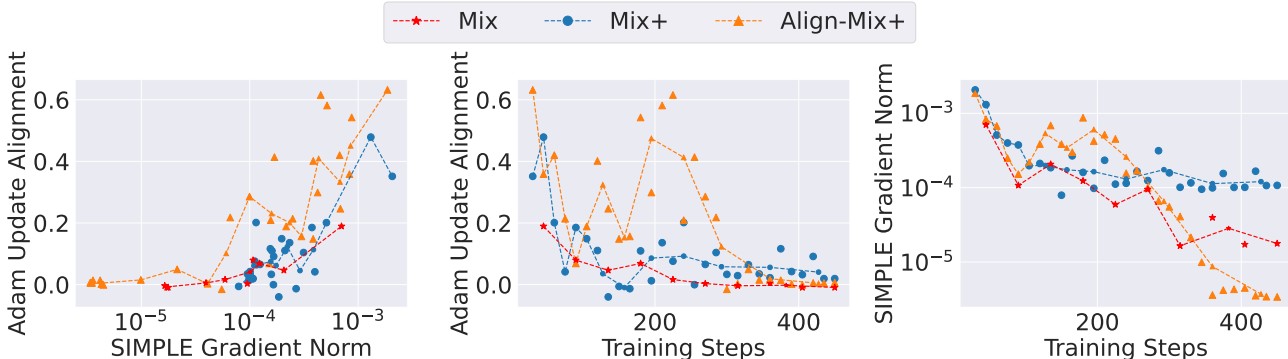

*Figure 20.* **Analysis of gradients on *Table Readout* (replacing gradient alignment score from Figures 9 and 17 with Adam update alignment score):** (Left) Average Gradient Norm on SIMPLE Image examples ($\mathbb{E}_{\mathbf{x} \in \mathcal{X}_{\text{SIMPLE}}} \|\nabla l_{(I;S)}(\mathbf{x})\|_2$) vs. Adam Update Alignment Score (Equation (8)) for different training checkpoints; (Middle) Adam Update Alignment Score; (Right) Average Gradient Norm. Similar results hold.

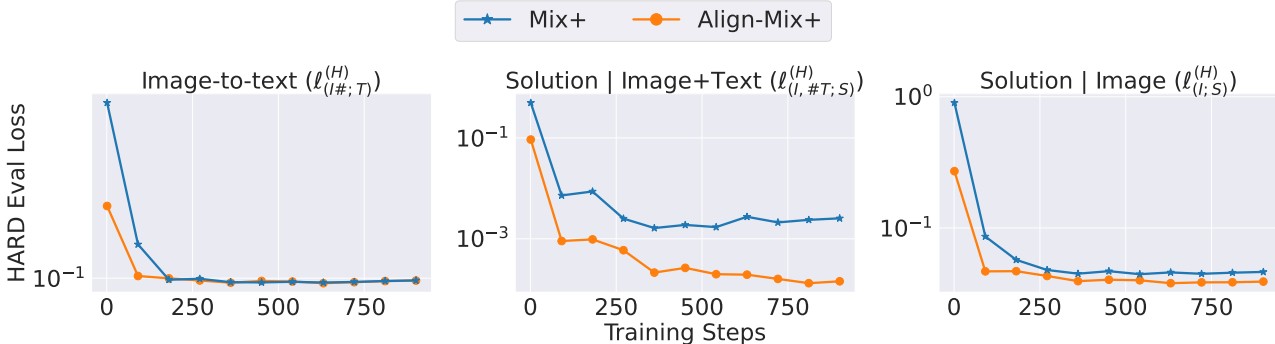

*Figure 21.* **Analysis of evaluation losses (repeating Figure 8 for *Visual Analogy*):** (Left) HARD *image-to-text conversion loss* ($l_{(I\#;T)}^{(H)}$ (Eq.4)); (Middle) *loss on solution given* HARD *image and text* ($l_{(I,\#T;S)}^{(H)}$ (Eq.5)); (Right) *loss on solution given* HARD *image* ($l_{(I;S)}^{(H)}$ (Eq.2)). Similar results hold.

# I. Additional Ablations

## I.1. Performance of other multimodal models

In Table 10, we present the performance of three closed source and two open source multimodal models on our three non S2H-generalizing tasks. Since we do not train these models, the format of the outputs is more flexible. For convenience, we propose alternative metrics to extract and evaluate on the models' predictions. For *Table Readout*, we instead evaluate with the final answer (sum of the sequence of numbers)[14]. For *Grid Navigation*, we evaluate with the same metric as in the main part of the paper — whether the proposed path can move from the start cell to the end without running into obstacles. For *Visual Analogy*, we evaluate with just the final option, which is a more lenient metric than the one suggested in the main part of the paper. Note that a random choice baseline should get 25%.

*Table 10.* **Performance of other multimodal models:** For convenience, we evaluate the models on a slightly different metric.

| Models | *Table Readout* | | *Grid Navigation* | | *Visual Analogy* | |
|---|---|---|---|---|---|---|
| | SIMPLE | HARD | SIMPLE | HARD | SIMPLE | HARD |
| Claude-3.5 Sonnet | 30.0 | 0.0 | 0.0 | 0.0 | 35.0 | 29.8 |
| GPT-4o | 19.0 | 0.0 | 0.0 | 0.0 | 19.8 | 18.4 |
| OpenAI o1 | 29.0 | - | 0.0 | - | 30.6 | - |
| Llama3.2-11B-Vision-Instruct | 4.0 | 0.0 | 0.0 | 0.0 | 16.2 | 17.8 |
| Pixtral-12B (Agrawal et al., 2024) | 9.0 | 0.2 | 0.0 | 0.0 | 24.6 | 21.2 |

## I.2. Ablation of the Mix+ supervision

The dataset composition of the **Mix+** supervision consists of three types of supervision in the SIMPLE task: **Text**, **Image**, and **Image-via-Text**. In this section, we ablate on the importance of each component of the data mixture in the training of **Mix+**, (**TW**) **Mix+**, and (**TW**) **Align-Mix+**. In Figure 22, we report the S2H generalization performance on image when the SIMPLE **Mix** supervision is replaced with *a varying data composition*.

In single-stage training (no text warm-up or alignment), **Image-via-Text** is the key component of success, as evidenced by the strong performance of **Image-via-Text+** supervision. As noted in Section 4, **Mix+** can match the performance by explicitly prompting the resulting model to convert the image first, which comes at a cost of around 1.7x generated tokens at inference time.

In multi-stage training (either (**TW**) or (**TW**) **Align-**), the benefits of **Mix+** are more significant. Specifically, among all other types of supervision with text warm-up, (**TW**) **Mix+** is able to outperform the others by at least 1.7x, while retaining efficient inference costs, unlike (**TW**) **Image-via-Text+**. Among all types of supervision with text warm-up and alignment, (**TW**) **Align-Mix+** achieves the highest performance.

## I.3. Ablation of the reasoning alignment phase (Align-)

We perform two ablations for the first phase of the **Align-Mix+** supervision. In Figure 23, we report the S2H generalization generalization performance on image of models trained with *a varying amount of data* in the first phase of **Align-Mix+**, with the amount of **Mix+** data fixed in the second phase. We don't observe a monotonic improvement in performance when increasing the amount of data in the first phase. In Table 11, we report the S2H generalization generalization performance on image of models trained with *a varying data composition* in the first phase of **Align-Mix+**. Our choice of **Text** and **Image-via-Text** from the main section gives the best performance on average on *Table Readout* and *Visual Analogy*.

## I.4. Ablation of the text warm-up pretraining phase (TW):

We ablate on the effect of the training data size during the text warm-up. In particular, we are interested in whether models with better reasoning capability on text can achieve better image generalization. To do so, we vary the number of training data used for text warm-up between $\{1, 2, 3\} \times 10^4$ and plot the performance of the warmed-up LLM on HARD-text examples against the performance of the final trained model on HARD-image examples. We report the performance on *Visual Analogy* in Figure 24. We observe that model's text performance improves with more training data being used for the text warm-up as expected. However, there is no clear linear correlation between the text capability of the model checkpoint after the warm-up

---

[14]The closed source models have access to tool-use, so in theory, this should be an equivalent, if not more lenient, metric.

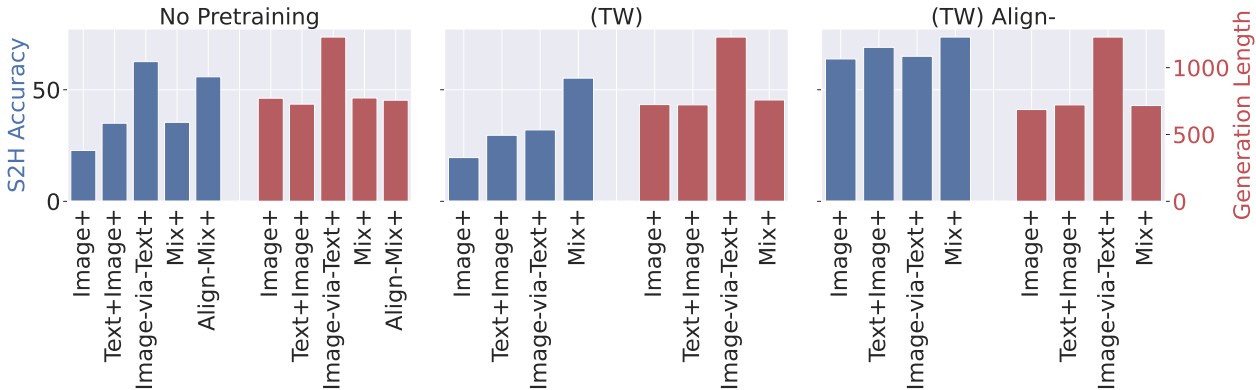

*Figure 22.* **Ablation on the SIMPLE data composition of Mix+ on *Visual Analogy*:** Instead of **Mix**, we use different types of SIMPLE supervision in (Left) **Mix+**; (Middle) (**TW**) **Mix+**; (Right) (**TW**) **Align-Mix+**. The main phase (**Mix+**) uses $12 \times 10^4$ training data.

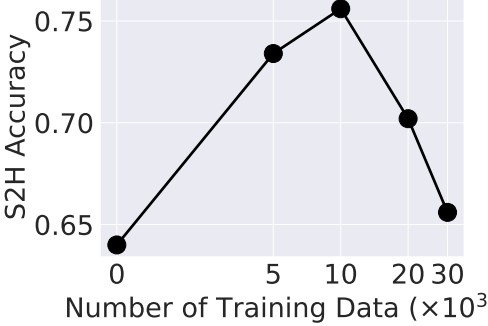

*Figure 23.* **Ablation on the amount of data for the alignment phase of Align-Mix+ on *Table Readout*:** Second phase (**Mix+**) uses $12 \times 10^4$ training data. We don't observe a monotonic improvement in generalization performance with increasing number of training samples in the first phase.

| Corresponding Name | SIMPLE data composition in phase 1 (**Align-Mix+**) | Accuracy (after phase 2) |
|---|---|---|
| | *Table Readout* | |
| **Align-** | Text, Image-via-Text | 0.76 |
| **Text+Image** | Text, Image | 0.52 |
| **Image-via-Text** | Image-via-Text | 0.77 |
| **Mix** | Text, Image, Image-via-Text | 0.74 |
| | *Visual Analogy* | |
| **Align-** | Text, Image-via-Text | 0.66 |
| **Text+Image** | Text, Image | 0.19 |
| **Image-via-Text** | Image-via-Text | 0.51 |
| **Mix** | Text, Image, Image-via-Text | 0.46 |

*Table 11.* **Ablation on the SIMPLE data composition for the alignment phase of Align-Mix+:** Amount of data for the alignment phase is fixed at $10^4$. Second phase (**Mix+**) uses $12 \times 10^4$ training data. Performance is reported on a validation set with 100 HARD-image examples. Our composition of **Text** and **Image-via-Text** on the SIMPLE task performs best on average on *Table Readout* and *Visual Analogy*.

training stage and image S2H generalization of the final model. Specifically, a model with $3 \times 10^4$ warm-up performs the best for the (**TW**) **Mix+** supervision, while a $10^4$ warm-up works the best with the (**TW**) **Align-Mix+** supervision. Meanwhile, we observe that (**TW**) **Align-Mix+** supervision can universally achieve better S2H generalization on image than the (**TW**) **Mix+** across all data scales. We conclude that an improved text capability by itself is insufficient to guarantee good transfer to image modality. We expect future VLMs with both stronger LLM backbone and better modality alignment can further leverage the text performance and transfer it to images.

### I.5. Requirement of text representation

One potential limitation of our proposed training strategies is the requirement of a text representation corresponding to the image. In *Consecutive Table Readout*, *Table Readout*, and *Grid Navigation*, we use the LaTeX code of the table or grid, which is considered to be perfectly aligned with the image. In reality, it may be challenging to find an exactly equivalent text description or representation of a real-world image, as many minute visual features cannot be captured by language. We show that our proposed training strategy does not require perfect alignment between the text and the image representation to work. For *Visual Analogy* experiments, the text description of the puzzle in the image is lossy: it only enumerates unique values of all task-relevant attributes without encoding the object to which each corresponds, so one cannot recover the original image given the description (see examples in Figures 36 and 37). Models trained with our proposed training

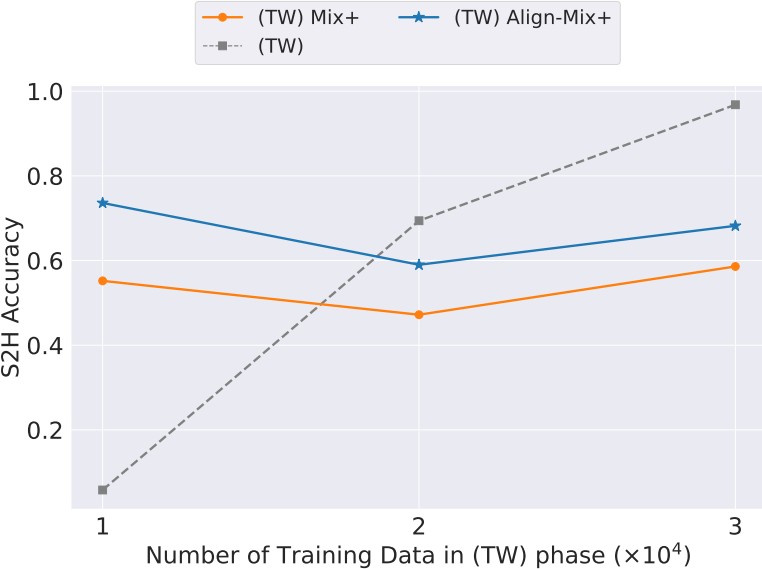

*Figure 24.* **Ablation on the amount of data for the text warm-up phase on *Visual Analogy*:** Second phase (**Mix+**) uses $12 \times 10^4$ training data. Using more data for the warm-up stage results in a stronger LLM backbone with better HARD-text performance (gray dashed line), but does not necessarily lead to better image S2H generalization of the final model trained with our proposed strategy. This suggests that a stronger text capability is not the only factor that induces S2H generalization on image.

strategies (**Mix+**, **Align-Mix+**, (**TW**) **Align-Mix+**) all demonstrate significant improvements in image generalization (Figure 6), testifying that our methods work with lossy text representation.

**Lossless text representation for *Visual Analogy*:** We additionally conduct experiments where the text representation of the puzzle in the image is a lossless representation. We represent the panels in the puzzle as a code defining each object as a set of attributes. Each geometric object is represented by the values its 5 attributes: {`shape type`, `shape_color`, `shape_size`, `shape_quantity`, `shape_position`}, while lines are defined by their 2 attributes: {`line_type`, `line_color`}. In order to fit to the context length of the VLM, we describe each object in shorthand notations. For example, for a panel in the puxxle that contains a circle and 2 rectangles, with attribute values {45 (gray-scale), 42 (pixels), 1, top-left} and {{0, 90}, {21, 21}, 2, top-right, bottom-left}, we will represent the panel as

CIR-45-42-TL;RECT-0-21-TR;RECT-90-21-BL

We give all details on how to parse the shorthand codes in the prompt. On the other hand, for the same example, the (Lossy) text representation would have been

type: circle, rectangle

color: 0,90

size: 21,42

quantity: 1,2

position: top-left, top-right, bottom-left

This substantially reduces the context length on average on our training dataset, and further removes the necessity of parsing a code. However, this isn't an exact representation of the image of the puzzle.

**Performance on lossy and lossless *Visual Analogy* tasks:** In Figure 25, we compare **Mix+**, **Align-Mix+**, and (**TW**) **Align-Mix+** for lossless and lossy *Visual Analogy* tasks at $12 \times 10^4$ training examples in the final phase (**Mix+**). Our observations reveal that a lossless text representation enhances S2H generalization performance on images for **Mix+**. However, for **Align-Mix+** and (**TW**) **Align-Mix+**, the lossy text representation leads to better S2H generalization performance on images. This discrepancy could be attributed to the complexity of the shorthand code in the lossless text representation, which requires additional parsing. We did not investigate this phenomenon further.

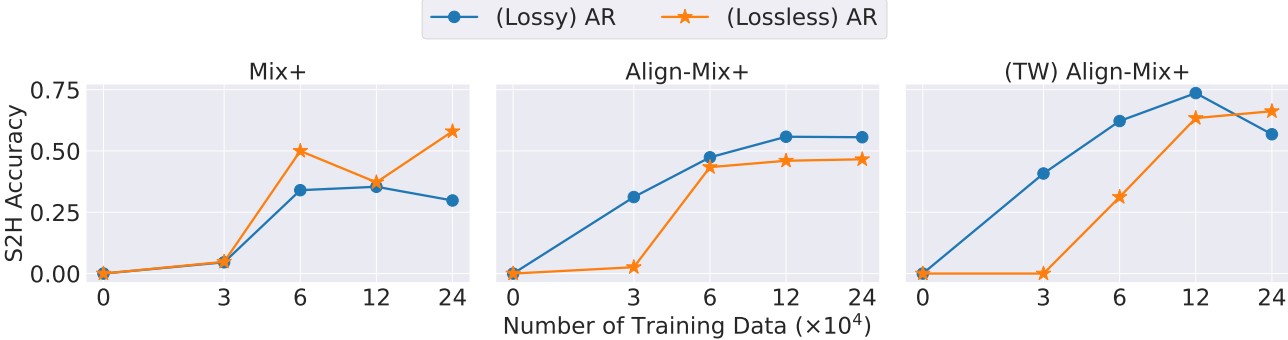

*Figure 25.* **Ablation on lossy vs. lossless *Visual Analogy*:** We measure the image S2H generalization of different types of supervision for two different versions of text representation for *Visual Analogy*. Models can perform better on Lossless *Visual Analogy* with **Mix+**. However, the trend can change with **Align-Mix+** and (**TW**) **Align-Mix+**.

### I.6. Explicit and implicit text conversion

In Appendix I.2, we find that explicit text conversion (**Image-via-Text**) is the key component in the data composition of the **Mix+** supervision. At inference time, however, models trained with **Mix+** reason directly on HARD images, without explicit text conversion. In Table 12, we observe that the trained models can still perform reasoning with explicit text conversion and that the conversion ability helps it reason.

**Mix+ models can convert image to text when prompted.**  If only an image input $\mathbf{x}^{(i)}$ is provided, **Mix+** models will *always* directly predict $CoT(\mathbf{x})$ and $f(\mathbf{x})$, never converting image to text (under greedy decoding). However, since the prompts used in the **Image** and **Image-via-Text** examples are the same, we can induce explicit text conversion in the final trained model by additionally providing the first word "Convert" of $P_{convert}$. We find that all trained model are *always* able to continue with explicit text conversion — they will generate the rest of $P_{convert}$ and an attempted conversion $\mathbf{x}^{(t)}$ before $CoT(\mathbf{x})$. The conversion accuracy is around $50\%$ on *Visual Analogy* and is almost $100\%$ on *Table Readout*.

**Explicit text conversion generally helps the model to reason on image data.**  Noticeably, the **Mix+** (240k) model improves S2H generalization accuracy from $73.2\%$ to $96.6\%$ with almost perfect text conversion accuracy of $99.2\%$ on *Table Readout*. On *Visual Analogy*, the **Mix+** (120k) model improves S2H generalization accuracy from $35.4\%$ to $51.8\%$ with a text conversion accuracy of $47\%$. The benefit of explicit text conversion gradually diminishes with multi-stage training strategies.

We also observe a slight drop in performance with prompted text conversion for models trained with (**TW**) **Align-Mix+** on *Visual Analogy*, which corresponds to a minor decline in reasoning performance with explicit text conversion. This suggests that the text warm-up training and alignment phase enable the model to close the gap between direct reasoning and reasoning with explicit text conversion, where the model learns to rely more equally on both text and image modalities, and doesn't require explicit text conversion for improved generalization performance.

**Models are robust against potential errors in the prompted text conversion.**  For models that are prompted to perform text conversion, we examine any negative side effects of this step. When the model does not correctly convert the image to its text format, we investigate whether to what extent the model's reasoning can be affected by the additional noises introduced by the text conversion step. Interestingly, we find that our final trained models are generally robust to such noises. On *Visual Analogy*, we find that the models trained with **Mix+**, (**TW**) **Mix+**, and (**TW**) **Align-Mix+** are still able to arrive at the correct reasoning solutions with accuracy $44.3\%$, $35.4\%$, and $63.1\%$ respectively on evaluation examples where the trained models make a mistake in text conversion.

### I.7. Explicit and implicit CoT

We use chain-of-thought (CoT) as a technique to boost the model's reasoning ability in all our experiments. In this section, we explore the role of CoT in our proposed strategies, as well as the possibility of transferring the reasoning capability from text to image modality without CoT. In Table 13, we report our observations on *Visual Analogy*. We note that similar observations hold for *Consecutive Table Readout* and *Table Readout*.

*Table 12.* **Ablation on explicitly prompting for text conversion:** When models are additionally prompted with "`Convert`," they exhibit the retained ability of text conversion. The conversion accuracy is near perfect on *Table Readout*. The S2H generalization performance with an additional prompt "`Convert`" (Prompted) improves from direct inference (Direct). The improvement margin diminishes with stronger Direct performance. All evaluations are on 500 HARD-image examples.

| Task | Supervision (Number of Training Data) | Direct | Prompted | Conversion acc |
|---|---|---|---|---|
| *Table Readout* | **Mix+** (240k) | 73.2 | 96.6 | 99.2 |
| | **Align**-**Mix+** (240k) | 87.6 | 98.0 | 100.0 |
| | (**TW**) **Align**-**Mix+** (240k) | 86.2 | 97.8 | 99.4 |
| *Visual Analogy* | **Mix+** (120k) | 35.4 | 51.8 | 47.0 |
| | (**TW**) **Mix+** (120k) | 55.2 | 62.8 | 49.0 |
| | (**TW**) **Align**-**Mix+** (120k) | 73.6 | 70.2 | 49.6 |

### I.7.1. REMOVING COT COMPLETELY

We first consider completely removing CoT from **Mix+** and observe the drop in performance measured by image S2H generalization. We experiment with **Mix+**, (**TW**) **Mix+**, and (**TW**) **Align**-**Mix+** supervision, in which we completely remove CoT from the last phase of training which has the **Mix+** supervision, while preserving the full CoT in the text warm-up (**TW**) and/or reasoning alignment (**Align**-) phases.

**Model does not learn when CoT is completely removed:** When CoT is completely removed from **Mix+**, performance drops to almost $0\%$ for all three types of supervision. We manually inspect the model's output and find that the generated reasoning on HARD-image inputs is identical to the expected behavior for SIMPLE instances, which indicates that the reasoning capability on HARD instances failed completely to transfer from the text to image modality.

### I.7.2. PROGRESSIVELY INTERNALIZING COT THROUGHOUT TRAINING

The failure above can be expected: for **Mix+** supervision, CoT may serve as a crucial technique to elicit good reasoning behaviors while for (**TW**) **Mix+** and (**TW**) **Align**-**Mix+**, the transition from training with full CoT to training without CoT can be too drastic for the model to adapt. Therefore, we consider a milder approach that trains the model to internalize reasoning by progressively removing CoT from the training (Deng et al., 2024). We train on the first $30\%$ of $12 \times 10^4$ **Mix+** examples with full CoT, the next $40\%$ of examples with progressively less CoT[15], and the last $30\%$ of examples with no CoT.

**Internalizing CoT during the Mix+ phase also fails:** In this scenario, we also observe that the model completely fails on image S2H generalization, getting almost $0\%$ S2H generalization on HARD-image examples for all three types of supervision strategies.

### I.7.3. INTERNALIZING COT DURING TEXT WARM-UP BEFORE REMOVING COT COMPLETELY

We also try a variant for the multi-phase approaches, where we internalize the CoT on the text input during a text warm-up ((**TW**)) stage and continue with **Mix+** with CoT completely removed.

**CoT can be internalized on text inputs:** We internalize the CoT on the text input during a slightly modified text warm-up phase of (**TW**) **Mix+**. Specifically, with $10^4$ training data that consists of an equal mix of SIMPLE **Text**, HARD **Text** supervision, and Eagle instruction tuning data (randomly sampled from 1.8M examples (Shi et al., 2025)), we train on the first $30\%$ examples with full CoT, the next $40\%$ examples with progressively less CoT, and the last $30\%$ examples without CoT as in previous experiments. After the warm-up phase of training, the model can achieve $97.8\%$ accuracy on HARD-text examples, which shows the model's ability to internalize reasoning on text inputs.

**Explicit CoT is "necessary" for the internalized reasoning to transfer to image:** We then continue with the **Mix+** supervision with all CoT removed. The final trained model completely fails with $0\%$ accuracy on the HARD-image examples. Similarly, examining model outputs reveals that the reasoning capability on HARD instances failed completely to transfer from the text to the image modality. Therefore, we conclude that CoT is "necessary" for the cross-modal transfer of knowledge to happen in our setting.

---

[15]split into 101 subsets of equal length, each training with $100\%, 99\%, \cdots, 0\%$ of total characters in the CoT.

*Table 13.* **Ablation on removing or internalizing CoT on *Visual Analogy*:** Preliminary attempts to completely or progressively remove CoT during the **Mix+** phase fails to generalize to HARD images, which shows the importance of CoT in our proposed strategies. *full, none, internalizing* CoT refer to including full CoT, completely removing CoT, and progressively removing CoT respectively. '-' means the corresponding phase was not included during training. Unless specified, all evaluations are reported on HARD-image examples.

| Type of supervision | Type of CoT | | | S2H accuracy (%) |
|---|---|---|---|---|
| | (**TW**) (10k) | **Align**- (10k) | **Mix+** (120k) | |
| **Mix+** | - | - | none | 0.6 |
| (**TW**) **Mix+** | full | - | none | 0.0 |
| (**TW**) **Align**-**Mix+** | full | full | none | 3.6 |
| **Mix+** | - | - | internalizing | 0.0 |
| (**TW**) **Mix+** | full | - | internalizing | 0.0 |
| (**TW**) **Align**-**Mix+** | full | full | internalizing | 3.4 |
| (**TW**) **Mix+** | internalizing | - | - | 97.8 (HARD-text) |
| (**TW**) **Mix+** | internalizing | - | none | 0.0 |

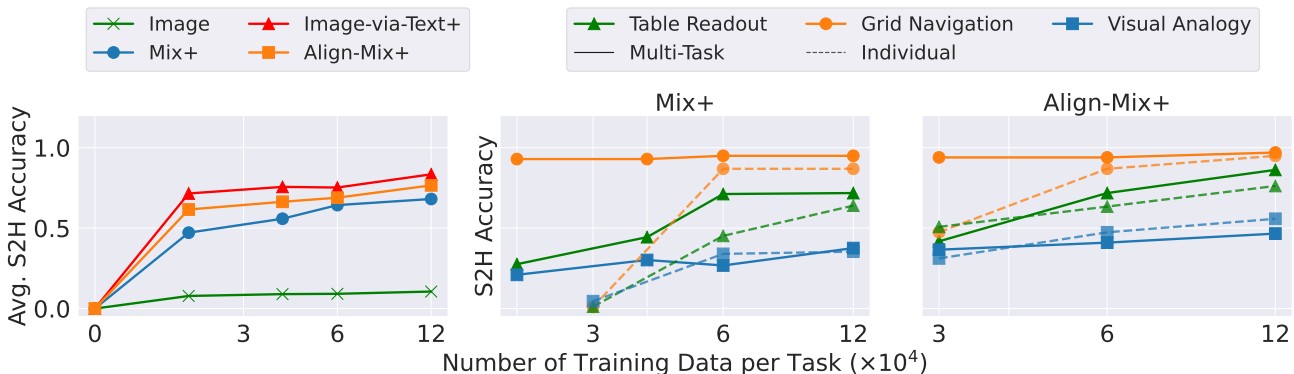

*Figure 26.* **Ablation on jointly training on all three non S2H-generalizing tasks:** (Left) Average S2H Generalization on image; (Middle, Right) Comparison of Trained Jointly vs. Individually. Similar to training on each task individually, **Mix+** and **Image**-**via**-**Text+** outperform **Image**, and **Align**-**Mix+** matches the performance of **Image**-**via**-**Text+**. Multi-task SFT boosts image S2H generalization for *Table Readout* and *Grid Navigation*, while *Visual Analogy* performance remains unchanged or slightly declines, indicating task interactions drive the cross-modal transfer of reasoning capabilities in multi-task training.

All results testify to our claim that CoT is important in our proposed training strategies. As the techniques used to internalize or remove the CoT dependency in our experiments are very preliminary, we are not eliminating the possibility of internalizing CoT in our setting. We note that to do so may require more careful, post hoc approaches, which we leave to future work.

### I.8. Multi-task training: jointly training on all three non S2H-generalizing tasks

In the main experiments, we have trained on each non S2H-generalizing task separately. In this section, we explore the ablation where we combine and randomly shuffle the training data for *Table Readout*, *Grid Navigation*, and *Visual Analogy*. In Figure 26, we compare the image S2H generalization performance when jointly training on all 3 tasks against training on each task separately.

Similar to training on each task individually, the average S2H generalization on image across all 3 tasks is strongest for **Image**-**via**-**Text+**, followed by **Align**-**Mix+** and **Mix+**. When analyzing the effect of multi-task training on each task, we observe that it benefits the model's performance on *Table Readout* and *Grid Navigation* but hurts performance on *Visual Analogy*. This is likely because *Table Readout* and *Grid Navigation* are similar in nature. They are both represented by LaTeX code in the text modality, require the model to identify the current location in a table / grid, and reason about neighboring cells. On the other hand, the skills required for *Visual Analogy* are quite distinct. This suggests that the interactions between tasks during a multi-task training can also affect how much reasoning can transfer across modalities.

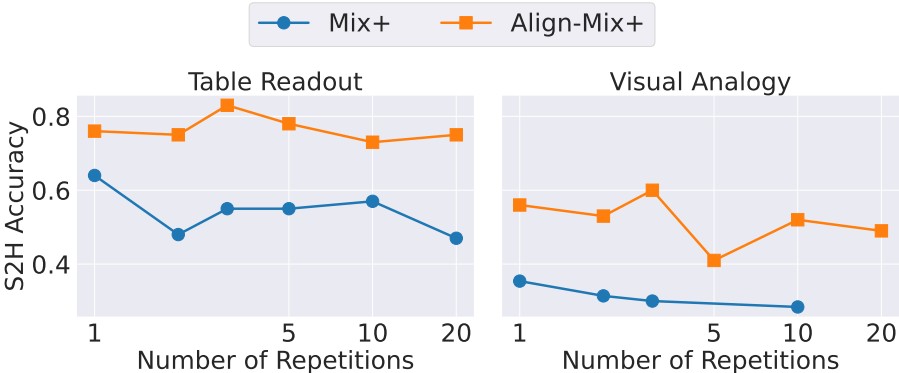

*Figure 27.* **Ablation on the number of repetitions of unique HARD examples, while maintaining the total amount of HARD training data, on *Table Readout* and *Visual Analogy*:** Image S2H generalization degrades with more repetitions of HARD **Text** examples, with the effect on **Mix+** being more drastic. Here, the amount of training data is fixed at $12 \times 10^4$, with $6 \times 10^4$ examples sampled from the HARD task. Interestingly, performance of **Align-Mix+** peaks at $3\times$ repetitions, implying the number of unique HARD **Text** examples can be reduced by $3\times$ for **Align-Mix+**.

### I.9. Ablation on repeated HARD examples

In the experiments reported in the main paper (summarized in Figure 5), we kept all HARD **Text** examples unique. In Figure 27, we present the ablation where we repeat each HARD **Text** example during training, while keeping the total number of training data fixed. Our primary observations are:

- Repeating HARD **Text** examples harms the performance of **Mix+**. Halving the number of unique HARD **Text** examples and repeating each example 2 times can drop the performance on HARD-image by at least $10\%$p on *Table Readout*.

- On the other hand, **Align-Mix+** is quite robust to repetitions on *Table Readout*. The number of unique HARD **Text** examples can be reduced by $10\times$ (and repeating each example $10\times$) with the performance on HARD-image dropping by no more than $1$-$2\%$p.

- On *Visual Analogy*, while the performance of **Align-Mix+** drops with large number of repetitions, the drop in performance is within $1$-$2\%$p if the number of repetitions is up to 3.

- Interestingly, the image S2H generalization performance reaches its peak at exactly 3 repetitions for **Align-Mix+** on both *Table Readout* and *Visual Analogy*. This suggests that we may only require $3\times$ less unique HARD **Text** examples than reported in Figure 5.

# J. Interpretability Experiments

We use gradient attribution to identify which pixel in the image is important when generating each token in the CoT. For a given data $\mathbf{x} \in \mathcal{X}$ and its corresponding image format $\mathbf{x}^{(i)}$, we label the set of pixels in the image as $\{\mathbf{x}_j^{(i)}\}$. For a given gold output $\mathbf{y} = \{CoT(\mathbf{x}), f(\mathbf{x})\}$, we label the sequence of CoT tokens as $\{y_k\}$, where $\mathbf{y}_{:k}$ refers to the subsequence of the CoT tokens, up to the $k$-th token.

For each pixel $\mathbf{x}_j^{(i)} \in \mathbf{x}^{(i)}$ and each CoT token $y_k$, we compute the attribution score as:

$$\text{Pixel Attribute Score:} \qquad \left\langle \nabla_{\mathbf{x}_j^{(i)}} l(f_\theta(\{\mathbf{x}^{(i)}, \mathbf{y}_{:k}\}), y_k), \quad \mathbf{x}_j^{(i)} \right\rangle$$

Informally, on **Image** examples, we take the gradient of the loss of the model's output (up to the $k$-th CoT token) with respect to each pixel, and project on the pixel values. Pixels that show positive alignment with the gradients are marked important for the model's prediction for the token $y_k$.

In Figure 28, we plot the pixel attribute values, averaged across tokens that correspond to different segments of a highlighted path of an example image from *Table Readout*. We observe that **Align-Mix+** improves over **Mix+** models by having more focused and concise pixel attributes around the path of highlighted cells and their corresponding row/column names. In Figure 29, we also show pixel attribute scores on *Visual Analogy*, where the pixel attributes are more aligned with objects of interest scattered around the grid.

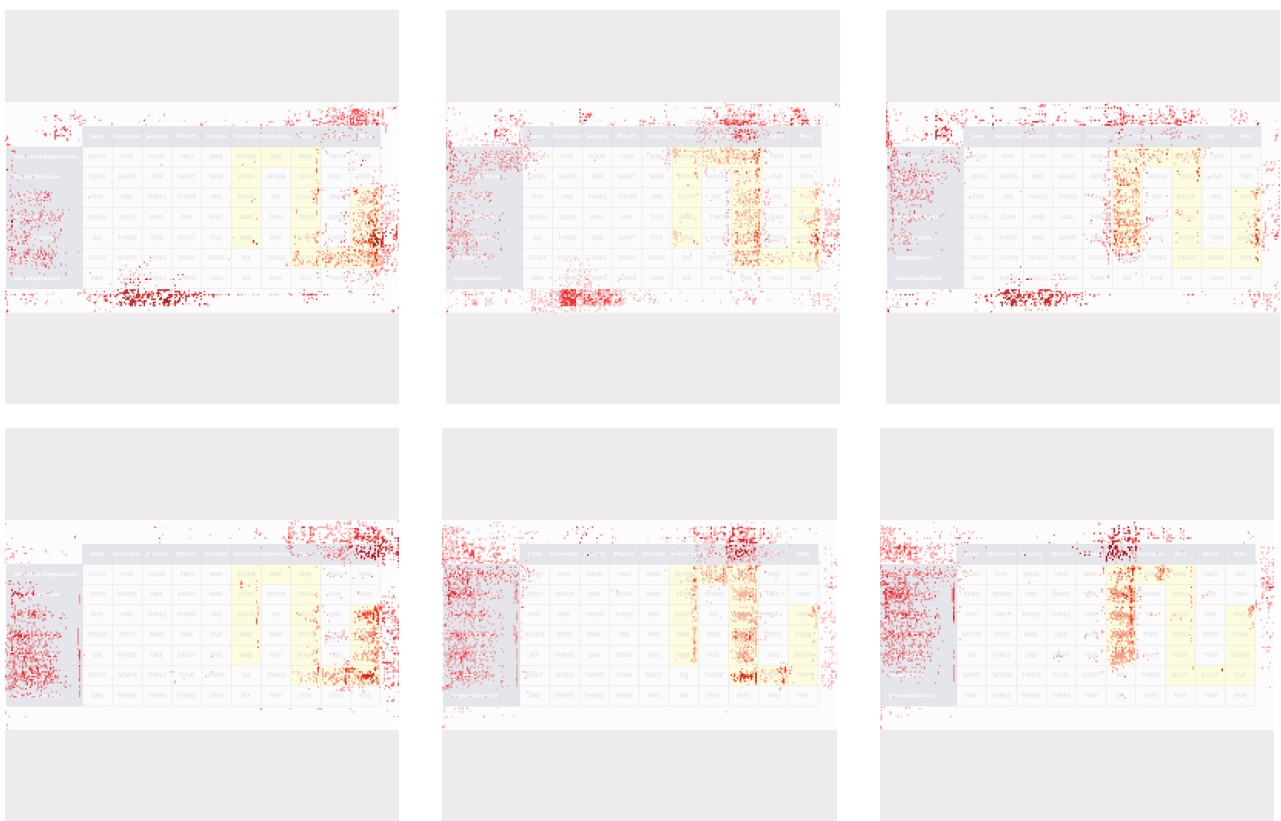

*Figure 28.* **Visualization of pixel attribute scores on *Table Readout*:** (Top) **Mix+**; (Bottom) **Align-Mix+**. Models are trained with $24 \times 10^4$ training data. Pixel attribute scores are averaged across CoT tokens that belong to the first 5 pixels roughly in the 10th column (left), the next 6 cells in the 8th column (middle), and the last 6 cells in the 6-th column (right). We show the top-1% pixels with the highest pixel attribution scores (marked as red). **Mix+** has more diffused pixel attributions in the image, while **Align-Mix+** focuses more on the path of cells (and their corresponding row/column names).

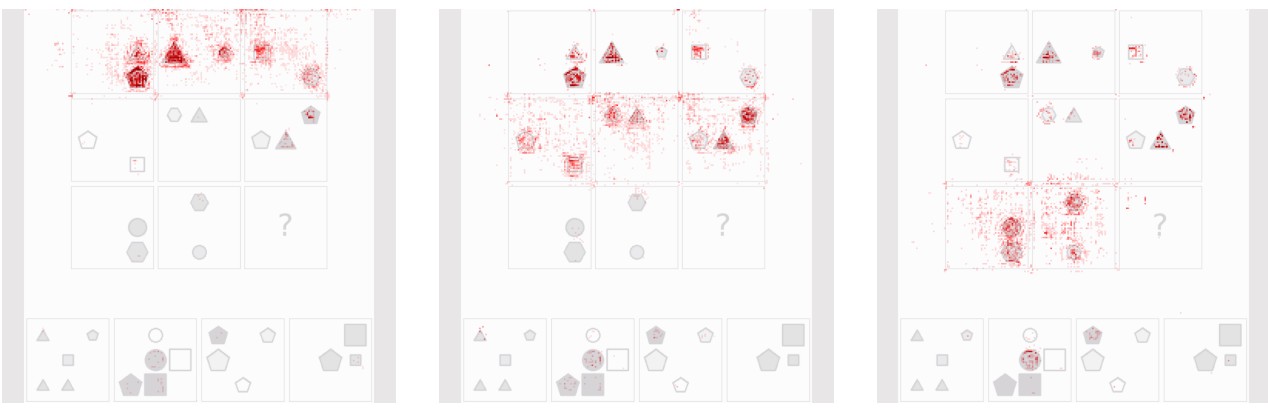

*Figure 29.* **Visualization of pixel attribute scores on *Visual Analogy*:** The model is trained with $12 \times 10^4$ training data of (**TW**) **Align-Mix+**. Pixel attribute scores are averaged across CoT tokens that belong to Example 1 (left), Example 2 (middle), and the query (right) respectively. We show the top-1% pixels with the highest pixel attribution scores (marked as red). The pixel attributes are focused on relevant objects across the grid. Interestingly, when reading relevant object attributes in Example 2, the model still attends to objects from Example 1.

# K. Analysis of Failure Modes

In this section, we briefly discuss the common failure modes of models trained on our synthetic data, when evaluated on examples from the HARD split.

## K.1. *Table Readout*

We analyze the outputs of **Text** on HARD-text, **Image** on HARD-image, and **Mix+** on both HARD-text and HARD-image, where all models have been trained on $24 \times 10^4$ examples.

Since the models perform almost perfectly on the SIMPLE examples, where the total length of the sequence is around 12, one may expect the models to read off the first 12 numbers from tables of the HARD split equivalently well but start making errors after the sequence length it was trained on. We find that this is not the case by analyzing the index of the first error; i.e., how many numbers the model reads off correctly before making the first mistake. Although the average index of the first error is around 14.7, about 56% of incorrect generations (equivalently, 26% of total generations) contain a mistake before the 12th number in the sequence.

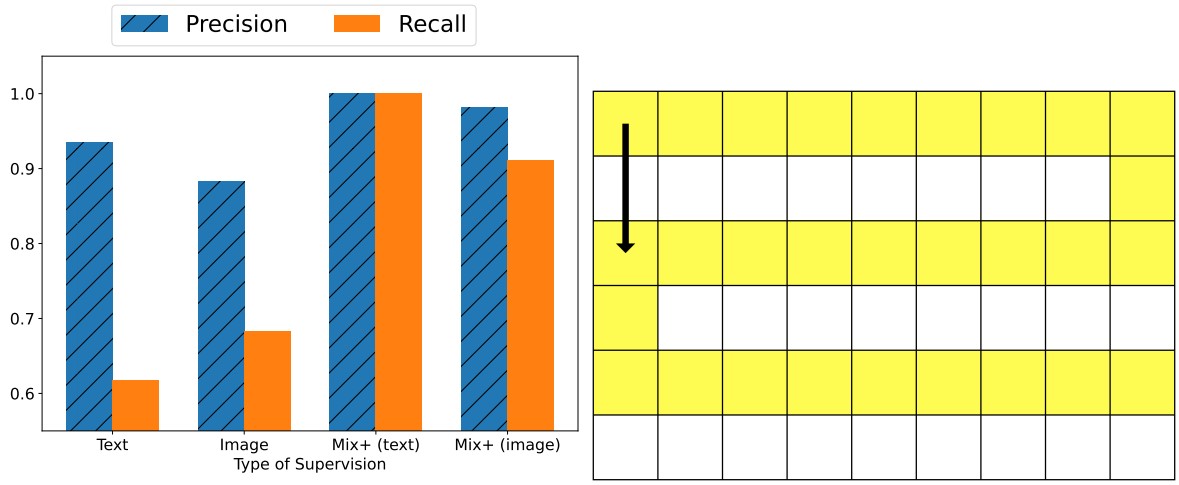

*Figure 30.* **Analysis of failure modes on *Table Readout*:** (Left) Precision and Recall; (Right) Example of a common mistake. Models are trained on $24 \times 10^4$ examples of **Text**, **Image** and **Mix+** supervision and evaluated on corresponding inputs from HARD. Models often hallucinate a "shortcut." In this case, precision would be 12/13 and recall would be 12/29.

To further analyze the behavior of the model when it makes a mistake, we extend the definition of precision and recall:

$$\text{Precision} = \frac{\text{Total \# correctly listed}}{\text{Total \# listed}} \qquad \text{Recall} = \frac{\text{Total \# correctly listed}}{\text{Total \# highlighted}}$$

where we take the sum in the numerator and denominator across all test examples and mark a cell as correctly listed only if the model generation contains it, regardless of the exact position in the sequence. See left of Figure 30 for the evaluation results. Note that for **Text** and **Image**, precision is significantly higher than recall, meaning that it rarely hallucinates that a cell is highlighted (when it is not), but it fails to list off many of the numbers that were highlighted. We find that this is mainly because once the model derails from the highlighted path, it just moves directly towards the destination cell, until it rejoins the path, unintentionally creating a "shortcut" that skips around 15 cells on the original path on average. See right of Figure 30 for a visualization. However, the recall improves significantly on both HARD-text and HARD-image when trained with **Mix+**.

## K.2. *Grid Navigation*

In Figure 31, we analyze the outputs of **Text** supervision on HARD-text, **Image** supervision on HARD-image, and **Mix+** supervision on HARD-image, where all models have been trained on a varying number of examples.

A successful evaluation on *Grid Navigation* requires completing multiple intermediate subtasks. The model first needs to correctly identify the source and destination cells from the grid and parse the row/column indices. We observe that the models can easily learn this subtask. Under any of the three types of supervision, the model can get at least 98% accuracy

on parsing the location of the source and destination cells with only $1.5 \times 10^4$ examples. With $6 \times 10^4$ or more examples, the accuracy is always $100\%$.

Next, we analyze whether the model returns a sequence of actions that leads from the source to the destination (ignoring any object or obstacle). We observe that there is some "phase transition" at $3 \times 10^4$ examples, where the model's accuracy on this subtask increases sharply. However, whereas **Mix+** continues to improve accuracy on this subtask, exceeding $90\%$ at $6 \times 10^4$ examples, **Text** and **Image** supervision fail to achieve $90\%$ even with $24 \times 10^4$ examples.

We then analyze the average fraction of objects collected while navigating the grid. The evaluation on this subtask also follows a similar "phase transition" at $3 \times 10^4$ examples. However, whereas **Mix+** immediately achieves $90\%$ at $3 \times 10^4$ examples and continues to improve to $96\%$ at $24 \times 10^4$ examples, **Text** and **Image** supervision fail to improve beyond $50\text{-}70\%$. This subtask becomes a strong bottleneck for **Text** and **Image** supervision which prevents them from improving S2H generalization performance.

Finally, we analyze the average number of obstacles that the model passes through. Across any of the three types of supervision, the metric improves with more training data. However, this metric drops as low as $0.12$ for **Mix+** at $24 \times 10^4$ examples, whereas **Text** supervision only achieves $0.78$ and **Image** supervision achieves $0.67$.

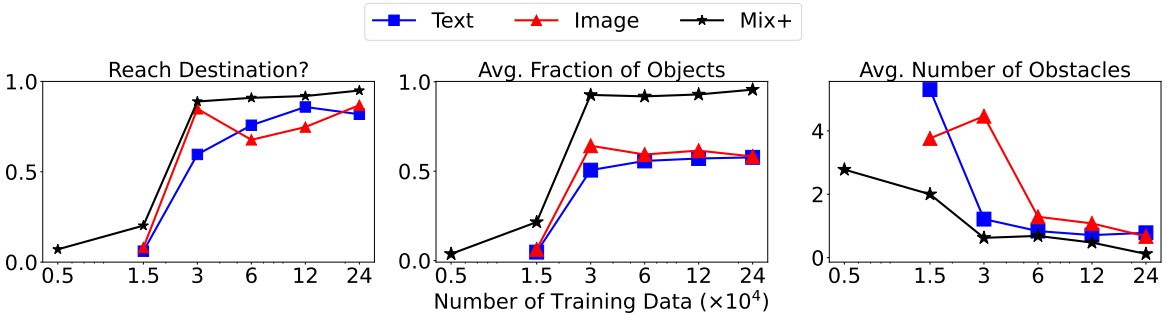

*Figure 31.* **Analysis of failure modes on *Grid Navigation*:** (Left) Whether model generates a sequence of actions that leads to the destination; (Middle) Average fraction of objects collected; (Right) Average number of obstacles passed through. Models trained with **Text** and **Image** fail to improve beyond a certain threshold for all three subtasks.

### K.3. *Visual Analogy*

We analyze the outputs of **Text** on HARD-text, **Image** on HARD-image, and **Mix+** on both HARD-text and HARD-image, where all models have been trained on $12 \times 10^4$ examples. Specifically, we analyze the CoT trace, focusing on the following structural steps as introduced in Appendix E.3 earlier:

- To reason about examples:
  1. given an attribute (e.g. `shape_type`), the model first needs to correctly enumerate the attribute values (e.g. `circle`) for each image in the examples;
  2. the model then needs to decide whether the values in all three images of that example are consistent with a logical relation (e.g. `XOR`);
  3. after repeating the process for both in-context examples, the model summarizes the two relational patterns $(d_1, r_1)$ and $(d_2, r_2)$ for the examples;
  4. finally, the model needs to identify the target relation $r_1 = r_2 = r_{\text{query}}$ from the examples.

- To reason about the query: the model needs to correctly enumerate the attribute values for each image in the query similarly.

- To reason about the options:
  1. assuming the query when combined with each option follows a relational pattern (domain $d$, relation $r$) (e.g. (`line type`, `XOR`), the model needs to identify the correct values of the attribute domain $d$ for each option image and the correct relation $r$;
  2. the model also needs to reason whether the identified relation $r$ is the desired target relation $r_{\text{query}}$.

*Table 14.* **Analysis of failure modes on *Visual Analogy*:** Models are trained on $12 \times 10^4$ examples of Text, Image, and Mix+ supervision and evaluated on corresponding HARD inputs. $^*$ means the evaluation is considered in-domain, as Mix+ supervision contains HARD-text examples in training. To evaluate the entire CoT (second to last row), we check if the generated output contains all the correct values in reasoning steps about the examples, query, and options. The main sources of errors for each type of supervision are highlighted.

| Types of failures | | Error rate (%) | | |
|---|---|---|---|---|
| | | Text (text) | Image (image) | Mix+ (text$^*$ / image) |
| Reasoning about examples | type values | 0.0 | 0.0 | 0.0 / 0.0 |
| | color values | 0.0 | 0.0 | 0.0 / 0.0 |
| | size values | 29.6 | 26.6 | 0.0 / 39 |
| | quantity values | 29.6 | 26.2 | 0.0 / 37.4 |
| | position values | 29.6 | 26.2 | 0.0 / 37.4 |
| | held-out $(d_i, r_i)$ | **86.8** | **81.0** | 0.0 / 42.2 |
| | $d_1 \neq d_2$ | **86.8** | **80.8** | 0.0 / 23.8 |
| | relation | 35.2 | 34.4 | 0.0 / 0.8 |
| Reasoning about query | type values | 0.0 | 0.0 | 0.0 / 0.0 |
| | color values | 0.0 | 0.0 | 0.0 / 0.0 |
| | size values | 0.4 | 7.6 | 0.0 / 16.8 |
| | quantity values | 0.4 | 7.6 | 0.0 / 16.2 |
| | position values | 0.4 | 7.6 | 0.0 / 16.2 |
| Reasoning about options | attribute domain | 79.8 | 65.2 | 0.4 / 44.2 |
| | attribute values | 8.4 | 32.0 | 0.0 / **65.0** |
| | relation | 71.4 | **82.4** | 0.2 / 45.0 |
| | identify solution | 45.2 | 51.8 | 0.2 / 21.0 |
| CoT | | 100.0 | 100.0 | 0.4 / 79.8 |
| Exact match | | 100.0 | 100.0 | 0.4 / 64.6 |

**Mix+ supervision enables significant improvement on reasoning steps that require compositional generalization where Text and Image supervision fail:** As shown in Table 14, we observe that models trained with Text and Image struggle primarily to identify the correct held-out relational pattern $(d_i, r_i)$ for in-context examples, and in particular to recognize $d_1 \neq d_2$, that is, the two examples vary along *different* attributes, with both error rates $\geq 80\%$. These two sources of error correspond exactly to the differences between the SIMPLE and HARD split of *Visual Analogy*, which requires the model to generalize in a compositional manner. With Mix+ supervision, the model significantly improves on these steps with a much smaller error rate of $42.2\%$ in identifying the held-out $(d_i, r_i)$ and $23.8\%$ in recognizing $d_1 \neq d_2$.

***Visual Analogy* focuses more on abstract relational reasoning rather than object detection:** We observe that even with a consistently higher error rate in identifying attribute values, models with Mix+ supervision can achieve a lower error rate in both CoT and exact match compared to their counterparts with Text and Image supervision. This makes sense since reasoning depends more on identifying the correct *logical relation* than on identifying the correct *attribute values*. Although achieving the latter can be an important reasoning step, it is not a necessary condition to arrive at the correct solution.

We also note that the error rate of CoT can be higher than the error rate in exact match. This indicates that in some cases the model can still arrive at the correct solution even though it makes slight mistakes in the reasoning trace: for example, it can still conclude with the correct relational pattern without identifying all the attribute values correctly.

**Even with Mix+ supervision, the model still exhibits sensitivity to CoT templates and hallucinations:** Interestingly, we find that the error rate in identifying values of `size`, `quantity`, and `position` consistently similar. Upon manual inspection of the model output, we find that models fail to switch between different reasoning templates about shape and line objects: while the general templates for the two object types are similar, the model needs to reason about five attributes for shapes and only `type` and `color` for lines. With Mix+ supervision, models can still be sensitive to this small difference in CoT templates and hallucinate about undefined `size`, `quantity`, and `position` attributes of the line objects. This highlights that models with Mix+ supervision are still brittle.

**Question**

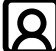

The image shows the sales data of different products in different months. For readability, values in the cells have been replaced by their English names, e.g. 0 is represented by ZERO. There is a path of highlighted cells (colored in yellow) starting from (RadiateTool, January) and ending at (EmberGadget, August). Please return the total of all values in the highlighted cells. Provide the final answer as 'Answer: [total]'.

| | May | November | January | December | October | March | June | February | August | April |
|---|---|---|---|---|---|---|---|---|---|---|
| **RadiateTool** | ONE | FOUR | TWO | SEVEN | NINE | NINE | NINE | SIX | EIGHT | EIGHT |
| **OmicronInstrument** | ONE | ONE | FOUR | SIX | NINE | ZERO | FIVE | ZERO | FOUR | ONE |
| **ShimmerApparatus** | ZERO | TWO | FIVE | ONE | FOUR | EIGHT | FOUR | SIX | NINE | SIX |
| **XiTool** | ONE | ONE | NINE | FOUR | ZERO | ZERO | ONE | NINE | SEVEN | ONE |
| **EmberGadget** | SEVEN | FOUR | SIX | SEVEN | SEVEN | EIGHT | NINE | THREE | THREE | SIX |
| **SizzleContraption** | ONE | SEVEN | ZERO | THREE | FIVE | FIVE | SIX | EIGHT | ONE | THREE |
| **ParamountInstrument** | THREE | EIGHT | SIX | ZERO | NINE | EIGHT | TWO | THREE | EIGHT | NINE |
| **SuperiorGadget** | ZERO | FIVE | TWO | TWO | EIGHT | FIVE | NINE | FIVE | FOUR | SIX |
| **QuantumApparatus** | THREE | TWO | TWO | SIX | THREE | EIGHT | EIGHT | NINE | NINE | FOUR |

**1. Text Conversion**

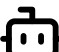

Convert the image into a latex code of the given table, where highlighted cells (colored in yellow) are marked with a * symbol next to the values.

```
\begin{tabular}{lrrlllllllr}
 & May & November & January & December & October & March & June & February
& August & April \\
 RadiateTool & 1 & 4 & 2 * & 7 & 9 & 9 & 9 & 6 & 8 & 8 \\
 OmicronInstrument & 1 & 1 & 4 * & 6 & 9 & 0 & 5 & 0 & 4 & 1 \\
 ShimmerApparatus & 0 & 2 & 5 * & 1 & 4 & 8 & 4 & 6 & 9 & 6 \\
 XiTool & 1 & 1 & 9 * & 4 & 0 & 0 & 1 & 9 & 7 & 1 \\
 EmberGadget & 7 & 4 & 6 * & 7 & 7 & 8 & 9 & 3 & 3 * & 6 \\
 SizzleContraption & 1 & 7 & 0 * & 3 & 5 & 5 & 6 & 8 & 1 * & 3 \\
 ParamountInstrument & 3 & 8 & 6 * & 0 & 9 & 8 & 2 & 3 & 8 * & 9 \\
 SuperiorGadget & 0 & 5 & 2 * & 2 & 8 & 5 & 9 & 5 & 4 * & 6 \\
 QuantumApparatus & 3 & 2 & 2 * & 6 * & 3 * & 8 * & 8 * & 9 * & 9 * & 4 \\
 \end{tabular}
\caption{Business Sales Data for products in different months}
\label{tab:business_sales}
\end{table}
```

**2. CoT**

We enumerate the relevant row indices, column indices, row names, column names, and their corresponding values.

Row Index, Column Index, Row Name, Column Name, Value

| 1 | 3 | RadiateTool | January | 2 |
|---|---|---|---|---|
| 2 | 3 | OmnicronInstrument | January | 4 |
| 3 | 3 | ShimmerApparatus | January | 5 |
| 4 | 3 | XiTool | December | 9 |
| 5 | 3 | EmberGadget | January | 6 |
| …… | | | | |
| 9 | 9 | QuantumApparatus | August | 9 |
| 8 | 9 | SuperiorGadget | August | 4 |
| 7 | 9 | ParamountInstrument | August | 8 |
| 6 | 9 | SizzleContraption | August | 1 |
| 5 | 9 | EmberGadget | August | 3 |

**3. Prediction**

The total is $2 + 4 + 5 + 9 + 6 + 0 + 6 + 2 + 2 + 6 + 3 + 8 + 8 + 9 + 9 + 4 + 8 + 1 + 3 = 95$.

Answer: 95

*Figure 32.* **A SIMPLE example from** *Table Readout*.

*Question*

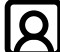

The image shows the sales data of different products in different months. For readability, values in the cells have been replaced by their English names, e.g. 0 is represented by ZERO. There is a path of highlighted cells (colored in yellow) starting from (PsiDevice, September) and ending at (DeltaTool, June). Please return the total of all values in the highlighted cells. Provide the final answer as 'Answer: [total]'.

| | September | December | July | October | February | November | June | May | January | March |
|---|---|---|---|---|---|---|---|---|---|---|
| ShineDevice | ONE | FOUR | ONE | SEVEN | FOUR | FIVE | ZERO | SIX | TWO | NINE |
| WaveTool | SEVEN | ZERO | EIGHT | ZERO | FIVE | THREE | ZERO | FOUR | THREE | THREE |
| FlashTool | THREE | SIX | FOUR | SEVEN | TWO | FOUR | FOUR | TWO | FOUR | ONE |
| ScorchMachine | SIX | THREE | NINE | ZERO | FIVE | TWO | ONE | ZERO | SIX | EIGHT |
| EliteMechanism | NINE | SIX | ONE | EIGHT | ZERO | EIGHT | FOUR | TWO | TWO | ONE |
| EpsilonInstrument | EIGHT | TWO | ZERO | ONE | ONE | ONE | THREE | SIX | FOUR | SIX |
| DeltaTool | EIGHT | ONE | NINE | THREE | FIVE | ZERO | ONE | ONE | ONE | NINE |
| ApexWidget | FIVE | ONE | ONE | EIGHT | SEVEN | THREE | THREE | NINE | ZERO | TWO |
| EmberGadget | ZERO | TWO | EIGHT | FIVE | THREE | TWO | FIVE | THREE | SEVEN | THREE |
| PsiDevice | FOUR | NINE | FOUR | ZERO | SIX | FOUR | SEVEN | FIVE | ZERO | SEVEN |

*1. Text Conversion*

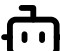

Convert the image into a latex code of the given table, where highlighted cells (colored in yellow) are marked with a * symbol next to the values.

```
\begin{table}[ht]
\centering
\begin{tabular}{lllllllllll}
 & September & December & July & October & February & November & June & May
& January & March \\
 ShineDevice & 1 * & 4 * & 1 * & 7 * & 4 * & 5 * & 0 * & 6 * & 2 * & 9 * \\
 WaveTool & 7 * & 0 & 8 & 0 & 5 & 3 & 0 & 4 & 3 & 3 * \\
 FlashTool & 3 * & 6 & 4 * & 7 * & 2 * & 4 * & 4 * & 2 * & 4 * & 1 * \\
 ScorchMachine & 2 * & 3 & 9 * & 0 & 5 & 2 & 1 & 0 & 6 & 8 \\
 EliteMechanism & 9 * & 6 & 1 * & 1 & 0 * & 8 * & 4 * & 2 * & 2 * & 1 * \\
 EpsilonInstrument & 8 * & 2 & 0 * & 1 & 1 * & 1 * & 1 & 3 & 6 & 4 & 6 * \\
 DeltaTool & 8 * & 1 & 9 * & 3 & 5 * & 0 & 1 * & 1 * & 6 & 9 * \\
 ApexWidget & 5 * & 1 & 1 * & 8 & 7 * & 3 & 3 & 9 * & 0 & 2 * \\
 EmberGadget & 0 * & 2 & 8 * & 5 & 3 * & 2 & 5 & 3 * & 7 & 3 * \\
 PsiDevice & 4 * & 9 & 4 * & 0 * & 6 * & 4 & 7 & 5 * & 0 * & 7 * \\
 \end{tabular}
\caption{Business Sales Data for products in different months}
\label{tab:business_sales}
\end{table}
```

*2. CoT*

We enumerate the relevant row indices, column indices, row names, column names, and their corresponding values.

Row Index, Column Index, Row Name, Column Name, Value

| | | | | |
|---|---|---|---|---|
| 10 | 1 | PsiDevice | September | 4 |
| 9 | 1 | EmberGadget | September | 0 |
| 8 | 1 | ApexWidget | September | 5 |
| 7 | 1 | DeltaTool | September | 8 |
| 6 | 1 | EpsilonInstrument | September | 8 |
| ...... | | | | |
| 10 | 8 | PsiDevice | May | 5 |
| 9 | 8 | EmberGadget | May | 3 |
| 8 | 8 | ApexWidget | May | 9 |
| 7 | 8 | DeltaTool | May | 1 |
| 7 | 7 | DeltaTool | June | 1 |

*3. Prediction*

The total is $4 + 0 + 5 + 8 + 8 + 9 + 2 + 3 + 7 + 1 + 4 + 1 + 7 + 4 + 5 + 0 + 6 + 2 + 9 + 3 + 1 + 4 + 2 + 4 + 4 + 2 + 7 + 4 + 9 + 1 + 0 + 9 + 1 + 8 + 4 + 0 + 6 + 3 + 7 + 5 + 1 + 0 + 8 + 4 + 2 + 2 + 1 + 6 + 9 + 2 + 3 + 7 + 0 + 5 + 3 + 9 + 1 + 1 = 233$.

Answer: 233

*Figure 33.* A **HARD** example from *Table Readout*.

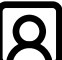

*Question*

The image shows a 2 dimensional grid. The grid is filled up with objects, which you will be asked to recognize and collect, and obstacles, which you should avoid. Possible objects and obstacles are given as follows:

Objects: heart, crown, flag, star, flower, umbrella, plane, phone, spark, diamond, queen, hammer, club, gear, arrow, sun, bishop, note, coffee, anchor, cloud, pawn, castle, horse, infinity, moon, null, approx, integral, product, sum

Obstacles: dot, cross, square, triangle, plus

Cells that don't contain any object or obstacle are left blank. A traveller starts at the source (colored in blue) and wishes to go to the destination (colored in green). Your task is to give a list of actions ( left right up down) each indicating a movement by 1 cell that the traveller needs to follow. The traveller must recognize all objects in the grid, collect them but avoid obstacles in the process. Provide the final answer as 'Answer: [list of actions]'. If no path exists, output 'Answer: No path exists'.

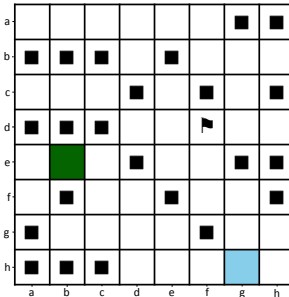

*1. Text Conversion*

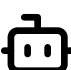

Convert the image into a text version of a table, where cells colored in blue, green are respectively marked as S, D. Blank cells are given by O. Objects and obstacles are mentioned by their names.

& a & b & c & d & e & f & g & h
a & O & O & O & O & O & O & square & square
b & square & square & square & O & square & O & O & O
c & O & O & O & square & O & square & O & square
d & square & square & square & O & O & flag & O & O
e & O & D & O & square & O & O & square & square
f & O & square & O & O & square & O & O & square
g & square & O & O & O & O & square & O & O
h & square & square & square & O & O & O & S & O

*3. Prediction*

Answer: up up left up up left down right down right down down left left up left up left up left

*2. CoT*

We follow a depth first search, always moving towards the destination.
Source:( h, g) Destination:( e, b)
Collect objects: flag( d, f)
Avoid square
h, g:
 up: g, g okay
g, g:
 up: f, g okay
f, g:
 up: e, g closed( square)
 left: f, f okay
f, f:
 up: e, f okay
e, f:

up: d, f okay
d, f: Collect flag
left: d, e okay
d, e:
left: d, d okay
d, d:
left: d, c closed( square)
down: e, d closed( square)
up: c, d closed( square)
right: d, e visited
No available actions.
retrace 1 step
d, e:
left: d, d visited
down: e, e okay
e, e:

left: e, d closed( square)
up: d, e visited
down: f, e closed( square)
right: e, f okay
e, f:
left: e, e visited
up: d, f visited
down: f, f okay
f, f:
left: f, e closed( square)
up: e, f visited
down: g, f closed( square)
right: f, g okay
f, g:
left: f, f visited
up: e, g closed( square)

down: g, g okay
g, g:
left: g, f closed( square)
up: f, g visited
down: h, g okay
h, g:
g, e:
left: g, d okay
g, d:
up: f, d okay
f, d:
left: f, c okay
f, c:
up: e, c okay
e, c:
left: e, b okay

*Figure 34.* A SIMPLE **example from** *Grid Navigation*.

## Question

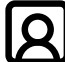

The image shows a 2 dimensional grid. The grid is filled up with objects, which you will be asked to recognize and collect, and obstacles, which you should avoid. Possible objects and obstacles are given as follows:

Objects: heart, crown, flag, star, flower, umbrella, plane, phone, spark, diamond, queen, hammer, club, gear, arrow, sun, bishop, note, coffee, anchor, cloud, pawn, castle, horse, infinity, moon, null, approx, integral, product, sum

Obstacles: dot, cross, square, triangle, plus

Cells that don't contain any object or obstacle are left blank. A traveller starts at the source (colored in blue) and wishes to go to the destination (colored in green). Your task is to give a list of actions ( left right up down) each indicating a movement by 1 cell that the traveller needs to follow. The traveller must recognize all objects in the grid, collect them but avoid obstacles in the process. Provide the final answer as 'Answer: [list of actions]'. If no path exists, output 'Answer: No path exists'.

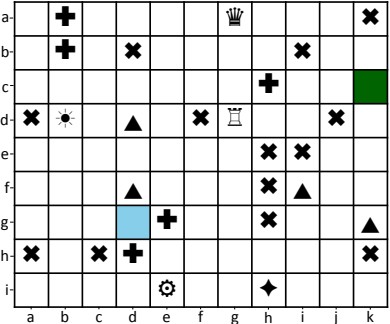

## 1. Text Conversion

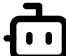

Convert the image into a text version of a table, where cells colored in blue, green are respectively marked as S, D. Blank cells are given by O. Objects and obstacles are mentioned by their names.

& a & b & c & d & e & f & g & h & i & j & k
a & O & plus & O & O & O & O & queen & O & O & O & cross
b & O & plus & O & cross & O & O & O & O & cross & O & O
c & O & O & O & O & O & O & O & plus & O & O & D
d & cross & sun & O & triangle & O & O & cross & castle & O & O & cross & O
e & O & O & O & O & O & O & O & cross & cross & O & O
f & O & O & O & triangle & O & O & O & cross & triangle & O & O
g & O & O & O & S & plus & O & O & O & cross & O & O & triangle
h & cross & O & cross & plus & O & O & O & O & O & O & cross
i & O & O & O & O & gear & O & O & spark & O & O & O

## 3. Prediction

Answer: left up left down down down right right right right right right up left up up up up up up up left left down down left left down left right up right right right right up right up right right down down right

## 2. CoT

We follow a depth first search, always moving towards the destination.
Source:( g, d) Destination:( c, k)
Collect objects: gear( i, e) spark( i, h) castle( d, g) queen( a, g) sun( d, b)
Avoid cross triangle plus
g, d:
down: h, d closed( plus)
right: g, e closed( plus)
left: g, c okay
g, c:
down: h, c closed( cross)
right: g, d visited
up: f, c okay
f, c:
down: g, c visited
right: f, d closed( triangle)
left: f, b okay
f, b:
down: g, b okay
g, b:
right: g, c visited
down: h, b okay
h, b:
right: h, c closed( cross)
down: i, b okay
i, b:
right: i, c okay

i, c:
right: i, d okay
i, d:
right: i, e okay
i, e: Collect gear
right: i, f okay
i, f:
right: i, g okay
i, g:
right: i, h okay
i, h: Collect spark
up: h, h okay
h, h:
up: g, h closed( cross)
left: h, g okay
h, g:
up: g, g okay
g, g:
up: f, g okay
f, g:
up: e, g okay
e, g:
up: d, g okay
d, g: Collect castle
up: c, g okay
c, g:
up: b, g okay
b, g:

up: a, g okay
a, g: Collect queen
left: a, f okay
a, f:
left: a, e okay
a, e:
down: b, e okay
b, e:
left: b, d closed( cross)
down: c, e okay
c, e:
left: c, d okay
c, d:
left: c, c okay
c, c:
down: d, c okay
d, c:
left: d, b okay
d, b: Collect sun
right: d, c okay
d, c:
right: d, d closed( triangle)
up: c, c okay
c, c:
right: c, d okay
c, d:
right: c, e okay
c, e:

right: c, f okay
c, f:
right: c, g okay
c, g:
right: c, h closed( plus)
up: b, g okay
b, g:
right: b, h okay
b, h:
right: b, i closed( cross)
down: c, h closed( plus)
up: a, h okay
a, h:
right: a, i okay
a, i:
down: b, i closed( cross)
right: a, j okay
a, j:
down: b, j okay
b, j:
down: c, j okay
c, j:
right: c, k okay

*Figure 35.* **A HARD example from** *Grid Navigation*.

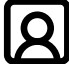

**Question**

The image shows a a puzzle in a 3 by 3 grid followed by 4 options. The puzzle consists of 2 examples (row 1 and 2), a query (row 3), and four options. Each example contains three images following a relation along certain attribute, and this relation is consistent across all examples. The query contains two images. Analyze the changes in the following attributes for each example: line type, line color, shape type, shape color, shape size, shape quantity, shape position, and consider the relations: Progression, XOR, OR, and AND. Progression requires the value of a certain attribute to strictly increase or decrease, but not necessarily by a fixed amount. Please provide your predictions in the format 'Example i: (attribute, relation)' for each example and similarly for options. Provide the final answer as 'Answer: [correct option]'.

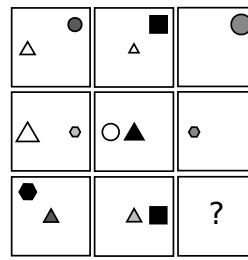

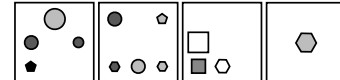

**1. Text Conversion**

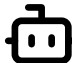

Convert the image into a text version of the puzzle.
Example 1:
Panel 1:
2 shapes; type: circle, … color: 90, 255, size: 27, position: (1, 0), (0, 2)
Panel 2: …
Example 2:
Panel 1:
…
Query:
Panel 1:
…
Options:
Option 1:
4 shapes; type: pentagon
…

**3. Prediction**

Example 1: (shape position, AND)
Example 2: (shape position, AND)
Option 1: (shape color, OR), not consistent
Option 2: (shape color, XOR), not consistent
Option 3: (shape type, XOR), not consistent
Option 4: (shape position, AND), consistent
Answer: 4

**2. CoT**

We first analyze the examples by considering the applicable attributes and relations:
Example 1:
type:
Image 1: circle, triangle
Image 2: triangle, rectangle
Image 3: circle
No pattern.
…
position:
Image 1: (1, 0), (0, 2)
Image 2: (0, 2), (1, 1)
Image 3: (0, 2)
This suggests the AND relation.
Example 2: …
The examples suggest the following patterns: (shape position, AND), (shape position, AND).
We conclude that AND is the common relation.
Now, we analyze the images in the query:
type:
Image 1: hexagon, triangle
Image 2: triangle, rectangle
color: …
Looking at the options, we find:
Option 1 has shape color of 189, 90, 0, so it is consistent with the OR relation on shape color, but OR is not the target relation.
…
Option 4 has shape position of (1, 1), so it is consistent with the AND relation on shape position, and AND is the target relation.
We conclude that option 4 is the answer since it is consistent with the AND relation.

*Figure 36.* **A SIMPLE example from *Visual Analogy*:** The common relation is $r = $ AND and the domains are $d_1 = d_2 = d_{query} = $ shape quantity, and the combinations $(d, r)$ are not in the held-out set $\mathcal{S} = \{$(line type, XOR), (line color, OR), (shape type, AND), (shape size, XOR), (shape color, Progression), (shape position, OR), (line type, AND), (line color, Progression)$\}$.

*Question*

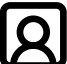 The image shows a a puzzle in a 3 by 3 grid followed by 4 options. The puzzle consists of 2 examples (row 1 and 2), a query (row 3), and four options. Each example contains three images following a relation along certain attribute, and this relation is consistent across all examples. The query contains two images. Analyze the changes in the following attributes for each example: line type, line color, shape type, shape color, shape size, shape quantity, shape position, and consider the relations: Progression, XOR, OR, and AND. Progression requires the value of a certain attribute to strictly increase or decrease, but not necessarily by a fixed amount. Please provide your predictions in the format 'Example i: (attribute, relation)' for each example and similarly for options. Provide the final answer as 'Answer: [correct option]'.

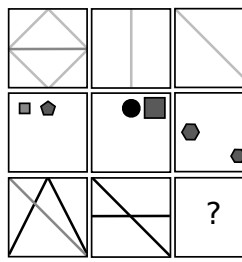

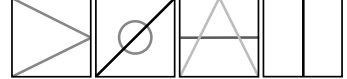

*1. Text Conversion*

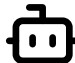 Convert the image into a text version of the puzzle.
Example 1:
Panel 1:
type: diamond lines, horizontal line, color: 189, 135
…
Example 2:
Panel 1:
2 shapes; type: pentagon, …
…
Query:
Panel 1:
…
Options:
Option 1:
type: V-shape facing left, color:
…

*2. CoT*

We first analyze the examples by considering the applicable attributes and relations:
Example 1:
type:
Image 1: diamond lines, horizontal line
Image 2: vertical line
Image 3: falling diagonal line
No pattern.
color:
Image 1: 189, 135
Image 2: 189
Image 3: 189
This suggests the AND relation.
Example 2: …
The examples suggest the following patterns: (line color, AND), (shape color, AND).
We conclude that AND is the common relation.
Now, we analyze the images in the query:
type:
Image 1: falling diagonal line, V-shape facing down
Image 2: horizontal line, falling diagonal line
color: …
Looking at the options, we find:
Option 1 has line color of 135, so it is consistent with the XOR relation on line color, but XOR is not the target relation.
…
Option 4 has line color of 0, so it is consistent with the AND relation on line color, and AND is the target relation.
We conclude that option 4 is the answer since it is consistent with the AND relation.

*3. Prediction*

Example 1: (line color, AND)
Example 2: (shape color, AND)
Option 1: (line color, XOR), not consistent
Option 2: (line color, OR), not consistent
Option 3: (line type, XOR), not consistent
Option 4: (line color, AND), consistent
Answer: 4

*Figure 37.* **A HARD example from *Visual Analogy*:** The common relation is $r = $ AND and the domains are distinct: $d_1 = $ line color, $d_2 = $ shape position, $d_{\text{query}} = $ line color, and the combinations $(d, r)$ are in the held-out set $\mathcal{S} = \{$(line type, XOR), (line color, OR), (shape type, AND), (shape size, XOR), (shape color, Progression), (shape position, OR), (line type, AND), (line color, Progression)$\}$. Note that the pattern for the confounding options may not be in $\mathcal{S}$.

