# OpenReview forum: "Generalizing from SIMPLE to HARD Visual Reasoning: Can We Mitigate Modality Imbalance in VLMs?"
_ICML.cc/2025/Conference — ICML 2025 poster_

### Official Review · Reviewer_NF8o · 2025-03-14

**Overall Recommendation:** 3

**Summary:**

The paper investigates modality imbalance issues in existing VLMs by designing multiple tasks with diverse settings. Comprehensive experiments reveal interesting findings, and gradient analysis further illustrates how different settings impact the final results.

**Claims And Evidence:**

Mostly, but with some caveats. While the experimental results support the idea that modality imbalance exists in VLMs and that Image-via-Text conversion helps alleviate this, the evidence could be more comprehensive. For instance, the paper focuses on synthetic tasks, which are controlled and may not fully represent the complexity and variety of real-world visual reasoning tasks. Also, the performance on the HARD-image tasks, even with the proposed strategies, is still far from perfect. This indicates that while the strategies help, they may not provide a complete solution to the problem, and there might be other underlying issues that are not addressed in the paper.

**Essential References Not Discussed:**

no

**Experimental Designs Or Analyses:**

Sound but with limitations. The experimental designs are reasonable, but there are several points of concern:

- Task limitations: As mentioned earlier, the tasks are quite specific and synthetic. The paper could benefit from including more varied tasks or even applying the methods to existing, large-scale benchmarks like VQA (Visual Question Answering) or other real-world multimodal reasoning benchmarks.

- Modality imbalance: While the paper focuses on mitigating modality imbalance, the results show that the gap between text and image reasoning remains significant, even with the proposed methods. This suggests that the current approaches (Image-via-Text and Mix supervision) may not fully solve the problem. In fact, the model's heavy reliance on image-to-text conversion at inference time indicates that while the training can alleviate the gap, it doesn't eliminate it, which may limit the practical applicability of the approach.

- Inference time cost: One major issue that the paper glosses over is the increased inference time cost due to the image-to-text conversion. While the authors acknowledge this, they do not explore alternative strategies to mitigate this overhead, which could be a significant limitation in real-world applications.

- Gradient alignment study: The paper’s analysis of gradient alignment and its impact on S2H generalization is interesting but lacks depth. While the alignment scores provide some insight into how well the model is learning to generalize across modalities, the paper doesn't go into much detail on how these gradients are computed or how reliable they are as a measure of generalization. Moreover, this analysis could benefit from more ablations or sensitivity analyses to assess the robustness of the observed improvements.

**Methods And Evaluation Criteria:**

Yes, but limited by synthetic tasks. The methods and evaluation criteria (s2h generalization) are reasonable for controlled experiments. However, relying on synthetic tasks like Table Readout and Grid Navigation might limit the generalization of the findings. These tasks are constructed to highlight specific issues (such as reasoning across modalities), but they do not reflect the real-world diversity of tasks VLMs are typically applied to. More diverse, real-world tasks would provide a better test of the proposed methods' effectiveness and generalizability.

**Other Comments Or Suggestions:**

no

**Other Strengths And Weaknesses:**

**Strengths:**

1. **Originality and Innovation**:
   - The paper introduces a novel perspective on mitigating modality imbalance in Vision-Language Models (VLMs), which is a relatively underexplored area. The idea of using simple-to-hard (S2H) generalization and specifically the concept of training with Image-via-Text supervision for improving VLMs’ reasoning capabilities across modalities is an innovative approach. This contribution is significant as it goes beyond just testing VLMs on a standard benchmark; it provides a systematic methodology to address a key issue in VLM performance.
   - The idea of "internalizing" the image-to-text conversion is also quite creative, offering an elegant solution to reduce inference time, even if the initial training strategy is more costly. This speaks to a potential real-world application of reducing the computational overhead in production systems without sacrificing performance.

2. **Experimental Rigor**:
   - The paper presents a comprehensive set of experiments with well-defined tasks, which allow the authors to study the issue of modality imbalance in great detail. The ablation studies, gradient alignment analysis, and task-specific results provide useful insights into how various training strategies impact performance. These experiments help demonstrate the practical effectiveness of the proposed methods.

3. **Clarity**:
   - Overall, the paper is well-structured and presents its methods, experimental design, and results in a clear and logical manner. The methodology is sufficiently detailed for readers to follow the steps and replicate experiments, which is essential for scientific transparency. Figures and diagrams complement the text and are helpful for understanding complex ideas like the training strategies and results.

**Weaknesses:**

1. **Limited Applicability of Synthetic Tasks**:
   - While the synthetic tasks used in the experiments are designed to probe specific aspects of VLM reasoning, they do not fully capture the complexity of real-world multimodal tasks. This limits the external validity of the results. Real-world applications often involve much more varied, noisy, and dynamic data, which may not be sufficiently represented by controlled tasks like Table Readout or Grid Navigation. As a result, the paper could have explored how these methods perform on real-world benchmarks such as VQA.

2. **Inference Time Cost**:
   - The increased inference time due to image-to-text conversion remains a significant concern. While the paper addresses this cost, it doesn't fully explore how this trade-off might impact practical deployments, especially in environments where latency or real-time responses are crucial. A more detailed analysis of how to mitigate this overhead would make the paper more practical.

3. **Lack of Theoretical Depth**:
   - The paper focuses heavily on empirical results, which are certainly valuable. However, there is a lack of deep theoretical exploration behind the proposed methods. For example, while the paper discusses gradient alignment, it doesn’t offer a formal mathematical formulation or sufficient explanation of why this approach works. The relationship between gradient alignment and generalization across modalities could have been made more rigorous.

4. **Modality Imbalance Still Present**:
   - Despite the promising results, the paper acknowledges that a gap remains in generalization performance between text and image inputs, even with the proposed strategies. This indicates that the solution is not fully comprehensive, and the challenge of modality imbalance is far from being solved. A deeper exploration of why some gaps remain and further refinement of the approach would strengthen the paper's claim.

5. **Experimental Results on Large-Scale Tasks**:
   - The experiments rely on relatively smaller synthetic datasets and do not provide results on large-scale, real-world benchmarks. While this helps control for specific variables, the results may not generalize well to larger, more complex datasets. It would be beneficial to see how the proposed strategies scale when applied to more realistic data, where other complexities such as noise, data sparsity, and multimodal interactions come into play.

6. **Insufficient Discussion on Limitations**:
   - While the paper briefly mentions limitations such as inference cost and the restricted scope of the tasks, it doesn't delve deeply into potential shortcomings of the proposed approach. A more balanced discussion of when and where these methods might fail would provide readers with a clearer understanding of the boundaries of the work.

**Questions For Authors:**

1. The paper relies on synthetic tasks like Table Readout and Grid Navigation to study modality imbalance. How do you justify using these tasks, and how well do you think the findings apply to real-world VLM applications? Can we generalize the results beyond these controlled tasks?

2. Even with the strategies you’ve proposed, there’s still a noticeable S2H generalization gap between text and image inputs. What do you see as the remaining challenges in addressing this gap, and are there any components or techniques missing that could help achieve full generalization?

3. You mention that the image-to-text conversion method helps but comes with increased inference time. What are the real-world implications of this, and do you have any suggestions or strategies for minimizing the extra cost during inference, especially in production environments?

4. The gradient alignment study is an interesting part of the paper, but I’m curious about how reliable those alignment scores are as a measure of S2H generalization. Have you tested their stability across different datasets or training conditions, and do you think they are robust enough to be a key indicator of generalization?

5. Lastly, how do you generate the Chain-of-Thought (CoT) data for the tasks you’ve presented? Could you walk us through the process a bit more?

**Relation To Broader Scientific Literature:**

no

**Theoretical Claims:**

No, I did not identify any formal proofs within the text. However, the paper discusses theoretical aspects like gradient alignment and the internalization of image-to-text conversion, which are supported by experimental data and loss function analysis. These are more empirical observations than formal mathematical proofs.

---

> ### Author Rebuttal · Authors · 2025-03-31
>
> # Authors' Response to Reviewer NF8o
> We thank the reviewer for their time and effort to review our paper. Please find responses to your comments and questions below.
>
> # Questions
>
> > Experiments are restricted to Synthetic Tasks
>
> We use synthetic tasks to enable controlled studies of modality imbalance and simple-to-hard generalization in VLMs. These tasks are carefully designed to require diverse skills such as OCR, spatial navigation, and visual reasoning, which are essential to real-world VLM applications. As shown in **Table 7 (Appendix I.1)** (referenced in lines 048–049), frontier VLMs struggle with these tasks, suggesting that even simplified settings can reveal gaps in current models. Importantly, incorporating these synthetic tasks into pre-training improves performance on a broad range of real-world benchmarks, show in **Table 6 (Appendix G.6)** (referenced in lines 427–428). Thus, the synthetic tasks that we use are relevant to the real-world applications and could be utilized to motivate training choices to boost VLM’s general abilities.
>
> > What are the challenges for the remaining modality gap?
>
> Thank you for the interesting question. Our work highlights a modality gap in simple-to-hard (S2H) generalization between text and image inputs, which we attribute in part to current VLM training paradigms. Current VLMs rely on adapter-based architectures that loosely integrate small visual encoders with large pre-trained LLMs, which have been pre-trained separately on image and text pre-training data.
>
> Our approach uses image-to-text conversion to better align image inputs with the model’s strong capabilities in language, partially mitigating this gap. However, fully closing it likely requires rethinking the model design itself. Moving beyond adapter-based setups towards early-fusion architectures or joint multi-modal pretraining could help models learn more unified representations and improve generalization across modalities.
>
> > How do the authors propose to mitigate additional inference cost with image-to-text conversion
>
> We would like to clarify that our proposed approaches, Mix, Mix+, and Align-Mix+ supervision types, use image-to-text conversion only during training the model (please see lines 151-152, 245-247, 256-257). During inference, the models trained with these supervision types **don’t explicitly perform image-to-text conversion**. Please see **Figures 3 and 6** in the main paper and lines 205-209, 257-264 for discussions on lengths of inference-time generations for Mix, Mix+, and Align-Mix+ supervisions.
>
> > Further analysis of gradient alignment for more models and datasets
>
> Thanks for the question! We evaluated gradient alignment on the Consecutive Table Readout and Table Readout datasets. Due to the high cost of saving frequent checkpoints, we did not extend these studies to other settings. Instead, we focused on fine-grained alignment measures (see Figures 17–20) and provided theoretical support in Theorems H.1 and H.2. Together, these offer strong initial evidence for the robustness of our alignment metric as a generalization indicator.
>
> > More details on CoT Generation
>
> All of our data (image, its text description, the CoT trace, and the solution) is generated with a Python script. When we generate an image (e.g., using matplotlib), we store relevant metadata (e.g., sequence of (row index, col index) of highlighted cells in Table Readout) and insert it in a fixed chat template. We give examples of our CoT data in **Figures 32-37** in the appendix. See Appendix D.6.2 (mentioned in footnote 1 of page 1) for alternate versions of CoT templates.

---

### Official Review · Reviewer_d6eT · 2025-03-15

**Overall Recommendation:** 1

**Summary:**

This paper studies visual reasoning using vision-language models (VLMs). The authors focus on three tasks: Table Readout, Grid Navigation, and Visual Analogy. They run experiments under simple to hard settings to test each task's generalization.
They propose distilling knowledge from large language models (LLMs) by converting images into text and extracting chain-of-thought reasoning. Their findings suggest that (1) converting images to text can improve visual reasoning and (2) applying this form of knowledge distillation enables the VLM to learn the transferred chain of thought directly.

**Claims And Evidence:**

No, the paper is hard to read. It lacks clearly defined sections for the $\textit{method, dataset, and experiments}$. Instead, these sections are mixed together, making it difficult to identify the evidence supporting their claims.

**Essential References Not Discussed:**

Multimodal Chain-of-Thought Reasoning in Language Models

**Ethical Review Concerns:**

The paper does not have ethical concerns

**Experimental Designs Or Analyses:**

The paper mixes experimental designs and methods, which makes it hard to follow the claims. For example, Section 3 should focus on the methods, but it gets interrupted by experiments (see Figure 2), so it's not clear how the methods lead to the conclusions.

**Methods And Evaluation Criteria:**

The datasets in this paper are self-created and not publicly available, and they lack clear descriptions. As a result, it is hard to assess the quality and difficulty of the datasets for Table Readout, Grid Navigation, and Visual Analogy. It would be better to propose a new dataset if none exists or to use existing benchmarks.

**Other Comments Or Suggestions:**

1. The paper does not separate methods, datasets, and experiments into clear sections. Instead, they are mixed together, which makes the manuscript hard to follow.
2. The fonts in the figures are too small and difficult to read.
3. The English needs further improvement and a thorough grammar check. For instance, the sentence in Line 106 (page 2, left column) is very confusing and its meaning is unclear. "for tasks where the S2H generalization failed in both modalities, the same idea as (i) led to S2H generalization in the image modality after text-only supervision was used to inject reasoning capability on HARD task in text modality", it's hard to understand what's the meanning."

**Other Strengths And Weaknesses:**

Strengths:
1. The abstract is well written and easy to read.

Weaknesses:
1. The main sections are not clearly defined, and the English needs improvement.
2. The paper deals with knowledge distillation and multimodal chain-of-thought but does not compare or discuss existing methods.
3. The datasets are self-created without detailed descriptions, and public benchmarks are not used.

**Questions For Authors:**

1. The paper transfers the reasoning process (Chain-of-Thought) from LLM to VLM through tuning. However, it is unclear how this approach differs from [1].
2. Since the work relates to multimodal Chain-of-Thought and knowledge distillation, it should discuss related work in these areas, such as the findings in "Multimodal Chain-of-Thought Reasoning in Language Models" [1].

[1] Multimodal Chain-of-Thought Reasoning in Language Models

**Relation To Broader Scientific Literature:**

The paper studies how vision-language models (VLMs) reason with images, similar to the way large language models (LLMs) reason. It is relevant to chain-of-thought, prompt engineering, and knowledge distillation.

**Theoretical Claims:**

The paper does not present theoretical claims or proofs; it relies on experimental findings.

---

> ### Author Rebuttal · Authors · 2025-03-31
>
> # Authors' Response to Reviewer d6eT
>
> We thank the reviewer for their comments and suggestions. Please find our responses to your comments below.
>
> > Is the paper related to knowledge distillation or prompt engineering?
>
> We would like to clarify that **we do not employ any knowledge distillation**. All of our data (image, its text description, the CoT trace, and the solution) is generated with a Python script. When we say the reasoning transfers from the LLM to a VLM, we mean that the innate (or learned) reasoning capability of the LLM backbone (i.e., Llama-3-8B-Instruct) of an adapter-based VLM helps the entire VLM learn to perform the same task in the image modality. There is no external LLM/VLM here.
>
> > The datasets are self-created without detailed descriptions, and public benchmarks are not used
>
> We propose to measure the modality imbalance in VLMs by looking at the different simple-to-hard generalization behaviors (e.g., length generalization, compositional generalization). For this, we need a dataset where 1) there is a clear level of difficulty and 2) each image has an equivalent text description. There is no public dataset that fits this description. We also mention in the Related Works paragraph that “current VLM benchmarks are often solvable without the visual input,” which makes our analysis impossible. Additionally, we provide details in generating the datasets in Appendix D and provide the example data points in Figure 32-37. We were planning on releasing the entire codebase with the final version of the paper, but here is an anonymous link to the code to generate the data: [https://github.com/asjdifpjadsi/VLM_S2H](https://github.com/asjdifpjadsi/VLM_S2H).  Also see our response to Q1 of Reviewer NF8o.
>
> > Please provide Comparisons with “Multimodal chain-of-thought in Language Models”
>
> We thank the reviewer for the suggestion. We would like to clarify that our paper focuses on **modality imbalance** and its measurement through **simple-to-hard generalization**, aiming to improve a VLM’s reasoning ability on images to a comparable reasoning performance on **equivalent** text data. We would like to reiterate that  **we do not employ any knowledge distillation**.
> - Firstly, although both Multimodal-CoT and our paper use CoT to boost the model’s reasoning capability, we would like to point out that it is a very common technique in practice.
> - Multimodal-CoT would be mostly comparable to our Image-via-Text Supervision as both attempt to leverage extra text generation to assist reasoning. However, a significant difference is that Multimodal-CoT optimizes for CoT while ours only relies on CoT with a fixed template to assist long-chain reasoning.
> - Furthermore, our method converts an image to an **equivalent** text – a step that is later internalized – which is not equivalent to (either human-annotated or AI-generated) CoT. This conversion introduces no new information, so the help of text conversion in the model’s reasoning, if there is any, is entirely from a difference in modality, as opposed to a more optimal reasoning trajectory. Our work is more aligned with the idea of vision-depicting-prompting in Zhang et al. (2023) and many more we cited in the Related Works section (Appendix B, mentioned in page 8). Instead of prompting, we perform SFT and **propose a testbed to quantify** the benefit of an extra step of text conversion in mitigating **modality imbalance** in terms of **S2H generalization**.
>
>
> # Comments on Presentation / Figures
> > English needs further improvement and a thorough grammar check
>
> We thank the reviewer for the comment. We have taken a lot of care when writing the paper. We will modify any remaining grammatical mistakes, if any, in our final version.
>
> [1] Zhang et al., Lost in Translation: When GPT-4V(ision) Can’t See Eye to Eye with Text A Vision-Language-Consistency Analysis of VLLMs and Beyond, 2023.
>
> [2] Zhang et al., Multimodal Chain-of-Thought Reasoning in Language Models, 2023.

---

### Official Review · Reviewer_pko5 · 2025-03-15

**Overall Recommendation:** 3

**Summary:**

This work investigates the modality imbalance in simple-to-hard generalization of VLMs. The main findings are: Explicit image-to-text conversion is important in improving S2H generalization on images, and the conversion can be internalized at test time.

## update after rebuttal
The rebuttal partly solves my concerns. I will maintain my evaluation.

**Claims And Evidence:**

Most claims are well supported by experimental results, while it can be improved by showing consistent results with other base models of different sizes and backbones.

**Essential References Not Discussed:**

- There are missing related works that directly relate to modality imbalance and VLM evaluation, including but not limited to:
    - https://arxiv.org/abs/2404.01266

**Experimental Designs Or Analyses:**

This work can benefit from more experiments with different settings of the threshold to split simple and hard examples, as well as insights from the “hardness” (degree of how hard it is) of the examples.

**Methods And Evaluation Criteria:**

The methodology is well designed and the evaluation metrics are appropriately selected.

**Other Comments Or Suggestions:**

- Although the authors pointed out using only one base model is a limitation, I would like to mention it here again for visibility. This work can be significantly strengthened if the authors can show the main results and the conclusion with other base models with different backbones and sizes. Especially when models scale up, findings might not be 100% consistent.
- I understand that the authors want to compress as much crucial results and discussions as possible into the main content. However, my feeling of reading this paper is that too many concepts/discussions/paragraphs/subtitles seem to be equally important, but they are in fact not. I am not opposed to any style of writing, but the authors can do a better job to help the audience focus on the main storyline.
- Note that not everyone will read the appendix, so the main content must be self-contained. Without a proper related work SECTION, it’s hard to position this work among many other works in the literature. For example, how did previous works approach S2H generalization in general? Any of them worth being used as a baseline (only for the text supervision)? Such questions need to be answered in the main content to make it self-contained.

**Other Strengths And Weaknesses:**

- The choice of threshold to split SIMPLE and HARD needs to be clarified.
- From the examples in Figure 32-33, the simple and hard table readout tasks are more like scaling up the recognition and counting complexity, rather than requiring significantly better reasoning capabilities, e.g., chess puzzles.
- This paper is generally well written and easy to follow. However, it’s hard for the readers to get used to the many interchangeably used “image” and “text”. For example, I find it difficult to distinguish whether the “text/image” denotes S2H setting or training strategy in Figure 2. It’s probably better to rename the types of supervision, especially (a) and (b).
- The study on loss dynamics and gradient alignment is particularly interesting and insightful

**Questions For Authors:**

How would the “hardness” impact the performance? How well can it generalize from simple to relatively hard vs extremely hard? Can relatively hard generalize to extremely hard?

**Relation To Broader Scientific Literature:**

The finding that image-to-text conversion can be internalized at test time is particularly insightful, which could potentially benefit further research in Mechanistic Interpretability

**Theoretical Claims:**

I tried to check the proof in the Appendix F, I did not find obvious issues about it. But it is possible that I overlook some details.

---

> ### Author Rebuttal · Authors · 2025-03-31
>
> # Authors' Response to Reviewer pko5
>
> We thank the reviewer for their thoughtful comments and suggestions regarding the paper. Please find our responses to your comments below.
>
> # Weaknesses
> > Experiments are limited to one base model
>
> Please see our reply to Reviewer fXux. We observe the same conclusion from Qwen 2.5 VL 3B and 7B.
>
> > The simple and hard table readout tasks are more like scaling up the recognition and counting complexity, rather than requiring significantly better reasoning capabilities, e.g., chess puzzles.
>
> We agree that the reasoning in Table Readout is simpler compared to e.g., solving chess puzzles. However, such simple synthetic tasks allow us to clearly define SIMPLE and HARD examples and enable a more controlled analysis of model behavior.
> That being said, Table Readout does not only involve recognition and counting, but is analogous to Grid Navigation. At any cell, the model needs to know where to read the next number from. E.g., to realize the next number is on the left, the model needs to check that up/right/down is not highlighted but left is. This decision process mirrors the one in Grid Navigation, where all non-highlighted cells are “obstacles” to avoid.
>
> > Please discuss IsoBench
>
> Thank you for suggesting the reference. We will include the citation in lines 020-021 (page 1, right).
>
> # Questions
> > How would ‘Hardness’ impact the performance? The choice of threshold to split SIMPLE and HARD needs to be clarified.
>
> As S2H generalization measures OOD performance, “hardness” is relative with respect to the difference between training and test distributions and is model-specific. On Consecutive Table Readout, we consider **two hardness levels**: HARD (15-20) and (25–30), each reading 15-20 or 25–30 consecutive numbers (line 150) and find **modality gap gets wider as the task becomes more challenging** (line 210-212, Fig. 2). For the other three tasks that also require compositional generalization, we decided the hardness by either increasing the number of components to be composed (Table Readout and Grid Navigation) or introducing held-out compositions (Visual Analogy).
>
> We agree that ultimately the SIMPLE-HARD split is a matter of judgment. We tried to choose the most intuitive splits between SIMPLE and HARD while being aligned with existing studies (e.g., Hill et al., 2019 and Barrett et al., 2018). Further justifications for these decisions are in Appendix D and the specific thresholds in Table 1.
>
> > How did previous works approach S2H generalization in general? Any of them worth being used as a baseline (only for the text supervision)?
>
> In Appendix B, we discuss several prior works on S2H generalization that explore ICL with scratchpads (Anil et al., 2022), positional encodings (Kazemnejad et al., 2024), train set priming (Jelassi et al., 2023), curriculum learning (Abbe et al., 2024), and looped transformers (Fan et al., 2024). However, as we focus on modality imbalance, we do not compare with these methods. The aforementioned works (1) focus only on LLMs and are not directly adaptable to a multimodal setup and (2) study only length generalization, but not compositional generalization.
> More recent efforts such as self-improvement (Lee et al., 2025) and generalizable verifiers (Sun et al., 2024) examine S2H generalization in different setups that rely on curriculum learning with progressively harder tasks or require reward models. In contrast, our study is restricted to supervised fine-tuning (SFT) approaches.
> While we acknowledge that Mix or Mix+ may not be the optimal strategies for improving image generalization on our proposed tasks (line 429), it is still notable that reasoning transfer from the text to image modality—enabling S2H generalization on images—emerges naturally through autoregressive SFT. This, in our view, is both non-trivial and interesting.
> We will include an explicit Related Works section in the main paper to clearly differentiate our contributions from prior work.
>
> # Comments on Presentation / Figures
> We appreciate the reviewer’s detailed feedback on this matter. We hope to address these issues in the final version of the paper.
>
> >  The authors interchangeably use  “image” and “text”, which might create confusion
>
> To reduce the number of new terms introduced, we used “Text/Image” (capitalized) for the supervision that trains on the corresponding modality. However, we understand the potential for confusion.
>
> >  The authors can do a better job to help the audience focus on the main storyline
>
> In Section 1.1, we presented a compressed overview of the paper to direct the readers to relevant sections. In the final version, we plan to trim some of the ablations but instead expand on the discussions that better highlight the main storyline.
>
> [1] Lee, et al., Self-Improving Transformers Overcome Easy-to-Hard and Length Generalization Challenges, 2025
>
> all other citations in Related Works / References of the paper

---

### Official Review · Reviewer_fXux · 2025-03-15

**Overall Recommendation:** 4

**Summary:**

This paper investigates the "modality imbalance" problem in Vision Language Models (VLMs), where models perform worse on visual reasoning tasks compared to equivalent text-based tasks. The authors introduce a framework for studying simple-to-hard (S2H) generalization in VLMs using three synthetic tasks: Table Readout, Grid Navigation, and Visual Analogy. Each task has SIMPLE and HARD versions with equivalent representations in both text and image modalities.

The paper's main contribution is the discovery that VLMs exhibit modality imbalance in S2H generalization, with models able to generalize from SIMPLE to HARD examples in text but failing to do so in vision. Through experiments and gradient alignment analysis, the authors reveal that explicit image-to-text conversion is crucial for transferring reasoning capabilities from text to image modalities. They find that this conversion process can be internalized at test time, and identify gradient alignment measures that predict the effectiveness of different training strategies. Their work provides insights into the mechanisms of cross-modal learning and how the modality gap can be bridged through different training approaches.

**Claims And Evidence:**

The claims made in this paper are well-supported by comprehensive empirical evidence. The authors:

1. Demonstrate the modality imbalance problem through controlled experiments on their synthetic tasks, showing significant performance gaps between text and image modalities
2. Show that their proposed Mix and Mix+ strategies improve S2H generalization in images by effectively transferring reasoning capabilities from text to vision.
3. Provide gradient alignment analysis that convincingly explains why these training strategies work.
4. Validate their approaches across multiple tasks with varying complexity, showing consistent improvements over baselines.
5. Conduct thorough ablation studies on key components like chain-of-thought reasoning, data composition, and text warm-up pretraining.

The authors are careful to test their approaches in both standard and challenging scenarios (missing modalities). The evidence presented is comprehensive and the conclusions are well-supported by the experimental results.

**Essential References Not Discussed:**

The literature review seems comprehensive and contains all the relevant references. The authors (in the main text and Appendix B.) have appropriately cited related works in modality imbalance, generalization transfer between input modes, and S2H generalization.

**Experimental Designs Or Analyses:**

The experimental designs are sound and well-executed. The authors use only one model architecture (Eagle-X2-LLAMA3-8B), which is a limitation but understandable given computational constraints.

Their synthetic tasks appear simple, but this simplicity actually strengthens the study by enabling precise isolation of reasoning capabilities and clear comparisons across modalities. The authors carefully control for data quantity and quality throughout, ensuring valid comparisons and reliable conclusions.

**Methods And Evaluation Criteria:**

The methods proposed in this paper are novel and well-designed. The authors:

1. Create three synthetic tasks that allow for controlled study of S2H generalization in both text and image modalities, with difficulty levels that can be systematically tuned to test generalization capabilities.
2. Develop multiple training strategies (Mix, Mix+, Align-Mix+) that progressively address more challenging S2H generalization scenarios.
3. Introduce gradient alignment measures that provide theoretical insights into the effectiveness of their approaches.

The evaluation criteria are appropriate and comprehensive:

- Performance is measured on both SIMPLE and HARD examples across modalities
- Comparisons include baselines and ablations to isolate the impact of each component
- Gradient analysis provides mechanistic understanding of the approaches

The authors carefully control for factors like training data size to ensure fair comparisons. They also test on both complete and missing modality scenarios to demonstrate the robustness of their approach. Additionally, they provide extensive ablation studies and supplementary experiments in the Appendix, including text warm-up pretraining, explicit and implicit CoT ablation, and multi-task training effects, which further substantiate their claims and provide deeper insights into the mechanisms behind their approaches.

**Other Comments Or Suggestions:**

I wonder if computing gradient alignment scores between modalities (rather than just between SIMPLE and HARD examples within a modality) might offer valuable insights. This cross-modal gradient alignment could potentially reveal how information transfers between vision and language representations for identical inputs presented in different formats.

In section 4.2 discussing the benefits of two-phase training, it would be helpful to cite the specific figure or table showing the 76%, 96%, and 56% performance metrics mentioned. This would make it easier for readers to connect your analysis with the supporting evidence.

In Figure 6, you only showed results for Table Readout and Visual Analogy. Could you also show the remaining task (Grid Navigation) to provide a complete picture of how Image-via-Text+ performs across all tasks?

I am a bit confused about the naming here : are “Consecutive Table Readout” the same as “Table Readout”? Appendix D suggests that they are a simliar but different tasks. However, in the Abstract, you state that you test on three tasks, which make this even more confusing (is the Consecutive Table Readout a “fourth” task?). Clarifying their difference in the main text would help with clarity.

I'm particularly intrigued by your findings on Chain of Thought reasoning. Given your ablation studies in Appendix I.7 showing that attempts to internalize CoT consistently fail to achieve image S2H generalization, could this suggest something fundamental about how reasoning occurs in these adapter-based VLMs? Perhaps the architectural design, with vision adapters attached to a text-based backbone, necessitates text-structured reasoning paths that CoT provides explicitly?

**Other Strengths And Weaknesses:**

**Strengths:**

1. The paper introduces a novel and practical approach to address an important problem in multimodal learning.
2. The synthetic tasks are well-designed to isolate and study S2H generalization.
3. The gradient alignment analysis provides valuable mechanistic insights.
4. The approaches are effective across a range of tasks and scenarios.
5. The paper includes thorough ablation studies and analyses.
6. The proposed methods (Mix, Mix+, Align-Mix+) are simple yet effective, making them likely to be adopted in practice.

**Weaknesses:**

1. The experiments are limited to one model architecture
2. While the synthetic tasks are useful for controlled study, more real-world examples would strengthen the paper's impact.
3. The paper focuses mainly on vision-text modalities; it would be interesting to see if the insights generalize to other modality pairs.
4. The approach requires task-specific chain-of-thought templates, which might limit its generality.

Overall, the strengths significantly outweigh the weaknesses, and the paper makes a valuable contribution to the field.

**Questions For Authors:**

I may have missed it, but it seems that Figures 5 and 7 themselves are in the main text but not the figures are not referenced anywhere in the main text, making it a bit confusion to read. (For example, it seems that the last parts of Page 6 are referring to Figure 7, but this is never mentioned in the main text)

Regarding Figure 2, I noticed the grey dotted line representing text supervision on the right panels lacks a corresponding entry in the legend, though it's explained in the caption. Adding this to the legend (with a name such as “Text Supervision on Text”) would improve readability at a glance.

Similarly, I believe that the description of “Text” (grey dash) in Figure 5 could be more clearly stated. In the caption, it says that they are S2H generalization on text from Text supervision, but the legend seems to imply that “Text” is S2H generalization on Image from Text Supervision. As with the Figure 2, I believe giving the “Text” legend name a clearer name (e.g., “Text supervision on Text”) would be better.

In Figure 8 in the legend, there is “Mix → Mix+”, but there is no explanation of what this means.

**Relation To Broader Scientific Literature:**

This paper bridges several key research areas in multimodal learning. It introduces controlled algorithmic “visual” reasoning tasks that enable precise multi-step reasoning evaluation, addressing a significant gap in VLM assessment. (add sth like : Furthermore, they allow variable difficulty) It extends work on modality imbalance by uniquely addressing this issue in adapter-based VLMs through knowledge transfer from text to image modalities, moving beyond traditional approaches such as modulating learning rates (Peng et al., 2022). Finally, it extends simple-to-hard generalization research from language-only settings (Abbe et al., 2024; Zhou et al., 2024) to cross-modal contexts.

Peng et al., 2022 : Peng, Xiaokang, et al. "Balanced multimodal learning via on-the-fly gradient modulation." *Proceedings of the IEEE/CVF conference on computer vision and pattern recognition*. 2022.

Abbe et al., 2024 : Abbe, Emmanuel, et al. "Generalization on the unseen, logic reasoning and degree curriculum." *Journal of Machine Learning Research* 25.331 (2024): 1-58.

Zhou et al., 2024 : Zhou, Hattie, et al. "What algorithms can transformers learn? a study in length generalization." *arXiv preprint arXiv:2310.16028* (2023).

**Theoretical Claims:**

I reviewed the theoretical claims related to gradient alignment in Section 5. The proofs for Theorem H.1 appear sound. The theorem establishes that (given a small enough learning rate) the gradient alignment score predicts how much loss reduction on HARD Image examples will be achieved by gradient updates from SIMPLE Image examples relative to updates from HARD Image examples directly.

However, as the main novelty is on the transfer between HARD text and HARD image examples (without training these HARD image examples), I am not sure whether the analysis is thorough enough. Cross-modality alignment should be examined.

---

> ### Author Rebuttal · Authors · 2025-03-31
>
> # Authors' Response to Reviewer fXux
>
> We thank the reviewer for their careful review of our paper. Please find responses to your concerns and questions below.
>
> # Weaknesses
> > Experiments are limited to one architecture
>
> We used EAGLE because it was the best performing open-source model which also released all details on **its training data and the set of hyperparameters**.
>
> To test the generalizability of our results, we ran additional experiments by continual finetuning 3B, 7B models from the Qwen2.5-VL family. See the accuracy on HARD-image below. CTR, TR, GN, VA are respectively short for Consecutive Table Readout, Table Readout, Grid Navigation, and Visual Analogy. On the text version of CTR, we notice that the 3B model doesn’t show S2H generalization (which is expected since task difficulty can be model specific). However, across TR, GN, and VA, we generally observe the same conclusion - **Mix+ achieves S2H generalization on the image modality and Align-Mix+ improves the generalization**.
>
> ### Qwen2.5-VL-3B-Instruct
> |Supervision|CTR(30k)|TR(30k)|TR(60k)|GN(30k)|GN(60k)|VA(30k)|VA(60k)|
> |:-|-:|-:|-:|-:|-:|-:|-:|
> |Image|0|11|10|22|22|0|0|
> |Text+Image|1|7|6|0|14|0|0|
> |Image-via-Text|1|12|8|13|14|0|0|
> |Mix|1|11|10|14|16|0|1|
> |Image-via-Text+|0|**81**|90|67|58|**48**|**48**|
> |Mix+|**4**|78|86|77|91|20|27|
> |Align-Mix+|-|66|**91**|**80**|**91**|38|42|
>
> ### Qwen2.5-VL-7B-Instruct
> |Supervision|CTR(30k)|TR(30k)|TR(60k)|GN(30k)|GN(60k)|VA(30k)|VA(60k)|
> |:-|-:|-:|-:|-:|-:|-:|-:|
> |Image|0|18|17|14|29|0|0|
> |Text+Image|4|8|5|6|11|0|0|
> |Image-via-Text|36|9|13|13|18|0|0|
> |Mix|52|8|17|15|12|0|0|
> |Image-via-Text+|**73**|82|88|**75**|67|**41**|**44**|
> |Mix+|72|13|66|69|**85**|12|17|
> |Align-Mix+|-|**93**|**92**|36|58|25|34|
>
> > Experiments are limited to synthetic settings
>
> Please see our reply to the first question from Reviewer NF8o.
>
> > Experiments are only on vision-text modalities
>
> Our motivation came from works that identify the modality gap in VLMs. Additionally, we require an open-weights model that has strong enough reasoning capability in each modality, which is available in a VLM. We would love to explore more modalities in the future when open models can also incorporate multiple modalities.
>
> > Task-specific chain-of-thought templates might limit generality of observations
>
> By keeping the chain-of-thought templates fixed throughout the task, we were able to accurately probe the existing modality imbalance. In the future, we would like to explore more on how the results would generalize to a more flexibly generated CoT trace.
>
> # Suggested Questions
>
> > Can gradient alignment be used to measure cross-modal alignment?
>
> We found that different supervision strategies could be distinguished only when analyzing gradient alignment between SIMPLE and HARD examples within each modality independently. This may be attributed to our use of a local definition of gradient alignment—examining gradients at each step in isolation. Capturing cross-modal alignment likely requires a more global perspective, as gradients at earlier time steps (e.g., on text) can influence gradients in later steps (e.g., on image). We consider the development of such alignment measures an important direction for future work.
>
> > Consecutive Table Readout vs. Table Readout
>
> Both tasks involve sequentially reading highlighted cells in a table. In Consecutive Table Readout, the path is always consecutive row-wise, and the model knows where the next cell should be. In Table Readout, the path can take turns in arbitrary directions at arbitrary locations, so the model needs to additionally check **which adjacent cell is highlighted** at every location.
>
> In the final version, we will list all 4 tasks in the abstract or rename Table Readout to e.g., Path Table Readout and use Table Readout as an umbrella term for both Consecutive and Path Table Readout.
>
> > Why does internalizing CoT fail
>
> We observed that CoT is crucial for the model to transfer its reasoning from text to image input by repeating the same reasoning steps. Our hypothesis is that during SFT, the model establishes an implicit equivalence between reasoning on text and image inputs via the common CoT tokens.
>
> There may exist multiple ways, e.g., a more fine-grained curriculum, to internalize CoT. For the scope of this paper, we limit our exploration to simple settings with randomly shuffled data and progressive internalization, and leave a complete study for future.
>
> > Can you include results on Grid Navigation in Figure 6
>
> The model already performs nearly perfectly on Grid Navigation with 60k examples of Mix+, so we did not compare with Image-via-Text+.
>
> > What is the meaning of Mix$\to$Mix+ in Figure 8
>
> Mix$\to$Mix+ means we took an intermediate checkpoint of Mix and resumed training with Mix+. This setup allows us to assess the effect of introducing Hard text examples on the loss curves of Mix training, enabling a fine-grained analysis of the differences between Mix and Mix+ (lines 334-341).

---

> > ### Comment · Reviewer_fXux · 2025-04-05
> >
> > I am generally satisfied with the author’s response and have raised my score. Despite some limitations regarding the use of synthetic tasks, I believe the paper provides meaningful contributions to the field.
> >
> > That being said, I hope the authors enhance their presentation. In particular, although (as the authors responded) Mix $\rightarrow{}$ Mix is explained in lines 334-341, the text in those lines does not explicitly call them “Mix $\rightarrow{}$ Mix”. Also, although the authors didn’t respond, there is still the issue of figures in the main text (Figures 5 and 7) not being mentioned in the main text.
> >
> > Raising the score to 4 (Accept).

---

> > > ### Author Response · Authors · 2025-04-05
> > >
> > > We thank the reviewer for carefully evaluating our manuscript and our response to the reviewer's concerns.
> > >
> > > We especially thank the reviewer for commenting on how we can improve the presentation of our work, and we will try our best to incorporate the comments into the final version of the paper. Due to the character limit, we were initially not able to respond to the reviewer's feedback on Figures 5, 7 not being mentioned in the main text. We will resolve this matter as well.
> > >
> > > If possible, could the reviewer edit the original review to reflect the updated score? Thanks!

---

### Decision · Program_Chairs · 2025-05-01

**Decision:**

Accept (poster)

**Comment:**

The submission introduces a synthetic framework for assessing the ability of VLMs to perform algorithmic visual reasoning.  Reviewers in general like the setup and believe the submission is a meaningful contribution to the community.  Some concerns, in particular those about the experimental validation, are mostly addressed in the rebuttal.  One reviewer remained negative, though they did not provide any justifications beyond minor language problems; their recommendation is therefore ignored.  Given all these considerations, the AC would like to recommend acceptance.  The authors should revise the submission based on the reviews and rebuttal in the camera ready.